# How Expressive are Knowledge Graph Foundation Models?

Xingyue Huang [1]  Pablo Barceló [2 3 4]  Michael M. Bronstein [1 5]  İsmail İlkan Ceylan [1]  Mikhail Galkin [6]
Juan L Reutter [2 3 4]  Miguel Romero Orth [2 4]

## Abstract

Knowledge Graph Foundation Models (KGFMs) are at the frontier for deep learning on knowledge graphs (KGs), as they can generalize to completely novel knowledge graphs with different relational vocabularies. Despite their empirical success, our theoretical understanding of KGFMs remains very limited. In this paper, we conduct a rigorous study of the expressive power of KGFMs. Specifically, we show that the expressive power of KGFMs directly depends on the *motifs* that are used to learn the relation representations. We then observe that the most typical motifs used in the existing literature are *binary*, as the representations are learned based on how pairs of relations interact, which limits the model's expressiveness. As part of our study, we design more expressive KGFMs using richer motifs, which necessitate learning relation representations based on, e.g., how triples of relations interact with each other. Finally, we empirically validate our theoretical findings, showing that the use of richer motifs results in better performance on a wide range of datasets drawn from different domains.

## 1. Introduction

Knowledge Graph Foundation Models (KGFMs) gained significant attention (Galkin et al., 2024; Mao et al., 2024) for their success in link prediction tasks involving *unseen entities and relations* on knowledge graphs (KGs). These models aim to generalize across different KGs by effectively learning *relation invariants*: properties of relations that are transferable across KGs of different relational vocabularies (Gao et al., 2023). KGFMs learn representations of relations by relying on their structural roles in the KG,

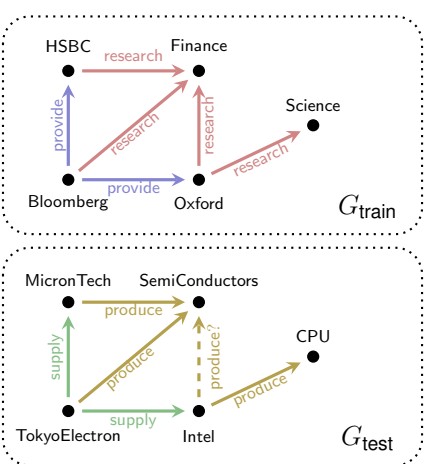

*Figure 1.* Training is over relations provide and research, and testing is over structurally similar relations supply and produce.

which can be transferred to novel relations based on their "structurally similarities" to the observed relations.

**Example 1.1.** Consider the two KGs from Figure 1 which use disjoint sets of relations. Suppose that the model is trained on $G_{\text{train}}$ and the goal is to predict the missing link produce(Intel, SemiConductors) in $G_{\text{test}}$. Ideally, the model will predict the link by learning relation invariants that map produce $\mapsto$ research and supply $\mapsto$ provide, as the *structural role* of these relations are similar in the respective graphs, even if the relations *themselves* differ. △

The dominant approach (Geng et al., 2023; Lee et al., 2023; Galkin et al., 2024) for computing relation invariants is by learning relational embeddings over a *graph of relations*. In this graph, each node represents a relation type from the original KG and each edge represents a connection between two relations. One prominent choice of these connections is to identify whether two relations in the original KG match a set of shared patterns known as *graph motifs*: small, recurring subgraphs that capture essential relational structures within the KG. Figure 2 depicts a simple motif: a path of length two, connecting three entities via two relations (i.e., a *binary*

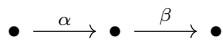

*Figure 2.* A simple motif of a path of length two.

[1]University of Oxford [2]Universidad Católica de Chile [3]IMFD [4]CENIA [5]AITHYRA [6]Google. Correspondence to: Xingyue Huang <University of Oxford>.

*Proceedings of the 42nd International Conference on Machine Learning*, Vancouver, Canada. PMLR 267, 2025. Copyright 2025 by the author(s).

*motif*). For instance, the relations provide and research are connected via the entity Oxford in Figure 1, and hence match this motif with the map $\alpha \mapsto$ provide, $\beta \mapsto$ research.

We saliently argue that existing models limit themselves to binary motifs, which restricts their ability to capture complex interactions involving more relations. We investigate the expressive power of these models, specifically, how the choice of motifs impacts the model's ability to capture relation invariants and help distinguish between different relational structures. Prior works have extensively studied relational message-passing neural networks on KGs (Barceló et al., 2022; Huang et al., 2023), but these results do *not* apply to KGFMs, which is the focus of our work.

**Approach.** We introduce M̲OT̲if-I̲nduced F̲ramework for KGFMs (MOTIF) based on learning relation invariants from arbitrary (not necessarily binary) graph motifs. MOTIF first constructs a *relational hypergraph*, where each node corresponds to a relation in the original KG, and each *hyperedge* represents a match of relations to a motif in the original KG. Then, it performs message passing over the relational hypergraph to generate relation representations, which are used in message passing over the original KG to enable link prediction over novel relations. Importantly, MOTIF is a general framework that strictly subsumes existing KGFMs such as ULTRA (Galkin et al., 2024) and INGRAM (Lee et al., 2023) as detailed in Section 5, and provides a principled approach for enhancing the expressive power of other KGFMs such as TRIX (Zhang et al., 2024) and RMPI (Geng et al., 2023).

**Contributions.** Our contributions can be summarized as follows:

- We introduce MOTIF as a general framework capable of integrating *arbitrary* graph motifs into KGFMs, subsuming existing KGFMs such as ULTRA and INGRAM.

- To the best of our knowledge, we provide the *first* rigorous analysis on the expressive power of KGFMs. Our study explains the capabilities and limitations of existing KGFMs in capturing relational invariants.

- We identify a *sufficient* condition under which a newly introduced motif enhances the expressiveness of a KGFM within the framework of MOTIF. Using this theoretical recipe, we derive a new KGFM that is provably more expressive than ULTRA.

- Empirically, we conduct an extensive analysis on 54 KGs from diverse domains and validate our theoretical results through synthetic and real-world experiments.

All proofs of the technical results can be found in the appendix of the paper, along with the additional experiments.

## 2. Related work

**Transductive and inductive (on node) link prediction.** Link prediction on KGs has been extensively studied in the literature. Early approaches like TransE (Bordes et al., 2013), RotatE (Sun et al., 2019), and BoxE (Abboud et al., 2020) focus on the *transductive* setting, where the learned entity embeddings are fixed, and thus inapplicable to unseen entities at test time. Multi-relational GNNs such as RGCN (Schlichtkrull et al., 2018) and CompGCN (Vashishth et al., 2020) remain transductive as they store entity and relation embeddings as parameters. To overcome this limitation, Teru et al. (2020) introduce GraIL, which enables *inductive* link prediction via the *labeling trick*. NBFNet (Zhu et al., 2021), A*Net (Zhu et al., 2023), RED-GNN (Zhang & Yao, 2022), and AdaProp (Zhang et al., 2023), provide improvements by leveraging conditional message-passing, which is provably more expressive (Huang et al., 2023). These models, once trained, can only be applied to KGs with the same relational vocabulary, limiting their applicability to graphs with unseen relations.

**Inductive (on node and relation) link prediction.** INGRAM (Lee et al., 2023) was one of the first approaches to study inductive link prediction over both new nodes and unseen relations by constructing a weighted relation graph to learn new relation representations. Galkin et al. (2024) extended this idea with the ULTRA architecture, which constructs a multi-relational graph of fundamental relations and leverages conditional message passing to enhance performance. ULTRA was among the first KGFMs to inspire an entire field of research (Mao et al., 2024). Concurrently, RMPI (Geng et al., 2023) explored generating multi-relational graphs through local subgraph extraction while also incorporating ontological schema. Gao et al. (2023) introduced the concept of *double-equivariant* GNNs, which establish invariants on nodes and relations by leveraging subgraph GNNs in the proposed ISDEA framework to enforce double equivariance precisely. MTDEA (Zhou et al., 2023a) expands this framework with an adaptation procedure for multi-task generalization. Further, TRIX (Zhang et al., 2024) expands on ULTRA with recursive updates of relation and entity embeddings, and is the first work to compare expressivity among KGFMs. Finally, KG-ICL (Cui et al., 2024) introduced a new KGFM utilizing in-context learning with a unified tokenizer for entities and relations.

**Link prediction on relational hypergraphs.** Relational hypergraphs are a generalization of KGs used to represent higher-arity relational data. Work on link prediction in relational hypergraphs first focused on shallow embeddings (Wen et al., 2016; Liu et al., 2020; Fatemi et al., 2020), and later G-MPNN (Yadati, 2020) and RD-MPNNs (Zhou et al., 2023b) advanced by incorporating message passing. Recently, Huang et al. (2024) conducted an in-depth expressiv-

ity study on these models and proposed HCNets, extending conditional message-passing to relational hypergraphs and achieving strong results on inductive link prediction.

## 3. Preliminaries

**Knowledge graphs.** A *knowledge graph* (KG) is a tuple $G = (V, E, R)$, where $V$ is a set of nodes, $R$ is the set of relation types and $E \subseteq V \times R \times V$ is a set of labeled edges, or *facts*. We write $r(u, v)$ to denote a labeled edge, or a fact, where $r \in R$ and $u, v \in V$. A (potential) *link* in $G$ is a triple $r(u, v)$ in $V \times R \times V$ that may or may not be a fact in $G$. The *neighborhood* of node $v \in V$ relative to a relation $r \in R$ is $\mathcal{N}_r(v) := \{u \mid r(u, v) \in E\}$.

**Relational hypergraphs.** A *relational hypergraph* is a tuple $G = (V, E, R)$, where $V$ is a set of nodes, $R$ is a set of relation types, and $E$ a set of *hyperedges* (or *facts*) of the form $e = r(u_1, \ldots, u_k) \in E$ where $r \in R$ is a relation type, $u_1, \ldots, u_k \in V$ are nodes, and $k = \mathrm{ar}(r)$ is the arity of the relation $r$. We write $\rho(e)$ as the relation type $r \in R$ of the hyperedge $e \in E$, and $e(i)$ to refer to the node in the $i$-th arity position of the hyperedge $e$.

**Homomorphisms.** A (*node-relation*) *homomorphism* from a KG $G = (V, E, R)$ to a KG $G' = (V', E', R')$ is a pair of mappings $h = (\pi, \phi)$, where $\pi : V \to V'$ and $\phi : R \to R'$, such that for every fact $r(u, v)$ in $G$, the fact $\phi(r)(\pi(u), \pi(v))$ belongs to $G'$. The *image* $h(G)$ of $h$ is the knowledge graph $(\pi(V), h(E), \phi(R))$, where $h(E)$ contains the fact $\phi(r)(\pi(u), \pi(v))$ for each $r(u, v)$ in $G$.

**Isomorphism, relation invariants, link invariants.** An *isomorphism* from KG $G = (V, E, R)$ to KG $G' = (V', E', R')$ is a pair of bijections $(\pi, \phi)$, where $\pi : V \to V'$ and $\phi : R \to R'$, such that every fact $r(u, v)$ is in $G$ if and only if the fact $\phi(r)(\pi(u), \pi(v))$ is in $G'$. Two graphs are *isomorphic* if there is an isomorphism between them. For $k \geq 1$, a *$k$-ary relation invariant* is a function $\xi$ associating to each KG $G = (V, E, R)$ a function $\xi(G)$ with domain $R^k$ such that for every isomorphic KGs $G$ and $G'$, every isomorphism $(\pi, \phi)$, and every tuple $\bar{r} \in R^k$ of relations in $G$, we have $\xi(G)(\bar{r}) = \xi(G')(\phi(\bar{r}))$. A *link invariant* is a function $\omega$ assigning to each KG $G = (V, E, R)$ a function $\omega(G)$ with domain $V \times R \times V$ such that for every isomorphic KGs $G$ and $G'$, every isomorphism $(\pi, \phi)$, and every link $q(u, v)$ in $G$, we have $\omega(G)(q(u, v)) = \omega(G')(\phi(q)(\pi(u), \pi(v)))$.

For instance, a unary relation invariant $\xi$ evaluated on $G_{\text{train}}$ and $G_{\text{test}}$ in Figure 1 may satisfy $\xi(G_{\text{train}})(\mathsf{research}) = \xi(G_{\text{test}})(\mathsf{produce})$ if the two relations play analogous structural roles. Similarly, a link invariant $\omega$ may assign equal values to $\mathsf{research}(\mathsf{Oxford}, \mathsf{Finance})$ in $G_{\text{train}}$ and $\mathsf{produce}(\mathsf{Intel}, \mathsf{SemiConductors})$ in $G_{\text{test}}$ under a structure-preserving mapping.

## 4. MOTIF: A general framework for KGFMs

We present MOTIF, a very general framework for KGFMs. Given a KG $G = (V, E, R)$, MOTIF computes an encoding for each potential link $q(u, v)$ in $G$ following three steps:

1. LIFT: Use a set $\mathcal{F}$ of motifs to compute a relational hypergraph $\mathrm{LIFT}_{\mathcal{F}}(G) = (V_{\mathrm{LIFT}}, E_{\mathrm{LIFT}}, R_{\mathrm{LIFT}})$, using a procedure called LIFT.

2. RELATION ENCODER: Apply a relation encoder on the hypergraph $\mathrm{LIFT}_{\mathcal{F}}(G)$ to obtain relation representations.

3. ENTITY ENCODER: Use the relation representations from Step 2 and apply an entity encoder on the KG $G$ to obtain final link encodings.

The overall process is illustrated in Figure 3 and we now detail each of the steps.

### 4.1. LIFT

A graph *motif* is a pair $P = (G_M, \bar{r})$, where $G_M = (V_M, E_M, R_M)$ is a connected KG and $\bar{r}$ is a tuple defining an order on the relation set $R_M$. Specifically, *$k$-ary motif* is a motif with a motif graph containing exactly $k$ relation types, i.e., $|R_M| = k$. When $k = 2$, we call it *binary motif*. The information extracted by motifs is defined via homomorphism.

**Definition 4.1.** Let $G$ be a KG and $P = (G_M, \bar{r})$ be a motif with $\bar{r} = (r_1, \ldots, r_n)$. The *evaluation* $\mathrm{Eval}(P, G)$ of $P$ over $G$ is the set of tuples $(\phi(r_1), \ldots, \phi(r_n))$, for each homomorphism $h = (\pi, \phi)$ from $G_M$ to $G$. $\qquad \square$

**The LIFT operation.** Given a KG $G = (V, E, R)$ and a set of motifs $\mathcal{F}$, the operation $\mathrm{LIFT}_{\mathcal{F}}(G)$ constructs a *relational hypergraph* $(V_{\mathrm{LIFT}}, E_{\mathrm{LIFT}}, R_{\mathrm{LIFT}})$ as follows:

- The node set is $V_{\mathrm{LIFT}} = R$, where each node corresponds to a relation type from the original KG.

- The relation set is $R_{\mathrm{LIFT}} = \mathcal{F}$, where each motif $P \in \mathcal{F}$ is treated as a distinct relation type in the relational hypergraph.

- The hyperedge set $E_{\mathrm{LIFT}}$ is defined as

$$\{P(r_n, \ldots, r_n) \mid P \in \mathcal{F}, (r_n, \ldots, r_n) \in \mathrm{Eval}(P, G)\}$$

where $\mathrm{Eval}(P, G)$ denotes the set of all $k$-tuples of relations in $G$ that match the motif $P$.

### 4.2. Relation encoder

A *relation encoder* is a tuple $(\mathcal{F}, \mathsf{Enc}_{\mathrm{LIFT}})$, where $\mathcal{F}$ is a set of motifs and $\mathsf{Enc}_{\mathrm{LIFT}}$ computes relation representations for the set $R = V_{\mathrm{LIFT}}$ of nodes of $\mathrm{LIFT}_{\mathcal{F}}(G) =$

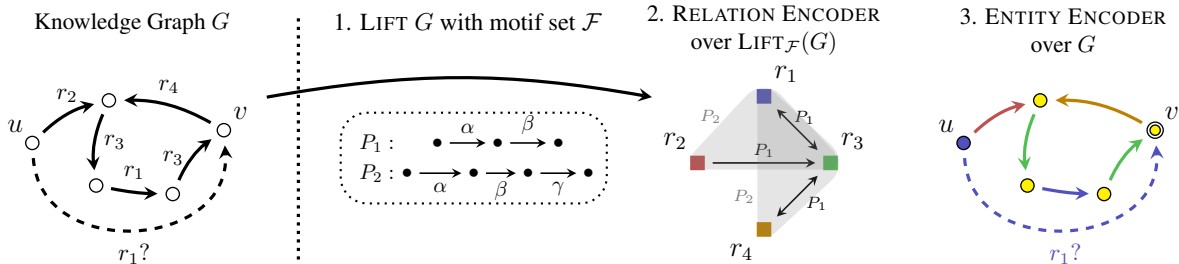

*Figure 3.* Overall framework of MOTIF($\mathcal{F}$) over the motif set $\mathcal{F}$: Given a query $r_1(u, v)$, MOTIF($\mathcal{F}$) first applies the LIFT$_\mathcal{F}$ operation to generate the relational hypergraph $(V_{\text{LIFT}}, E_{\text{LIFT}}, R_{\text{LIFT}})$ over the set of motifs $\mathcal{F}$. Then a relation encoder is applied to generate a relation representation (indicated by color) conditioned on query $r_1$, followed by an entity encoder that conducts conditional message passing.

$(V_{\text{LIFT}}, E_{\text{LIFT}}, R_{\text{LIFT}})$. Given that our aim is to encode links of the form $q(u, v)$, the encoding computed by $\mathsf{Enc}_{\text{LIFT}}$ for relation $r \in R$ is *conditioned* on the relation $q \in R$. These encodings are determined by a sequence of features $\boldsymbol{h}_{r|q}^{(t)} \in \mathbb{R}^{d(t)}$, for $0 \le t \le T$, defined iteratively as follows:

$$\boldsymbol{h}_{r|q}^{(0)} = \text{INIT}_1(q, r), \quad \boldsymbol{h}_{r|q}^{(t+1)} = \text{UP}_1(\boldsymbol{h}_{r|q}^{(t)}, \text{AGG}_1(M)),$$

where INIT$_1$, UP$_1$, AGG$_1$ are differentiable *initialization*, *update*, and *aggregation* functions, respectively, and:

$$M = \{\!\!\{\text{MSG}_{\rho(e)}(\{\!\!\{(\boldsymbol{h}_{r'|q}^{(t)}, j) \mid \\ (r', j) \in \mathcal{N}^i(e)\}\!\!\}) \mid (e, i) \in E_{\text{LIFT}}(r)\}\!\!\}.$$

In this setting, $\text{MSG}_{\rho(e)}$ is a motif-specific *message* function and we further define $E_{\text{LIFT}}(r) = \{(e, i) \mid e(i) = r, e \in E_{\text{LIFT}}, 1 \le i \le \text{ar}(\rho(e))\}$ as the set of edge-position pairs of a node $r$, and $\mathcal{N}^i(e)$ as the *positional neighborhood* of a hyperedge $e$ with respect to a position $i$: $\mathcal{N}^i(e) = \{(e(j), j) \mid j \ne i, 1 \le j \le \text{ar}(\rho(e))\}$. We use $\mathsf{Enc}_{\text{LIFT},q}[r]$ to denote the final relation encoding of a relation $r$ in $G$ when conditioned on $q$, that is, the last layer vector $\boldsymbol{h}_{r|q}^{(T)} \in \mathbb{R}^{d(T)}$.

**Remark 1.** Observe that the relation encoder computes binary relation invariants provided INIT$_1$ is an invariant.

### 4.3. Entity encoder

MOTIF uses an entity encoder to compute a link-level representation of a KG, regardless of its relation vocabulary. This works as follows. MOTIF is defined as a tuple $(\mathsf{Enc}_{\text{KG}}, (\mathcal{F}, \mathsf{Enc}_{\text{LIFT}}))$, where $(\mathcal{F}, \mathsf{Enc}_{\text{LIFT}})$ is a relation encoder and $\mathsf{Enc}_{\text{KG}}$ is an entity encoder that computes node representations over KGs of the form $G = (V, E, R)$, conditioned on a node $u \in V$ and the relation $q$, according to the following iterative scheme:

$$\boldsymbol{h}_{v|u,q}^{(0)} = \text{INIT}_2(u, v, q), \quad \boldsymbol{h}_{v|u,q}^{(\ell+1)} = \text{UP}_2(\boldsymbol{h}_{v|u,q}^{(\ell)}, \text{AGG}_2(N)),$$



| | | |
|---|---|---|
| *h2t*: | $(\alpha, \beta)$ | $\bullet \xrightarrow{\alpha} \bullet \xrightarrow{\beta} \bullet$ |
| *t2h*: | $(\alpha, \beta)$ | $\bullet \xleftarrow{\alpha} \bullet \xleftarrow{\beta} \bullet$ |
| *t2t*: | $(\alpha, \beta)$ | $\bullet \xrightarrow{\alpha} \bullet \xleftarrow{\beta} \bullet$ |
| *h2h*: | $(\alpha, \beta)$ | $\bullet \xleftarrow{\alpha} \bullet \xrightarrow{\beta} \bullet$ |



*Figure 4.* The 4 motifs used by ULTRA are shown here. *h2t* represents "head-to-tail", with the others following similarly. Note that *h2t* and *t2h* are isomorphic but with different relation ordering.

where INIT$_2$, UP$_2$, AGG$_2$, and MSG are differentiable *initialization*, *update*, *aggregation* and *message* functions, respectively, $v$ is an arbitrary node in $V$, and:

$$N = \{\!\!\{\text{MSG}(\boldsymbol{h}_{w|u,q}^{(\ell)}, \mathsf{Enc}_{\text{LIFT},q}[r]) \mid w \in \mathcal{N}_r(v), r \in R\}\!\!\}.$$

We use $\mathsf{Enc}_{\text{KG},u,q}[v]$ to denote the encoding of $v$, conditioned on relation $q$ and source node $u$, which corresponds to $\boldsymbol{h}_{v|u,q}^{(L)} \in \mathbb{R}^{d(L)}$ ($L$ being the number of layers). Finally, a unary decoder $\mathsf{Dec} : \mathbb{R}^{d(L)} \mapsto [0, 1]$ is applied on $\mathsf{Enc}_{\text{KG},q,u}[v]$ to obtain the probability score for the existence of the potential link $q(u, v)$.

**Remark 2.** Observe that the entity encoder computes link invariants provided INIT$_2$ is an invariant.

## 5. KGFMs captured by MOTIF

We revisit the ULTRA architecture (Galkin et al., 2024) (See details in Appendix F). The four fundamental relations used in ULTRA can be interpreted as the graph motifs illustrated in Figure 4. Hence, any ULTRA architecture can be represented as a MOTIF architecture, with $\mathsf{Enc}_{\text{LIFT}}$ and $\mathsf{Enc}_{\text{KG}}$ being specific variants of NBFNets (Zhu et al., 2021).

MOTIF can also represent a slight variant of the INGRAM architecture (Lee et al., 2023) when we replace the constructed weighted relation graph with a KG. Here the construction of

LIFT uses a graph motif set $\mathcal{F} = \{h2h, t2t\}$, with $\mathsf{Enc}_{\text{LIFT}}$ and $\mathsf{Enc}_{\text{KG}}$ being variants of GATv2 (Brody et al., 2022), and replace the random initialization in INIT₁ by an invariant function. Note that the definition of MOTIF allows for unconditional message passing (Huang et al., 2023) when both INIT₁ and INIT₂ are agnostic to the query $q$.

We also note that, while other KGFMs like RMPI (Geng et al., 2023) and TRIX[1] (Zhang et al., 2024) do not technically fall under the framework of MOTIF, these models do rely on a LIFT operation to construct a graph of relations based on a predefined set of motifs, followed by relation and entity encoders. In particular, RMPI constructs an alternative relation graph inspired by the line graph, incorporating the motifs used in ULTRA along with two additional motifs, PARA and LOOP. In turn, TRIX considers the same set of motifs as ULTRA but employs alternating updates on entity and relation representations.

# 6. Expressive power

MOTIF computes encodings for links in KGs. In this section, we aim to understand which links MOTIF can separate and whether equipping MOTIF with more graph motifs increases its separation power. Hence, our focus is on the potential power obtained when using a particular set $\mathcal{F}$ of motifs. To that extent, we define MOTIF($\mathcal{F}$) as the set of all MOTIF instances that use a relation encoder built over $\mathcal{F}$, that is, the set of all MOTIF instances of the form $(\mathsf{Enc}_{\text{KG}}, (\mathcal{F}, \mathsf{Enc}_{\text{LIFT}}))$.

**Definition 6.1.** Let $\mathcal{F}$ and $\mathcal{F}'$ be sets of graph motifs. Then $\mathcal{F}$ *refines* $\mathcal{F}'$, denoted $\mathcal{F} \preceq \mathcal{F}'$, if for every KG $G$ and potential links $q(u, v), q'(u', v')$ over $G$, whenever every MOTIF instance in MOTIF($\mathcal{F}$) encodes $q(u, v)$ and $q'(u', v')$ in the same way, then every MOTIF instance in MOTIF($\mathcal{F}'$) also encodes $q(u, v)$ and $q'(u', v')$ in the same way. Furthermore, $\mathcal{F}$ and $\mathcal{F}'$ are said to have the same *separation power* if both $\mathcal{F} \preceq \mathcal{F}'$ and $\mathcal{F}' \preceq \mathcal{F}$ hold. We denote this as $\mathcal{F} \sim \mathcal{F}'$. □

Definition 6.1 captures the idea of the potential to separate, or distinguish, pairs of links in a KG using different sets of graph motifs. Specifically, whenever $\mathcal{F} \preceq \mathcal{F}'$, we know that using $\mathcal{F}'$ will not result in architectures that can separate links that cannot be separated by any MOTIF instance in MOTIF($\mathcal{F}$). On the other hand, if $\mathcal{F} \sim \mathcal{F}'$, then we know that $\mathcal{F}$ and $\mathcal{F}'$ can be used interchangeably without altering the potential for separating links in our architectures.

We start by observing a simple fact: isomorphic graph motifs do not make a difference in terms of separation power.

**Proposition 6.2.** *Let $P = (G_M, \bar{r})$ and $P' = (G'_M, \bar{r}')$ be two graph motifs such that $G_M$ is isomorphic to $G'_M$. Then,*

*for any set $\mathcal{F}$ of graph motifs:*

$$(\mathcal{F} \cup \{P\}) \sim (\mathcal{F} \cup \{P'\}) \sim (\mathcal{F} \cup \{P, P'\}).$$

Returning to Figure 4, it is easy to see that the motifs *h2t* and *t2h* are isomorphic. Hence, $\mathcal{F} = \{h2t, t2h, h2h, t2t\}$ has the same separation power as $\{h2t, h2h, t2t\}$ or $\{t2h, h2h, t2t\}$. But although these sets of motifs have the same separation power, this does not imply that the ULTRA architecture can drop either *h2t* or *t2h*. Indeed, while the set of links distinguished by any instance in MOTIF($\mathcal{F}$) is the same as those in MOTIF($\{h2t, h2h, t2t\}$) or MOTIF($\{t2h, h2h, t2t\}$), the ULTRA framework is a strict subset of MOTIF($\mathcal{F}$), so we cannot directly transfer these results to ULTRA framework. From a practical standpoint, the difference lies in the support for inverse relations: ULTRA bases its relation encoding on NBFNets (Zhu et al., 2021), while MOTIF uses HCNets (Huang et al., 2024), which supports inverse relations by design (see Appendices A and F).

## 6.1. When a new motif leads to more links separated?

As a tool to assert when adding a new motif $P$ increases the number of links that can be separated using a set of graph motifs $\mathcal{F}$, we provide a necessary condition for the refinement of sets of motifs $\mathcal{F}$ and $\mathcal{F}'$. We need some terminology. A homomorphism $h = (\pi, \phi)$ from a KG $G = (V, E, R)$ to a KG $G' = (V', E', R')$ is *relation-preserving* if $R \subseteq R'$ and $\phi(R) = R$. Further, a graph $H$ is a *relation-preserving core* if every relation-preserving homomorphism $h$ from $H$ to $H$ is *onto*, that is, $h(H) = H$. It is possible to show the following:

**Proposition 6.3.** *For every KG $G$, up to isomorphism, there is a unique KG $H$ such that:*

- *$H$ is a relation-preserving core, and*

- *there are relation-preserving homomorphisms from $G$ to $H$ and from $H$ to $G$.*

The KG $H$ is called the *relation-preserving core of $G$*.

Our condition is laid in terms of a specific notion of homomorphism that we call *core-onto* homomorphisms. Formally, a homomorphism $h = (\pi, \phi)$ from $G$ to $G'$ is *core-onto* if the KG $h(G)$ is isomorphic to the relation-preserving core of $G'$. If such a homomorphism exists, we write $G \to^{co} G'$. Let us also say that a motif is *trivial* if its relation-preserving core is isomorphic to the graph with a single fact $r(u, v)$. We are now ready to state the main result of this section.

**Theorem 6.4.** *Let $\mathcal{F}, \mathcal{F}'$ be two sets of motifs, and assume that $\mathcal{F} \preceq \mathcal{F}'$. Then, for every non-trivial motif $P' \in \mathcal{F}'$ there is a motif $P \in \mathcal{F}$ such that $P \to^{co} P'$.*

To prove this result, we define a coloring scheme analogous to the WL-test for relational hypergraphs presented in

---

[1] See Appendix B for a detailed comparison of the expressive powers of TRIX and MOTIF.

Huang et al. (2024), which simulates the relation encoding defined by $\text{Enc}_{\text{LIFT}}$. We then combine this scheme with techniques from Huang et al. (2023) to produce a coloring for each link in the KG, effectively simulating the encoding provided by $\text{Enc}_{\text{KG}}$ (See Appendix C). We show that this scheme is as powerful as MOTIF in its ability to distinguish links over KGs. With this tool in hand, we prove the converse of Theorem 6.4: we show how to construct a KG $G$ and links $q(u, v)$, $q'(u', v')$ such that every MOTIF instance in $\text{MOTIF}(\mathcal{F})$ encodes them in the same way, but there is a MOTIF instance in $\text{MOTIF}(\mathcal{F}')$ that encodes them differently, whenever $\mathcal{F}'$ contains a motif that cannot be covered with a core-onto homomorphism from any motif in $\mathcal{F}$.

### 6.2. Consequences for the design of MOTIF models

Consider MOTIF models in $\text{MOTIF}(\mathcal{F})$. Theorem 6.4 implies that if $P \notin \mathcal{F}$ cannot be covered via a core-onto homomorphism from any motif already in $\mathcal{F}$, then incorporating $P$ into the set of graph motifs used by the architecture should lead to more expressive encodings. This has consequences for the design of MOTIF architectures.

**$k$-path motif.** Take a motif representing a path of length $k$:

$$r_1(u_1, u_2), r_2(u_2, u_3), \ldots, r_k(u_k, u_{k+1}).$$

For $n \geq 1$, let $\mathcal{F}_n^{\text{path}}$ denote the set of motifs formed from changing the orientation of any number of edges $r_j$ in the path of length $k$, with $k \leq n$, up to isomorphism. Theorem 6.4 tells us that $\mathcal{F}_n^{\text{path}} \not\preceq \mathcal{F}_m^{\text{path}}$ whenever $n < m$, since there is no core-onto homomorphism from any motif with $k \leq n$ edges to a motif with $m$ edges. Hence, every bigger path motif adds power for more expressive encodings. Further, the set of motifs used in ULTRA (Figure 4) is $\mathcal{F} = \{h2t, t2h, h2h, t2t\}$, which, by Proposition 6.2, has the same separation power as $\mathcal{F}_2^{\text{path}} = \{h2t, h2h, t2t\}$. We can leverage this observation to prove:

**Theorem 6.5.** *ULTRA has the same expressive power as* $\text{MOTIF}(\mathcal{F}_2^{path})$.

**$k$-star motif.** As another example, consider the $k$-star motif :

$$r_1(v_1, u), r_2(v_2, u), \ldots, r_k(v_k, u).$$

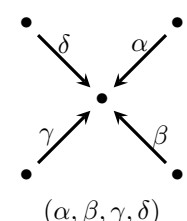

Figure 5 shows an example of 4-star motif. Once again, the set $\mathcal{F}_n^{\text{star}}$ of motifs containing all $k$-stars with $k \leq n$ gives us a hierarchy in terms of separation power, where we have that $\mathcal{F}_n^{\text{star}} \not\preceq \mathcal{F}_m^{\text{star}}$ whenever $n < m$.

*Figure 5.* The 4-star motif.

Hence, every wider star motif adds power for more expressive encodings.

**A more expressive KGFM.** We also present new ways to improve the ULTRA architecture. Adding any motif with

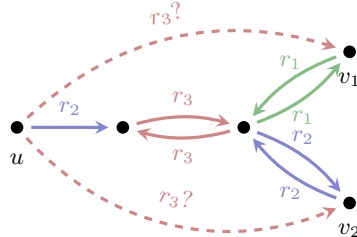

*Figure 6.* The 3-path motifs ($\mathcal{F}_3^{\text{path}}$). *tfh* stands for *tail-forward-head*, and the rest follows.

*Figure 7.* ULTRA cannot distinguish between $r_3(u, v_1)$ and $r_3(u, v_2)$ whereas $\text{MOTIF}(\mathcal{F}_3^{\text{path}})$ can.

3 edges (shown in Figure 6) to the set of four motifs used in ULTRA enhances its expressive power. As an example task shown in Figure 7, we show that ULTRA with its motifs generates a complete relation graph and fails to distinguish between $r_1$ and $r_2$. In contrast, $\text{MOTIF}(\mathcal{F}_3^{\text{path}})$ constructs a hyperedge $(r_2, r_3, r_1)$ but not $(r_1, r_3, r_2)$, thereby differentiating between $r_1$ and $r_2$ and solving the task. (See Appendix K for detailed proof.)

Furthermore, the core idea of augmenting existing KGFMs via more expressive motifs to provably enhance their expressivity fits seamlessly in the large bodies of KGFMs in the literature, such as INGRAM (Lee et al., 2023), RMPI (Geng et al., 2023), and TRIX (Zhang et al., 2024), providing an orthogonal avenue for boosting their expressive power.

### 6.3. Comparison with existing link prediction models

Existing inductive link prediction models, such as C-MPNNs and R-MPNNs (Huang et al., 2023), can be viewed as special instances of MOTIF with an empty motif set $\emptyset$ and $\text{INIT}_1$ being a one-hot encoding of relations. However, this configuration breaks relation invariance on relational hypergraphs. Somewhat counter-intuitively, we note that an instance of MOTIF that preserves relation invariance is inherently *less expressive* than its corresponding entity encoder $\text{Enc}_{\text{KG}}$, which acts as a node invariant (but not a relation invariant). This implies, for example, that NBFNet (Zhu et al., 2021) is strictly more expressive than

ULTRA when measured only over graphs with the *same relations types*. This increased expressive power comes at the cost of limited generalization to unseen relations.

# 7. Experimental analysis

We evaluate MOTIF over a broad range of knowledge graphs to answer the following questions:

**Q1**. Does MOTIF equipped with more expressive graph motifs exhibit a stronger performance?

**Q2**. Does ULTRA have the same expressive power as MOTIF($\mathcal{F}_2^{\text{path}}$)?

**Q3**. How does a more refined relation invariant help link prediction tasks?

**Q4**. What is the trade-off between expressiveness and scalability for MOTIF? (See Appendices H and I.)

In the experiments, we consider a basic architecture which replaces the relation encoder $\text{Enc}_{\text{LIFT}}$ by HCNet (Huang et al., 2024) and the entity encoder $\text{Enc}_{\text{KG}}$ by a slight modification of NBFNet (Zhu et al., 2021) (see Appendix E).

## 7.1. Synthetic experiments: ConnectHub

We construct synthetic datasets ConnectHub($k$) to validate MOTIF's enhanced expressiveness with richer motifs (**Q1**).

**Task & Setup.** The dataset ConnectHub($k$) consists of multiple synthetic KGs, where in each KG the relations are partitioned into a *positive* class $P$, a *negative* class $N$, both of size $k + 1$, and a query relation $q$. Each KG contains a $(k+1)$-star *hub* with positive relations, and for each possible subset of relations of $P$ and $N$ of size $k$, the KG contains a positive and negative $k$-star *community*, respectively. An example of such KG is shown in Figure 8 with $k = 2$. We consider the following classification task: *predict whether there exists a link of relation $q$ from the hub center to positive communities but NOT to negative communities.* The key to solving this task is successfully differentiating between relations from positive and negative classes. We consider ULTRA and MOTIF($\mathcal{F}_m^{\text{star}}$) for varying $m$, and show the empirical results on these datasets in Table 1. (Further details in Appendix L.)

**ULTRA & MOTIF with inexpressive motifs.** We claim both ULTRA and MOTIF($\mathcal{F}_m^{\text{star}}$) with $m \leq k$ cannot solve the task on ConnectHub($k$). Since the only difference between $P$ and $N$ is the presence of the $(k + 1)$-star hub of relations in $P$, these models cannot detect this higher-order motif. Thus, $\text{LIFT}_\mathcal{F}(G)$ will contain two disjoint isomorphic hypergraphs: one with positive relations and the other with negative relations. This results in indistinguishability

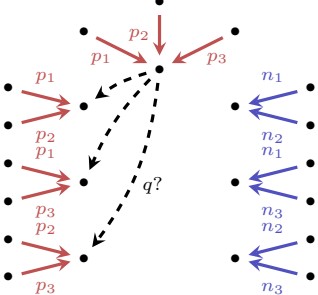

*Figure 8.* A example of a KG in one of the synthetic datasets ConnectHub(2). Relations in $P$ are in red and $N$ are in blue.

*Table 1.* Accuracy for ConnectHub($k$) datasets with ULTRA and MOTIF with different $m$-star MOTIF($\mathcal{F}_m^{\text{star}}$) for varying $k$.

| $k = ?$ | | **2** | **3** | **4** | **5** | **6** |
|---|---|---|---|---|---|---|
| ULTRA | | 0.50 | 0.50 | 0.50 | 0.50 | 0.50 |
| MOTIF | $\mathcal{F}_2^{\text{star}}$ | 0.50 | 0.50 | 0.50 | 0.50 | 0.50 |
| | $\mathcal{F}_3^{\text{star}}$ | **1.00** | 0.50 | 0.50 | 0.50 | 0.50 |
| | $\mathcal{F}_4^{\text{star}}$ | – | **1.00** | 0.50 | 0.50 | 0.50 |
| | $\mathcal{F}_5^{\text{star}}$ | – | – | **1.00** | 0.50 | 0.50 |
| | $\mathcal{F}_6^{\text{star}}$ | – | – | – | **1.00** | 0.50 |
| | $\mathcal{F}_7^{\text{star}}$ | – | – | – | – | **1.00** |

between relations in $P$ and $N$, failing the task. Empirically, we find that MOTIF($\mathcal{F}_m^{\text{star}}$) with $m \leq k$ and ULTRA only reach 50% accuracy, no better than a random guess, aligning with our theoretical studies.

**MOTIF with expressive motifs.** We claim that MOTIF($\mathcal{F}_m^{\text{star}}$) with $m > k$ can detect the presence of the positive hub, precisely due to the additional $(k + 1)$-star motifs. This allows the model to construct a hyperedge in $\text{LIFT}_\mathcal{F}(G)$ across all relations in $P$, while no such hyperedge exists for $N$, thus distinguishing $P$ from $N$. Empirically, MOTIF($\mathcal{F}_m^{\text{star}}$) with $m = k+1$ reaches 100% accuracy on ConnectHub($k$), validating our theoretical results.

## 7.2. Pretraining and fine-tuning experiments

**Datasets & Evaluations.** For pretraining and fine-tuning experiments, we follow the protocol of Galkin et al. (2024) and pretrain on FB15k237 (Toutanova & Chen, 2015), WN18RR (Dettmers et al., 2018), and CoDEx Medium (Safavi & Koutra, 2020). We then apply zero-shot inference and fine-tuned inference over 51 KGs across three settings: inductive on nodes and relations (**Inductive** $e, r$), inductive on nodes (**Inductive** $e$), and **Transductive**. The detailed information of datasets, model architectures, implementations, and hyper-parameters used in the experiments are presented in Appendix M. Following convention (Zhu et al., 2021), on each knowledge graph and for each triplet $r(u, v)$, we augment the correspond-

ing inverse triplet $r^{-1}(v, u)$ where $r^{-1}$ is a fresh relation symbol. For evaluation, we follow *filtered ranking protocol* (Bordes et al., 2013) and report Mean Reciprocal Rank (MRR), and Hits@10 for each dataset. We report both head and tails prediction results for each triplet on all datasets except three from Lv et al. (2020), where only tails prediction results are reported. The code is available at https://github.com/HxyScotthuang/MOTIF/.

**Setup.** To evaluate the impact of graph motifs, we construct four variants of MOTIF models with different $\mathcal{F}$: 3-path ($\mathcal{F}_3^{\text{path}}$), 2-path ($\mathcal{F}_2^{\text{path}}$), $\{h2t\}$ only, and no motifs ($\emptyset$), defined in Section 6. We present the average zero-shot and fine-tuned results of MOTIF over 51 KGs in Table 2, along with the corresponding results for ULTRA taken from Galkin et al. (2024). The detailed per-dataset results of pretrained, zero-shot, and finetuned inference are shown in Tables 7 to 10, respectively. Note that in the zero-shot inference setting, **Inductive** $e, r$, **Inductive** $e$, and **Transductive** are in principle indifferent, as all models encounter unseen relations in the inference graph.

**Results.** First of all, note that MOTIF($\mathcal{F}_3^{\text{path}}$) outperforms MOTIF($\mathcal{F}_2^{\text{path}}$) over all 51 datasets on average in zero-shot inference. In the case of fine-tuned inference, a similar trend is present: MOTIF($\mathcal{F}_3^{\text{path}}$) consistently outperforms MOTIF($\mathcal{F}_2^{\text{path}}$). It is also worth noting that both MOTIF($\mathcal{F}_3^{\text{path}}$) and MOTIF($\mathcal{F}_2^{\text{path}}$) achieve further performance improvements after fine-tuning on the training set compared to zero-shot inference. These findings demonstrate that the additionally introduced motifs help the model to learn richer relation invariants (**Q1**), which are then leveraged for better predictions on new KGs with unseen nodes and relations. This observation is true on both zero-shot and fine-tuned inference.

Secondly, recall that MOTIF($\mathcal{F}_2^{\text{path}}$) and ULTRA have the same expressive power as shown in Theorem 6.5. This theoretical finding is supported in our empirical study, as we see that ULTRA performs similarly with MOTIF($\mathcal{F}_2^{\text{path}}$) in these experiments (**Q2**).

Lastly, when it comes to comparing MOTIF($\mathcal{F}_2^{\text{path}}$) and MOTIF($\{h2t\}$), we observe a gradual decrease in all metrics on all datasets as the motifs are removed. This aligns with Theorem 6.4: adding 3-paths on top of 2-paths yields more expressive power on MOTIF since there is no core-onto homomorphism from any 2-path to 3-path motifs. Similarly, adding $h2h$ or $t2t$ on top of $h2t$ also yields more expressive power as there is no homomorphism from $h2t$ to $h2h$ or $t2t$. In the extreme case, we observe that removing all of the motifs in MOTIF($\emptyset$) prevents the model from learning any non-trivial relation invariants, thus exhibiting a complete failure in generalization to unseen KGs.

**Results on datasets with complex relational patterns**

**(Q1).** Our findings on domain-specific benchmarks further validate the benefits of increased expressiveness. On Metafam (Zhou et al., 2023a), a dataset designed to challenge models with conflicting and compositional relational patterns, MOTIF achieves a 45% improvement in MRR over ULTRA in the zero-shot setting. This underscores the model's ability to capture non-trivial relational invariants when higher-order motifs are available. Similarly, on WIKITOPICS-MT (Zhou et al., 2023a), where multiple KGs from diverse topics are combined to form a multi-task setting, MOTIF($\mathcal{F}_3^{\text{path}}$) consistently outperforms baselines. Averaged across 8 datasets, MOTIF($\mathcal{F}_3^{\text{path}}$) improves MRR from 0.331 to 0.358 and Hits@10 from 0.442 to 0.481; gains are even more pronounced on certain subsets (e.g., a 58% relative improvement on MT1-tax). These results confirm that expressiveness improvements translate into practical gains when the structure of the data aligns with the capabilities of the underlying motifs.

### 7.3. End-to-end experiments

We also conduct end-to-end experiments where for each dataset, we train a MOTIF($\mathcal{F}_3^{\text{path}}$) on the training set from scratch and evaluate its corresponding validation and test sets. We present the average results over all 54 KGs in Table 3 (Full results in Table 11 and Table 12). We continue to see improved performance of MOTIF($\mathcal{F}_3^{\text{path}}$) over ULTRA due to the additional motifs, allowing models to capture a richer set of information over relations (**Q1**).

**Degradation of relation learning (Q3).** As a case study, we focus on the end-to-end experiments with WN-v2 (Teru et al., 2020), an inductive (on node) dataset with only 20 relations after inverse augmentation. The MRR for ULTRA is 0.296, severely lower than that of MOTIF($\mathcal{F}_3^{\text{path}}$) (0.684). To investigate, we plot the cosine similarity[2] among the computed relation embeddings when queried relations are $r_0 = \text{derivationally\_related\_form}$ and $r_2 = \text{hypernym}$ in Figure 9. ULTRA produces highly similar relation representations, whereas MOTIF with richer motifs generates distinguishing relation representations, aiding the entity encoder in link prediction.

Nevertheless, we notice that such degradation does not appear in the pretraining experiments, suggesting that either enriching relation learning with diverse motifs or training over multiple KGs could help avoid such degradation.

## 8. Conclusions

We introduced MOTIF, a general framework that extends existing KGFMs via incorporating arbitrary motifs. In the context of this framework, we studied the expressive power

---

[2]The block structure is due to the concatenation of augmented inverse relations.

Table 2. Average zero-shot and fine-tuned link prediction MRR and Hits@10 over 51 KGs.

| Model | Motif ($\mathcal{F}$) | Inductive $e, r$ (23 graphs) | | Inductive $e$ (18 graphs) | | Transductive (10 graphs) | | Total Avg (51 graphs) | | Pretrained (3 graphs) | |
|---|---|---|---|---|---|---|---|---|---|---|---|
| | | MRR | H@10 | MRR | H@10 | MRR | H@10 | MRR | H@10 | MRR | H@10 |
| **Zero-shot Inference** | | | | | | | | | | | |
| ULTRA | | 0.345 | 0.513 | 0.431 | 0.566 | 0.339 | 0.494 | 0.374 | 0.529 | - | - |
| | $\mathcal{F}_3^{\text{path}}$ | **0.349** | **0.525** | **0.436** | **0.577** | **0.343** | **0.496** | **0.378** | **0.537** | - | - |
| MOTIF | $\mathcal{F}_2^{\text{path}}$ | 0.337 | 0.509 | 0.431 | 0.570 | 0.335 | 0.492 | 0.370 | 0.527 | - | - |
| | $\{h2t\}$ | 0.330 | 0.503 | 0.422 | 0.559 | 0.325 | 0.482 | 0.361 | 0.518 | - | - |
| | $\emptyset$ | 0.074 | 0.121 | 0.107 | 0.169 | 0.054 | 0.101 | 0.082 | 0.134 | - | - |
| **Finetuned Inference** | | | | | | | | | | | |
| ULTRA | | 0.397 | 0.556 | 0.442 | 0.582 | 0.384 | 0.543 | 0.410 | 0.563 | 0.407 | **0.568** |
| MOTIF | $\mathcal{F}_3^{\text{path}}$ | **0.401** | **0.558** | **0.455** | **0.594** | **0.394** | **0.549** | **0.419** | **0.569** | **0.415** | 0.565 |

Table 3. Average end-to-end MRR and Hits@10 results over 54 KGs.

| Model | Total Avg (54 graphs) | |
|---|---|---|
| | MRR | H@10 |
| ULTRA | 0.394 | 0.552 |
| MOTIF($\mathcal{F}_3^{\text{path}}$) | **0.409** | **0.561** |

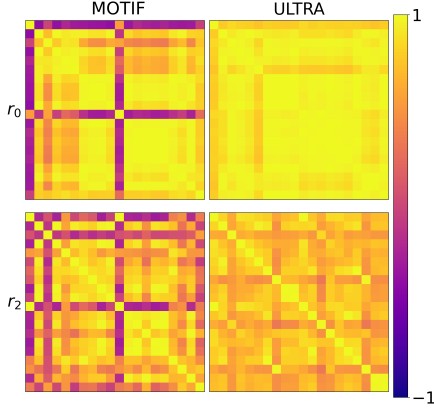

Figure 9. Cosine similarities between generated relations embeddings when queried relation $q$ is $r_0$ and $r_2$ in test set of WN-v2.

of KGFMs and identified *sufficient* conditions under which adding new motifs improves MOTIFs' ability to compute finer relation invariants. We thoroughly discussed the implications of our results on existing KGFMs. The theoretical findings are empirically validated on a diverse set of datasets.

Our work provides key insights and concrete recipes for designing more expressive KGFMs. However, the typical trade-off between expressive power and computational complexity clearly also applies to our study. While using richer motifs provably increases MOTIF's expressive power, this may come at the cost of computational efficiency in terms

of both time and memory (please see the discussion in Appendices H and I). A promising avenue for future work is to enhance MOTIF's computational efficiency, thus enabling scalability to real-world large-scale graphs.

## Acknowledgment

Bronstein is supported by EPSRC Turing AI World-Leading Research Fellowship No. EP/X040062/1 and EPSRC AI Hub on Mathematical Foundations of Intelligence: An "Erlangen Programme" for AI No. EP/Y028872/1. Reutter is funded by Fondecyt grant 1221799. Barceló and Reutter are funded by ANID–Millennium Science Initiative Program - CodeICN17002. Barceló and Romero are funded by the National Center for Artificial Intelligence CENIA FB210017, BasalANID.

## Impact Statement

This paper presents work aimed at advancing the field of Machine Learning. Our methods have potential applications in recommendation systems and knowledge graph completion, among others. While these applications can have significant societal impacts, including improving information retrieval and personalization, as well as potential risks such as bias amplification and misinformation propagation, we do not identify any specific, immediate concerns requiring special attention in this work.

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

# A. C-MPNNs and HC-MPNNs

In this section, we follow Huang et al. (2023) and Huang et al. (2024) to define *conditional message passing neural networks* (C-MPNNs) and *hypergraph conditional message passing neural networks* (HC-MPNNs). We then follow Huang et al. (2024) to define *hypergraph conditional networks* (HCNets) as an architecture from HC-MPNNs. We also present their corresponding relational Weisfeiler Leman test to study their expressive power rigorously. For ease of presentation, we omit the discussion regarding history functions and readout functions from Huang et al. (2023). Also, some notations are adapted to simplify the exposition.

## A.1. Model definitions

**C-MPNNs.** Let $G = (V, E, R)$ be a KG. A *conditional message passing neural network* (C-MPNN) iteratively computes node representations, relative to a fixed query $q \in R$ and a fixed source node $u \in V$, as follows:

$$
\boldsymbol{h}_{v|u,q}^{(0)} = \text{INIT}(u, v, q)
$$
$$
\boldsymbol{h}_{v|u,q}^{(\ell+1)} = \text{UP}\Big(\boldsymbol{h}_{v|u,q}^{(\ell)}, \text{AGG}(\{\!\!\{\text{MSG}_r(\boldsymbol{h}_{w|u,q}^{(\ell)}, \boldsymbol{z}_q)| \, w \in \mathcal{N}_r(v), r \in R\}\!\!\})\Big),
$$

where INIT, UP, AGG, and MSG$_r$ are differentiable *initialization*, *update*, *aggregation*, and relation-specific *message* functions, respectively.

We denote by $\boldsymbol{h}_q^{(\ell)} : V \times V \to \mathbb{R}^{d(\ell)}$ the function $\boldsymbol{h}_q^{(\ell)}(u, v) := \boldsymbol{h}_{v|u,q}^{(\ell)}$, and denote $\boldsymbol{z}_q$ to be a learnable vector representing the query $q \in R$. We can then interpret a C-MPNN as computing representations of pair of nodes. Given a fixed number of layers $L \geq 0$, a C-MPNN computes the final pair representations as $\boldsymbol{h}_q^{(L)}$. In order to decode the likelihood of the fact $q(u, v)$ for some $q \in R$, we consider a unary decoder $\text{Dec} : \mathbb{R}^{d(L)} \to \mathbb{R}$. We additionally require INIT$(u, v, q)$ to satisfy the property of *target node distinguishability*: for all $q \in R$ and $v \neq u \in V$, it holds that $\text{INIT}(u, u, q) \neq \text{INIT}(u, v, q)$.

**HC-MPNNs.** Given a relational hypergraph $H = (V, E, R)$ and a query $\boldsymbol{q} = (q, \tilde{\boldsymbol{u}}, t) := q(u_1, \cdots, u_{t-1}, ?, u_{t+1}, \cdots, u_m)$, for $\ell \geq 0$, an HC-MPNN computes a sequence of feature maps $\boldsymbol{h}_{v|\boldsymbol{q}}^{(\ell)}$ as follows:

$$
\boldsymbol{h}_{v|\boldsymbol{q}}^{(0)} = \text{INIT}(v, \boldsymbol{q}),
$$
$$
\boldsymbol{h}_{v|\boldsymbol{q}}^{(\ell+1)} = \text{UP}\Big(\boldsymbol{h}_{v|\boldsymbol{q}}^{(\ell)}, \text{AGG}(\boldsymbol{h}_{v|\boldsymbol{q}}^{(\ell)}, \{\!\!\{\text{MSG}_{\rho(e)}(\{(\boldsymbol{h}_{w|\boldsymbol{q}}^{(\ell)}, j) \mid (w, j) \in \mathcal{N}_i(e)\}, \boldsymbol{q}), | \, (e, i) \in E(v)\}\!\!\})\Big),
$$

where INIT, UP, AGG, and MSG$_{\rho(e)}$ are differentiable *initialization*, *update*, *aggregation*, and relation-specific *message* functions, respectively. An HC-MPNN has a fixed number of layers $L \geq 0$, and the final conditional node representations are given by $\boldsymbol{h}_{v|\boldsymbol{q}}^{(L)}$. We denote by $\boldsymbol{h}_{\boldsymbol{q}}^{(\ell)} : V \to \mathbb{R}^{d(\ell)}$ the function $\boldsymbol{h}_{\boldsymbol{q}}^{(\ell)}(v) := \boldsymbol{h}_{v|\boldsymbol{q}}^{(\ell)}$.

**HCNets.** HCNets can be seen as an architecture of HC-MPNN frameworks by choosing appropriate initialization, aggregation, and message functions. Let $H = (V, E, R)$ be a relational hypergraph. For a query $\boldsymbol{q} = (q, \tilde{\boldsymbol{u}}, t) := q(u_1, \cdots, u_{t-1}, ?, u_{t+1}, \cdots, u_m)$, an HCNet computes the following representations for all $\ell \geq 0$:

$$
\boldsymbol{h}_{v|\boldsymbol{q}}^{(0)} = \sum_{i \neq t} \mathbb{1}_{v=u_i} * (\boldsymbol{p}_i + \boldsymbol{z}_q),
$$
$$
\boldsymbol{h}_{v|\boldsymbol{q}}^{(\ell+1)} = \sigma\Big(\boldsymbol{W}^{(\ell)}\Big[\boldsymbol{h}_{v|\boldsymbol{q}}^{(\ell)} \Big\| \sum_{(e,i) \in E(v)} g_{\rho(e),q}^{(\ell)}\Big(\odot_{j \neq i}\big(\alpha^{(\ell)}\boldsymbol{h}_{e(j)|\boldsymbol{q}}^{(\ell)} + (1 - \alpha^{(\ell)})\boldsymbol{p}_j\big)\Big)\Big] + \boldsymbol{b}^{(\ell)}\Big),
$$

where $g_{\rho(e),q}^{(\ell)}$ is a learnable message function which is often a diagonal matrix, $\sigma$ is an activation function, $\boldsymbol{W}^{(\ell)}$ is a learnable weight matrix, $\boldsymbol{b}^{(\ell)}$ as learnable bias term per layer, $\boldsymbol{z}_q$ is the learnable query vector for $q \in R$, and $\mathbb{1}_C$ is the indicator function that returns 1 if condition $C$ is true, and 0 otherwise. $\alpha$ is a learnable scalar and $\boldsymbol{p}_i$ to refer to the sinusoidal positional encoding (Vaswani et al., 2017) at position $i$.

## A.2. Relational WL tests

To characterize these models' expressive power, Huang et al. (2023) and Huang et al. (2024) proposed the corresponding relational Weisfeiler-Leman algorithms to capture these models exactly.

**Relational assymetric local 2-WL test** ($\mathsf{rawl}_2$). Following Huang et al. (2023), we define the relational asymmetric local 2-WL test ($\mathsf{rawl}_2$) as follows: Let $G = (V, E, R)$ be a KG and $\eta : V \times V \to D$ an initial pairwise node coloring. We say $\eta$ satisfies *target node distinguishability* if $\eta(u, u) \neq \eta(u, v)$ for all $u \neq v \in V$. Then, for each $\ell \geq 0$, we update the coloring as:

$$\mathsf{rawl}_2^{(0)}(u, v) = \eta(u, v),$$
$$\mathsf{rawl}_2^{(\ell+1)}(u, v) = \text{HASH}\big(\mathsf{rawl}_2^{(\ell)}(u, v), \{\!\!\{ (\mathsf{rawl}_2^{(\ell)}(u, w), r) \mid w \in \mathcal{N}_r(v), r \in R \}\!\!\} \big),$$

where HASH injectively maps the above pair to a fresh unique color. It is shown in Theorem 5.1, Huang et al. (2023) that $\mathsf{rawl}_2^{(\ell)}$ is a relational WL test that characterizes the expressive power of C-MPNNs, i.e., for all C-MPNN, given any knowledge graph $G$, $\ell \geq 0$, we have $\mathsf{rawl}_2^{(\ell)}$ refines the $\ell$ layer of this C-MPNN. In addition, given any knowledge graph $G$, there exists a C-MPNN has equivalent expressive power of $\mathsf{rawl}_2^{(\ell)}$ for all $\ell \geq 0$.

For any knowledge graph $G$, we will sometimes write $\mathsf{rawl}_2^+(G)$ as a shorthand of $\mathsf{rawl}_2(G^+)$, i.e., $\mathsf{rawl}_2$ applied on *augmented knowledge graph* $G^+$. Formally speaking, let $G = (V, E, R)$ be a knowledge graph. Following Huang et al. (2023), we define its *augmented knowledge graph* to be $G^+ = (V, E^+, R^+)$, where $R^+$ is the disjoint union of $R$ and $R^- := \{ r^- \mid r \in R \}$, and $E^+ = E \cup E^-$ where $E^- = \{ r^-(v, u) \mid r(u, v) \in E, u \neq v \}$.

**Hypergraph relational local 1-WL test** ($\mathsf{hrwl}_1$). Similarly, following Huang et al. (2024), we define the hypergraph relational local 1-WL test ($\mathsf{hrwl}_1$) as follows: Let $H = (V, E, R)$ be a relational hypergraph and $c : V \to D$ an initial node coloring. Then, for each $\ell \geq 0$, we update the coloring as:

$$\mathsf{hrwl}_1^{(0)}(v) = c(v),$$
$$\mathsf{hrwl}_1^{(\ell+1)}(v) = \text{HASH}\Big(\mathsf{hrwl}_1^{(\ell)}(v), \{\!\!\{ \big( \{ (\mathsf{hrwl}_1^{(\ell)}(w), j) \mid (w, j) \in \mathcal{N}_i(e) \}, \rho(e) \big) \mid (e, i) \in E(v) \}\!\!\} \Big).$$

where the function HASH is an injective mapping that maps the above pair to a unique color that has not been used in the previous iteration. When we restrict $\mathsf{hrwl}_1$ to initial colorings $c$ that respect a given query $\boldsymbol{q} := (q, \tilde{\boldsymbol{u}}, t) = q(u_1, \cdots, u_{t-1}, u_{t+1}, \cdots, u_k)$, we say that the coloring $c$ satisfies *generalized target node distinguishability with respect to* $\boldsymbol{q}$ if:

$$c(u) \neq c(v) \quad \forall u \in \tilde{\boldsymbol{u}}, v \notin \tilde{\boldsymbol{u}} \qquad \text{and} \qquad c(u_i) \neq c(u_j) \quad \forall u_i, u_j \in \tilde{\boldsymbol{u}}, u_i \neq u_j.$$

Then, Theorem 5.1, Huang et al. (2024) shows that $\mathsf{hrwl}_1$ with initial coloring $c$ satisfying *generalized target node distinguishability* characterizes the expressive power of HC-MPNNs.

**Hypergraph relational local 2-WL test** ($\mathsf{hcwl}_2$). Following Huang et al. (2024), if we further restrict the test and apply only on knowledge graphs where all the edges have arity 2 and additionally only with tail prediction, we can instead assign pair-wise coloring $\eta : V \times V \mapsto D$ satisfying *target node distinguishability*, i.e. $\forall u \neq v, \eta(u, u) \neq \eta(u, v)$. Then, we define a *relational hypergraph conditioned local 2-WL test*, denoted as $\mathsf{hcwl}_2$. $\mathsf{hcwl}_2$ iteratively updates binary coloring $\eta$ as follow for all $\ell \geq 0$:

$$\mathsf{hcwl}_2^{(0)} = \eta(u, v)$$
$$\mathsf{hcwl}_2^{(\ell+1)}(u, v) = \text{HASH}\Big(\mathsf{hcwl}_2^{(\ell)}(u, v), \{\!\!\{ \big( \{ (\mathsf{hcwl}_2^{(\ell)}(u, w), j) \mid (w, j) \in \mathcal{N}_i(e) \}, \rho(e) \big) \mid (e, i) \in E(v) \}\!\!\} \Big)$$

## B. Differences between TRIX and MOTIF

TRIX (Zhang et al., 2024) is one of the first works to study its expressivity of KGFMs by comparing it to ULTRA (Galkin et al., 2024). However, our proposed framework MOTIF is inherently incomparable from TRIX in terms of theoretical expressivity.

TRIX enhances expressivity by introducing a model capable of implicitly counting the number of homomorphism matches in a knowledge graph. This capability allows TRIX to distinguish between relation pairs based on their frequency of occurrence within specific structural patterns. For instance, consider a knowledge graph with edges

$$E = \{ r_1(u_1, v_1), r_2(v_1, w_1), r_3(u_2, v_2), r_4(v_2, w_2), r_3(u_3, v_3), r_4(v_3, w_3) \}.$$

Here, the pair $(r_3, r_4)$ appears twice in 2-hop paths, while $(r_1, r_2)$ appears only once. TRIX can distinguish between these pairs due to its ability to count such occurrences.

In contrast, MOTIF focuses on the existence of specific higher-order motifs within the knowledge graph, rather than their frequency. This design enables MOTIF to capture complex relational structures that may not be distinguishable through frequency counts alone. For example, consider a knowledge graph with edges

$$E = \{r_1(x_1, x_2), r_2(x_1, x_2), r_1(x_3, x_4), r_2(x_3, x_4), r_3(y_1, y_2), r_4(y_1, y_4), r_3(y_3, y_2), r_4(y_3, y_4)\}.$$

In this case, the pairs $(r_1, r_2)$ and $(r_3, r_4)$ may be indistinguishable under TRIX due to isomorphic relation graphs under head-to-head and tail-to-tail mappings. However, MOTIF can distinguish between these pairs by identifying specific motifs, such as the PARA motif with edges $\{\alpha(x, y), \beta(x, y)\}$, which maps only to $(r_1, r_2)$.

Therefore, while TRIX excels at distinguishing relation pairs based on frequency counts of specific patterns, MOTIF provides a complementary approach by focusing on the structural existence of complex motifs. This fundamental difference renders the two frameworks incomparable in terms of expressivity, as each can capture relational distinctions that the other cannot. However, note that incorporating higher-order motifs into TRIX could similarly boost its expressivity, highlighting our framework as a complementary way for improving KGFMs.

## C. A WL test for MOTIF

We start by introducing a two-stage WL test that matches the separating power of $\text{MOTIF}(\mathcal{F})$, for a set of motifs $\mathcal{F}$. Using this equivalence, we show Proposition 6.2 and Theorem 6.4, in sections D.1 and D.3, respectively. Characterizing the separating power of $\text{MOTIF}(\mathcal{F})$ via a suitable test, allows us to reason about simpler coloring schemes, which, in turn, makes the proofs more manageable. Also, the equivalence between $\text{MOTIF}(\mathcal{F})$ and our two-stage WL test may be of independent interest.

We assume a given KG $G = (V, E, R)$. Our test assigns one color to each link $q(u, v)$, for $q \in R$ and $u, v \in V$ (recall that links are not necessarily facts in $G$). In the first stage, we color the nodes of $\text{LIFT}_{\mathcal{F}}(G) = (V_{\text{LIFT}}, E_{\text{LIFT}}, R_{\text{LIFT}})$, that is, the set $V_{\text{LIFT}} = R$. When coloring a relation $r \in R$, we always condition on another relation $q \in R$. We denote by $\text{hcwl}_{\mathcal{F}}^{(t)}(q, r)$ the color assigned to $r$, conditioned on $q$, on iteration $t \geq 0$. We assume a predefined number of iterations $T \geq 0$, which will be a parameter of our test. In the second stage, we color the nodes $v$ of the KG $G$, conditioned on a relation $q \in R$ and a source node $u \in V$. This determines the final color $\text{col}_{\mathcal{F},T}^{(\ell)}(q(u, v))$ for the link $q(u, v)$. Crucially, the second stage relies on the first stage in that it uses the colors $\text{hcwl}_{\mathcal{F}}^{(T)}(q, r)$ internally. Roughly speaking, the first-stage coloring follows the coloring strategy of $\text{hrwl}_1$ applied to $\text{LIFT}_{\mathcal{F}}(G)$, while the second-stage coloring follows the strategy of $\text{rawl}_2$, after incorporating the relation encodings obtained in the first stage. (see Section A for the definition of $\text{hrwl}_1$ and $\text{rawl}_2$).

Formally, the first-stage colors $\text{hcwl}_{\mathcal{F}}^{(t)}$ are defined via the following rules:

$$\text{hcwl}_{\mathcal{F}}^{(0)}(q, r) = \mathbb{1}_{r=q} * 1$$
$$\text{hcwl}_{\mathcal{F}}^{(t+1)}(q, r) = \text{HASH}\left(\text{hcwl}_{\mathcal{F}}^{(t)}(q, r), \{\!\{(\{(\text{hcwl}_{\mathcal{F}}^{(t)}(q, s), j) \mid (s, j) \in \mathcal{N}^i(e)\}, \rho(e)) \mid (e, i) \in E_{\text{LIFT}}(r)\}\!\}\right).$$

Recall that
$$E_{\text{LIFT}}(r) = \{(e, i) \mid e(i) = r, e \in E_{\text{LIFT}}, 1 \leq i \leq \text{ar}(\rho(e))\}.$$

is the set of edge-position pairs of a node $r$, and $\mathcal{N}^i(e)$ is the positional neighborhood of a hyperedge $e$ with respect to a position $i$:
$$\mathcal{N}^i(e) = \{(e(j), j) \mid j \neq i, 1 \leq j \leq \text{ar}(\rho(e))\}.$$

$\mathbb{1}_{r=q}$ takes value 1 if $r = q$, and 0 otherwise. The function HASH is a fixed injective function taking values in the natural numbers.

The second-stage colors $\text{col}_{\mathcal{F},T}^{(\ell)}$, where $T \geq 0$ is a parameter, are defined via the following rules:

$$\text{col}_{\mathcal{F},T}^{(0)}(q(u, v)) = \mathbb{1}_{v=u} * \text{hcwl}_{\mathcal{F}}^{(T)}(q, q)$$
$$\text{col}_{\mathcal{F},T}^{(\ell+1)}(q(u, v)) = \text{HASH}\left(\text{col}_{\mathcal{F},T}^{(\ell)}(q(u, v)), \{\!\{(\text{col}_{\mathcal{F},T}^{(\ell)}(q(u, w)), \text{hcwl}_{\mathcal{F}}^{(T)}(q, r)) \mid w \in \mathcal{N}_r(v), r \in R\}\!\}\right).$$

We start by showing that our coloring scheme upper bounds the separation power of any model in $\text{MOTIF}(\mathcal{F})$. For simplicity, we assume that any model in $\text{MOTIF}(\mathcal{F})$ has the form $(\text{Enc}_{\text{KG}}, (\mathcal{F}, \text{Enc}_{\text{LIFT}}))$, where

1. the initialization function $\text{INIT}_1$ used by the relational encoder $\text{Enc}_{\text{LIFT}}$ is defined as $\text{INIT}_1(q, r) = \mathbb{1}_{r=q} * \mathbf{1}^d$, and

2. the initialization function $\text{INIT}_2$ used by the entity encoder is defined as $\text{INIT}_2(u, v, q) = \mathbb{1}_{v=u} * \mathbf{h}_{q|q}^{(T)}$.

This is a common choice, and in particular, it is the choice we take in our basic architecture for the experiments (see Section E).

**Proposition C.1.** *Let $G = (V, E, R)$ be a KG and $(\text{Enc}_{\text{KG}}, (\mathcal{F}, \text{Enc}_{\text{LIFT}}))$ be a $\text{MOTIF}(\mathcal{F})$ instance, where $\text{Enc}_{\text{LIFT}}$ and $\text{Enc}_{\text{KG}}$ involve $T \geq 0$ and $L \geq 0$ layers, respectively. Then for every pair of links $q(u, v)$ and $q'(u', v')$, we have that*

$$\text{col}_{\mathcal{F},T}^{(L)}(q(u, v)) = \text{col}_{\mathcal{F},T}^{(L)}(q'(u', v')) \implies \text{Enc}_{\text{KG},u,q}[v] = \text{Enc}_{\text{KG},u',q'}[v'].$$

*Proof.* We show first the following claim:

($\dagger$) for every quadruple of relations $q, r, q', r' \in R$, we have that

$$\text{hcwl}_{\mathcal{F}}^{(T)}(q, r) = \text{hcwl}_{\mathcal{F}}^{(T)}(q', r') \implies \text{Enc}_{\text{LIFT},q}[r] = \text{Enc}_{\text{LIFT},q'}[r'].$$

Recall that $\text{Enc}_{\text{LIFT},p}[s]$ is determined by the sequence of vectors $\mathbf{h}_{s|p}^{(t)}$ and that $\text{Enc}_{\text{LIFT},p}[s] = \mathbf{h}_{s|p}^{(T)}$.

Take a quadruple $q, r, q', r' \in R$. We prove by induction that for every $0 \leq t \leq T$, we have that

$$\text{hcwl}_{\mathcal{F}}^{(t)}(q, r) = \text{hcwl}_{\mathcal{F}}^{(t)}(q', r') \implies \mathbf{h}_{r|q}^{(t)} = \mathbf{h}_{r'|q'}^{(t)}.$$

For the base case, take $t = 0$. Assume that $\text{hcwl}_{\mathcal{F}}^{(0)}(q, r) = \text{hcwl}_{\mathcal{F}}^{(0)}(q', r')$. By the definition of $\text{hcwl}_{\mathcal{F}}^{(0)}$, either $q = r$ and $q' = r'$, or $q \neq r$ and $q' \neq r'$. In either case, we obtain $\mathbf{h}_{r|q}^{(0)} = \mathbb{1}_{r=q} * \mathbf{1}^d = \mathbb{1}_{r'=q'} * \mathbf{1}^d = \mathbf{h}_{r'|q'}^{(0)}$.

For the inductive case, suppose that

$$\text{hcwl}_{\mathcal{F}}^{(t)}(q, r) = \text{hcwl}_{\mathcal{F}}^{(t)}(q', r') \implies \mathbf{h}_{r|q}^{(t)} = \mathbf{h}_{r'|q'}^{(t)}$$

and assume that $\text{hcwl}_{\mathcal{F}}^{(t+1)}(q, r) = \text{hcwl}_{\mathcal{F}}^{(t+1)}(q', r')$. By the definition of $\text{hcwl}_{\mathcal{F}}$, it follows that

$$\text{hcwl}_{\mathcal{F}}^{(t)}(q, r) = \text{hcwl}_{\mathcal{F}}^{(t)}(q', r')$$

$$\{\!\!\{ \big( \{\!\!\{ (\text{hcwl}_{\mathcal{F}}^{(t)}(q, s), j) \mid (s, j) \in \mathcal{N}^i(e) \}\!\!\}, \rho(e) \big) \mid (e, i) \in E_{\text{LIFT}}(r) \}\!\!\} =$$
$$\{\!\!\{ \big( \{\!\!\{ (\text{hcwl}_{\mathcal{F}}^{(t)}(q', s'), j) \mid (s', j) \in \mathcal{N}^i(e) \}\!\!\}, \rho(e) \big) \mid (e, i) \in E_{\text{LIFT}}(r') \}\!\!\}.$$

We can apply the inductive hypothesis to the first equation and obtain that $\mathbf{h}_{r|q}^{(t)} = \mathbf{h}_{r'|q'}^{(t)}$. The second equation together with the inductive hypothesis implies that

$$\{\!\!\{ \big( \{\!\!\{ (\mathbf{h}_{s|q}^{(t)}, j) \mid (s, j) \in \mathcal{N}^i(e) \}\!\!\}, \rho(e) \big) \mid (e, i) \in E_{\text{LIFT}}(r) \}\!\!\} =$$
$$\{\!\!\{ \big( \{\!\!\{ (\mathbf{h}_{s'|q'}^{(t)}, j) \mid (s', j) \in \mathcal{N}^i(e) \}\!\!\}, \rho(e) \big) \mid (e, i) \in E_{\text{LIFT}}(r') \}\!\!\}.$$

As a consequence, we have that

$$M = \{\!\!\{ \text{MSG}_{\rho(e)} \big( \{\!\!\{ (\mathbf{h}_{s|q}^{(t)}, j) \mid (s, j) \in \mathcal{N}^i(e) \}\!\!\} \big) \mid (e, i) \in E_{\text{LIFT}}(r) \}\!\!\} =$$
$$\{\!\!\{ \text{MSG}_{\rho(e)} \big( \{\!\!\{ (\mathbf{h}_{s'|q}^{(t)}, j) \mid (s', j) \in \mathcal{N}^i(e) \}\!\!\} \big) \mid (e, i) \in E_{\text{LIFT}}(r) \}\!\!\} = M'$$

where $\text{MSG}_{\rho(e)}$ is the motif-specific message function used in the $(t + 1)$-th layer. It follows that

$$\mathbf{h}_{r|q}^{(t+1)} = \text{UP}_1 \big( \mathbf{h}_{r|q}^{(t)}, \text{AGG}_1(M) \big) = \text{UP}_1 \big( \mathbf{h}_{r'|q'}^{(t)}, \text{AGG}_1(M') \big) = \mathbf{h}_{r'|q'}^{(t+1)}$$

as required. ($\text{UP}_1$ and $\text{AGG}_1$ are the update and aggregate functions in the $(t+1)$-th layer.)

Take a pair of links $q(u,v)$ and $q'(u',v')$. Now we are ready to show

$$\text{col}_{\mathcal{F},T}^{(L)}(q(u,v)) = \text{col}_{\mathcal{F},T}^{(L)}(q'(u',v')) \implies \text{Enc}_{\text{KG},u,q}[v] = \text{Enc}_{\text{KG},u',q'}[v'].$$

Recall that $\text{Enc}_{\text{KG},x,p}[y]$ is determined by the sequence of vectors $\boldsymbol{h}_{y|x,p}^{(\ell)}$ and that $\text{Enc}_{\text{KG},x,p}[y] = \boldsymbol{h}_{y|x,p}^{(L)}$.

We show by induction that for every $0 \le \ell \le L$, it holds that

$$\text{col}_{\mathcal{F},T}^{(\ell)}(q(u,v)) = \text{col}_{\mathcal{F},T}^{(\ell)}(q'(u',v')) \implies \boldsymbol{h}_{v|u,q}^{(\ell)} = \boldsymbol{h}_{v'|u',q'}^{(\ell)}.$$

In the base case $\ell = 0$. Suppose $\text{col}_{\mathcal{F},T}^{(0)}(q(u,v)) = \text{col}_{\mathcal{F},T}^{(0)}(q'(u',v'))$, i.e., $\mathbb{1}_{v=u} * \text{hcwl}_{\mathcal{F}}^{(T)}(q,q) = \mathbb{1}_{v'=u'} * \text{hcwl}_{\mathcal{F}}^{(T)}(q',q')$. Note that without loss of generality, we can assume that HASH never takes the value 0, and hence $\text{hcwl}_{\mathcal{F}}^{(T)}(q,q) \ne 0$ and $\text{hcwl}_{\mathcal{F}}^{(T)}(q',q') \ne 0$. We then have two cases:

1. $v \ne u$ and $v' \ne u'$. We obtain $\boldsymbol{h}_{v|u,q}^{(0)} = \boldsymbol{h}_{v'|u',q'}^{(0)}$ as required.

2. $v = u$, $v' = u'$ and $\text{hcwl}_{\mathcal{F}}^{(T)}(q,q) = \text{hcwl}_{\mathcal{F}}^{(T)}(q',q')$. By the claim (†), we have that $\boldsymbol{h}_{q|q}^{(T)} = \boldsymbol{h}_{q'|q'}^{(T)}$. We conclude that $\boldsymbol{h}_{v|u,q}^{(0)} = \boldsymbol{h}_{v'|u',q'}^{(0)}$ as desired.

For the inductive step, suppose that

$$\text{col}_{\mathcal{F},T}^{(\ell)}(q(u,v)) = \text{col}_{\mathcal{F},T}^{(\ell)}(q'(u',v')) \implies \boldsymbol{h}_{v|u,q}^{(\ell)} = \boldsymbol{h}_{v'|u',q'}^{(\ell)}.$$

Assume that $\text{col}_{\mathcal{F},T}^{(\ell+1)}(q(u,v)) = \text{col}_{\mathcal{F},T}^{(\ell+1)}(q'(u',v'))$. It follows that

$$\text{col}_{\mathcal{F},T}^{(\ell)}(q(u,v)) = \text{col}_{\mathcal{F},T}^{(\ell)}(q'(u',v'))$$

$$\{\!\{\big(\text{col}_{\mathcal{F},T}^{(\ell)}(q(u,w)), \text{hcwl}_{\mathcal{F}}^{(T)}(q,r)\big) \mid w \in \mathcal{N}_r(v), r \in R\}\!\} = \{\!\{\big(\text{col}_{\mathcal{F},T}^{(\ell)}(q'(u',w)), \text{hcwl}_{\mathcal{F}}^{(T)}(q',r)\big) \mid w \in \mathcal{N}_r(v'), r \in R\}\!\}.$$

Using the inductive hypothesis and claim (†), we obtain that

$$\boldsymbol{h}_{v|u,q}^{(\ell)} = \boldsymbol{h}_{v'|u',q'}^{(\ell)}$$

$$\{\!\{\big(\boldsymbol{h}_{w|u,q}^{(\ell)}, \boldsymbol{h}_{r|q}^{(T)}\big) \mid w \in \mathcal{N}_r(v), r \in R\}\!\} = \{\!\{\big(\boldsymbol{h}_{w|u',q'}^{(\ell)}, \boldsymbol{h}_{r|q'}^{(T)}\big) \mid w \in \mathcal{N}_r(v'), r \in R\}\!\}.$$

It follows that

$$N = \{\!\{\text{MSG}\big(\boldsymbol{h}_{w|u,q}^{(\ell)}, \text{Enc}_{\text{LIFT},q}[r]\big) \mid w \in \mathcal{N}_r(v), r \in R\}\!\} = \{\!\{\text{MSG}\big(\boldsymbol{h}_{w|u',q'}^{(\ell)}, \text{Enc}_{\text{LIFT},q'}[r]\big) \mid w \in \mathcal{N}_r(v'), r \in R\}\!\} = N'.$$

We conclude that

$$\boldsymbol{h}_{v|u,q}^{(\ell+1)} = \text{UP}_2\big(\boldsymbol{h}_{v|u,q}^{(\ell)}, \text{AGG}_2(N)\big) = \text{UP}_2\big(\boldsymbol{h}_{v'|u',q'}^{(\ell)}, \text{AGG}_2(N')\big) = \boldsymbol{h}_{v'|u',q'}^{(\ell+1)}.$$

as required. ($\text{UP}_2$ and $\text{AGG}_2$ are the update and aggregate functions in the $(\ell+1)$-th layer.) $\qquad\square$

We now show that $\text{MOTIF}(\mathcal{F})$ can be as powerful as our test, regarding separating links.

**Proposition C.2.** *Let $G = (V, E, R)$ be a KG, and let $T, L \ge 0$ be natural numbers. Then there exists an instance $(\text{Enc}_{\text{KG}}, (\mathcal{F}, \text{Enc}_{\text{LIFT}}))$ of $\text{MOTIF}(\mathcal{F})$ such that for every pair of links $q(u,v)$ and $q'(u',v')$, we have that*

$$\text{col}_{\mathcal{F},T}^{(L)}(q(u,v)) = \text{col}_{\mathcal{F},T}^{(L)}(q'(u',v')) \iff \text{Enc}_{\text{KG},u,q}[v] = \text{Enc}_{\text{KG},u',q'}[v'].$$

*Proof.* It suffices to choose $(\mathsf{Enc}_{\mathrm{KG}}, (\mathcal{F}, \mathsf{Enc}_{\mathrm{LIFT}}))$ such that the functions $\mathrm{UP}_1^{(t)}$, $\mathrm{AGG}_1^{(t)}$ and $\mathrm{MSG}_{\rho(e)}^{(t)}$ defining $\mathsf{Enc}_{\mathrm{LIFT}}$, and the functions $\mathrm{UP}_2^{(\ell)}$, $\mathrm{AGG}_2^{(\ell)}$ and $\mathrm{MSG}^{(\ell)}$ defining $\mathsf{Enc}_{\mathrm{KG}}$, are injective. Indeed, suppose this condition holds. We prove first by induction that for every $q, r, q', r' \in R$ and $0 \le t \le T$,

$$\boldsymbol{h}_{r|q}^{(t)} = \boldsymbol{h}_{r'|q'}^{(t)} \implies \mathsf{hcwl}_{\mathcal{F}}^{(t)}(q, r) = \mathsf{hcwl}_{\mathcal{F}}^{(t)}(q', r').$$

For $t = 0$, the result follows by our assumption on $\mathrm{INIT}_1$. Suppose that $\boldsymbol{h}_{r|q}^{(t+1)} = \boldsymbol{h}_{r'|q'}^{(t+1)}$. As $\mathrm{UP}_1^{(t)}$ is injective, we have that $\boldsymbol{h}_{r|q}^{(t)} = \boldsymbol{h}_{r'|q'}^{(t)}$ and $\mathrm{AGG}_1^{(t)}(M) = \mathrm{AGG}_1^{(t)}(M')$. From the former and by induction, we obtain $\mathsf{hcwl}_{\mathcal{F}}^{(t)}(q, r) = \mathsf{hcwl}_{\mathcal{F}}^{(t)}(q', r')$. From the later, as $\mathrm{AGG}_1^{(t)}$ is injective, we obtain $M = M'$. Using the fact that $\mathrm{MSG}_{\rho(e)}^{(t)}$ is injective, we conclude that

$$\{\!\{\{(\boldsymbol{h}_{s|q}^{(t)}, j) \mid (s, j) \in \mathcal{N}^i(e)\} \mid (e, i) \in E_{\mathrm{LIFT}}(r)\}\!\} = \{\!\{\{(\boldsymbol{h}_{s|q'}^{(t)}, j) \mid (s, j) \in \mathcal{N}^i(e)\} \mid (e, i) \in E_{\mathrm{LIFT}}(r')\}\!\}.$$

By induction, we have

$$\{\!\{\{(\mathsf{hcwl}_{\mathcal{F}}^{(t)}(q, s), j) \mid (s, j) \in \mathcal{N}^i(e)\} \mid (e, i) \in E_{\mathrm{LIFT}}(r)\}\!\} = \{\!\{\{(\mathsf{hcwl}_{\mathcal{F}}^{(t)}(q', s), j) \mid (s, j) \in \mathcal{N}^i(e)\} \mid (e, i) \in E_{\mathrm{LIFT}}(r')\}\!\}.$$

This implies that $\mathsf{hcwl}_{\mathcal{F}}^{(t+1)}(q, r) = \mathsf{hcwl}_{\mathcal{F}}^{(t+1)}(q', r')$ as required. Now we prove by induction that for every pair of links $q(u, v)$ and $q'(u', v')$, and $0 \le \ell \le L$, it holds that

$$\boldsymbol{h}_{v|u,q}^{(\ell)} = \boldsymbol{h}_{v'|u',q'}^{(\ell)} \implies \mathsf{col}_{\mathcal{F},T}^{(\ell)}(q(u, v)) = \mathsf{col}_{\mathcal{F},T}^{(\ell)}(q'(u', v')).$$

Note that this implies that

$$\mathsf{col}_{\mathcal{F},T}^{(L)}(q(u, v)) = \mathsf{col}_{\mathcal{F},T}^{(L)}(q'(u', v')) \iff \mathsf{Enc}_{\mathrm{KG},u,q}[v] = \mathsf{Enc}_{\mathrm{KG},u',q'}[v'].$$

The implication holds for $\ell = 0$ by the assumption on $\mathrm{INIT}_2$ and the fact that $\boldsymbol{h}_{q|q}^{(T)} = \boldsymbol{h}_{q'|q'}^{(T)} \implies \mathsf{hcwl}_{\mathcal{F}}^{(T)}(q, q) = \mathsf{hcwl}_{\mathcal{F}}^{(T)}(q', q')$. Assume that $\boldsymbol{h}_{v|u,q}^{(\ell+1)} = \boldsymbol{h}_{v'|u',q'}^{(\ell+1)}$. As $\mathrm{UP}_2^{(t)}$ is injective, we have that $\boldsymbol{h}_{v|u,q}^{(\ell)} = \boldsymbol{h}_{v'|u',q'}^{(\ell)}$ and $\mathrm{AGG}_2^{(\ell)}(N) = \mathrm{AGG}_2^{(\ell)}(N')$. From the former and by induction, we obtain $\mathsf{col}_{\mathcal{F},T}^{(\ell)}(q(u, v)) = \mathsf{col}_{\mathcal{F},T}^{(\ell)}(q'(u', v'))$. From the later, since $\mathrm{AGG}_2^{(\ell)}$ is injective, we have $N = N'$. As $\mathrm{MSG}^{(\ell)}$ is injective, we obtain that

$$\{\!\{(\boldsymbol{h}_{w|u,q}^{(\ell)}, \boldsymbol{h}_{r|q}^{(T)}) \mid w \in \mathcal{N}_r(v), r \in R\}\!\} = \{\!\{(\boldsymbol{h}_{w|u',q'}^{(\ell)}, \boldsymbol{h}_{r|q'}^{(T)}) \mid w \in \mathcal{N}_r(v'), r \in R\}\!\}.$$

By induction, we have

$$\{\!\{(\mathsf{col}_{\mathcal{F},T}^{(\ell)}(q(u, v)), \mathsf{hcwl}_{\mathcal{F}}^{(T)}(q, r)) \mid w \in \mathcal{N}_r(v), r \in R\}\!\} = \{\!\{(\mathsf{col}_{\mathcal{F},T}^{(\ell)}(q'(u', v')), \mathsf{hcwl}_{\mathcal{F}}^{(T)}(q', r)) \mid w \in \mathcal{N}_r(v'), r \in R\}\!\}.$$

This implies that $\mathsf{col}_{\mathcal{F},T}^{(\ell+1)}(q(u, v)) = \mathsf{col}_{\mathcal{F},T}^{(\ell+1)}(q'(u', v'))$ as desired.

It is easy to find injective funtions $\mathrm{UP}_1^{(t)}$, $\mathrm{AGG}_1^{(t)}$, $\mathrm{MSG}_{\rho(e)}^{(t)}$, $\mathrm{UP}_2^{(\ell)}$, $\mathrm{AGG}_2^{(\ell)}$, $\mathrm{MSG}^{(\ell)}$. For instance $\mathrm{UP}_1^{(t)}$ and $\mathrm{UP}_2^{(\ell)}$ can be vector concatenation, and $\mathrm{AGG}_1^{(t)}$, $\mathrm{MSG}_{\rho(e)}^{(t)}$, $\mathrm{AGG}_2^{(\ell)}$, $\mathrm{MSG}^{(\ell)}$ can be chosen using Lemma 5 from Xu et al. (2019) (see also Lemma VIII.5 from Grohe (2021)). Another alternative is to choose $\mathrm{UP}_1^{(t)}$, $\mathrm{AGG}_1^{(t)}$, $\mathrm{MSG}_{\rho(e)}^{(t)}$ according to Theorem 4.1 in Huang et al. (2024), and $\mathrm{UP}_2^{(\ell)}$, $\mathrm{AGG}_2^{(\ell)}$, $\mathrm{MSG}^{(\ell)}$ according to Theorem 5.1 in Huang et al. (2023).

$\square$

As a direct corollary of Propositions C.1 and C.2, we obtain:

**Corollary C.3.** *Let $\mathcal{F}$ and $\mathcal{F}'$ be two set of motifs. Then $\mathcal{F} \preceq \mathcal{F}'$ if and only if for every KG $G = (V, E, R)$ and link $q(u, v)$, $q'(u', v')$, whenever $\mathsf{col}_{\mathcal{F},T}^{(L)}(q(u, v)) = \mathsf{col}_{\mathcal{F},T}^{(L)}(q'(u', v'))$ for every $T, L \ge 0$, then $\mathsf{col}_{\mathcal{F}',T}^{(L)}(q(u, v)) = \mathsf{col}_{\mathcal{F}',T}^{(L)}(q'(u', v'))$ for every $T, L \ge 0$.*

# D. Missing proofs in the paper

## D.1. Proof of Proposition 6.2

*Let $P = (G_M, \bar{r})$ and $P' = (G'_M, \bar{r}')$ be two graph motifs such that $G_M$ is isomorphic to $G'_M$. Then, for any set $\mathcal{F}$ of graph motifs:*

$$(\mathcal{F} \cup \{P\}) \sim (\mathcal{F} \cup \{P'\}) \sim (\mathcal{F} \cup \{P, P'\}).$$

*Proof.* We only show that $(\mathcal{F} \cup \{P\}) \sim (\mathcal{F} \cup \{P, P'\})$, the part that $(\mathcal{F} \cup \{P\}) \sim (\mathcal{F} \cup \{P'\})$ is analogous.

For readability, let us write $\mathcal{F}_1 = \mathcal{F} \cup \{P\}$ and $\mathcal{F}_2 = \mathcal{F} \cup \{P, P'\}$. Further, $E_1$ and $E_2$ correspond to the edges of $\text{LIFT}_{\mathcal{F}_1}(G)$ and $\text{LIFT}_{\mathcal{F}_2}(G)$, respectively.

We make use of Corollary C.3, and show that for every KG $G = (V, E, R)$, every pair of links $q(u, v), q'(u', v')$ and every $T, L \geq 0$,

$$\text{col}^{(L)}_{\mathcal{F}_1, T}(q(u, v)) = \text{col}^{(L)}_{\mathcal{F}_1, T}(q(u', v')) \iff \text{col}^{(L)}_{\mathcal{F}_2, T}(q(u, v)) = \text{col}^{(L)}_{\mathcal{F}_2, T}(q(u', v')).$$

By definition of the second stage colors, all we need to show is that for every $s, s' \in R$,

$$\text{hcwl}^{(T)}_{\mathcal{F}_1}(q, s) = \text{hcwl}^{(T)}_{\mathcal{F}_1}(q', s') \iff \text{hcwl}^{(T)}_{\mathcal{F}_2}(q, s) = \text{hcwl}^{(T)}_{\mathcal{F}_2}(q', s').$$

Once again, we only need to show one direction, the other one is analogous. We prove the ($\Rightarrow$) direction by induction on $0 \leq t \leq T$. The base case is immediate because the inizialization does not depend on $\mathcal{F}_1$ or $\mathcal{F}_2$. Assume then this condition holds for all $t < T$. On step $t$, let $q, s$ and $q', s'$ be pairs satisfying the left hand side. According to the coloring step, $\text{hcwl}^{(t)}_{\mathcal{F}_1}(q, s)$ corresponds to the hash of a pair given by the previous color $\text{hcwl}^{(t-1)}_{\mathcal{F}_2}(q, s)$ and the multiset of the colors of all neighbours of $q$ in any tuple in any relation in $\mathcal{F}_1$. For $\mathcal{F}_2$, the color is the same except we include

$$\{\!\!\{ \big(\{(\text{hcwl}^{(t)}_{\mathcal{F}_2}(q, x), j) \mid (x, j) \in \mathcal{N}^i(P')\}, \rho(e)\big) \mid (P', i) \in E_2(s) \}\!\!\},$$

where $E_2(s)$ is the set of pairs $(e, i)$ where $e$ is an edge in $E_2$ and $e(i) = s$. Hence, since this is the only term that differs from the coloring according to $\mathcal{F}_1$, to prove the right hand side, it suffices to show that the above multiset is the same as

$$\{\!\!\{ \big(\{(\text{hcwl}^{(t)}_{\mathcal{F}_2}(q', x'), j') \mid (x', j') \in \mathcal{N}^{i'}(P')\}, \rho(e)\big) \mid (P', i') \in E_2(s') \}\!\!\},$$

Now, because of isomorphism, there is a bijection $f$ that maps each index position $i$ of $\bar{r}$ to a position $f(i)$ of $\bar{r}'$, in such a way that a pair $(P, i)$ is in $E_1(s)$ if and only if $(P', f(i))$ is in $E_2(s)$. Further, for every fact $P(r_1, \ldots, r_n)$ in $\text{LIFT}_{\mathcal{F}_1}(G)$, there is a fact $P(r'_1, \ldots, r'_n)$ such that $r_i = r'_f(i)$. Hence, for every pair $(x, j) \in \mathcal{N}^k(P')$, $(P', k) \in E_2(s)$, there is a pair $(x, p)$ in $\mathcal{N}^{f^{-1}(k)}(P)$, $(P, f^{-1}(i)) \in E_1$. By induction, since $\text{hcwl}^{(t)}_{\mathcal{F}_1}(q, s) = \text{hcwl}^{(t)}_{\mathcal{F}_1}(q', s')$, the color of $(q, x)$ by $\text{hcwl}^{(t-1)}_{\mathcal{F}_1}$ must then correspond to the color of some node $(q', x')$ in $\mathcal{N}^{f^{-1}(k)}(P)$, $(P, f^{-1}(i)) \in E_1(s')$. Once again, by isomorphism, we find the pair $(x', j) \in \mathcal{N}^k(P')$, $(P', k) \in E_2(s')$, where by induction the color of $(q', x')$ by $\text{hcwl}^{(t-1)}_{\mathcal{F}_2}$ now corresponds to that of $(q, x)$. This proves a bijection between all elements in the multiset, from which we directly obtain that $\text{hcwl}^{(t)}_{\mathcal{F}_2}(q, s) = \text{hcwl}^{(t)}_{\mathcal{F}_2}(q', s')$. $\square$

## D.2. Proof of Proposition 6.3

*For every KG $G$, up to isomorphism, there is a unique KG $H$ such that:*

- *$H$ is a relation-preserving core, and*

- *there are relation-preserving homomorphisms from $G$ to $H$ and from $H$ to $G$.*

*Proof.* Let us assume there are two such cores $H_1$ and $H_2$. Since there are relation-preserving homomorphisms from $G$ to $H_1$, from $G$ to $H_2$, from $H_1$ to $G$ and from $H_2$ to $G$, componing these homomorphisms establishes there are relation-preserving homomorphisms $f$ from $H_1$ to $H_2$ and $g$ from $H_2$ to $H_1$; note that the composition of two relation-preserving homomorphisms is also a relation-preserving homomorphism.

But now $f \circ g$ is a relation-preserving homomorphism from $H_1$ to $H_1$ and $g \circ f$ is a homomorphsim from $H_2$ to $H_2$. This means that $f \circ g$ and $g \circ f$ are onto. Moreover, since they are endomorphisms, they must be an automorphism. Hence, $f$ and $g$ are both onto, which means that $H_1$ and $H_2$ are isomorphic.

$\square$

### D.3. Proof of Theorem 6.4

*Let $\mathcal{F}, \mathcal{F}'$ be two sets of motifs, and assume that $\mathcal{F} \preceq \mathcal{F}'$. Then, for every non-trivial motif $P' \in \mathcal{F}'$ there is a motif $P \in \mathcal{F}$ such that $P \to^{co} P'$.*

*Proof.* We prove the converse of this statement: assume that there is a non-trivial motif $P' \in \mathcal{F}'$ that does not satisfy the conditions of the Theorem, that is, that there is no motif $P \in \mathcal{F}$ such that $P \to^{co} P'$. We show how to construct a KG $G$ and a pair of links $q(u, v)$ and $q'(u', v')$ such that $\mathsf{col}_{\mathcal{F},T}^{(L)}(q(u,v)) = \mathsf{col}_{\mathcal{F},T}^{(L)}(q(u',v'))$ for every $T, L \geq 0$, but $\mathsf{col}_{\mathcal{F}',T}^{(L)}(q(u,v)) \neq \mathsf{col}_{\mathcal{F}',T}^{(L)}(q(u',v'))$, for some $T, L \geq 0$. This, by Corollary C.3, suffices for the proof.

Let us write $P' = (G_{P'}, \bar{r}')$. We split the proofs into two cases, depending on whether there exists a motif $P'$ in $\mathcal{F}$ such that there is no homomorphism from any motif $P$ in $\mathcal{F}$ to $P'$.

**Case 1**. Assume that $\mathcal{F}'$ contains a motif $P'$ such that there is no node-relation homomorphism from any motif $P$ in $\mathcal{F}$ to $P'$. Notice that this implies that all motifs in $\mathcal{F}$ cannot be mapped to the trivial motif with just the fact $Y(x, z)$, as otherwise, we would have motifs in $\mathcal{F}$ that can be mapped to any edge in $P'$, resulting in a node-relation homomorphism to motif $P'$.

Let $y$ be an arbitrary relation in $P'$ and $s$ a fresh relation. Construct $G$ by taking $G_{P'}$ plus two triples $y, (a, b)$ and $s(c, d)$, with $a, b, c, d$ fresh nodes. We claim that $\mathsf{col}_{\mathcal{F}'}(G, y(a, b)) \neq \mathsf{col}_{\mathcal{F}'}(G, s(c, d))$ but $\mathsf{col}_{\mathcal{F}}(G, y(a, b)) = \mathsf{col}_{\mathcal{F}}(G, s(c, d))$, which is what we were looking for.

By our assumptions it follows that $\mathrm{LIFT}_{\mathcal{F}}(G)$ is a graph without relations, as there are no homomorphisms from any motif in $\mathcal{F}$ to $G$. Therefore, we have that $\mathsf{hcwl}_y(y) = \mathsf{hcwl}_s(s)$ over $\mathrm{LIFT}_{\mathcal{F}}(G)$: in both cases the hcwl coloring involves a single node with the same color, not connected to any other node, so we stop after just one iteration. In turn, this entails that $\mathsf{col}_{\mathcal{F}}(y(a, b)) = \mathsf{col}_F(s(c, d))$, as once again nodes $a, b$ and $c, d$ are not connected to any other part in $G$.

On the other hand, $\mathrm{LIFT}_{\mathcal{F}'}(G)$ contains at least one tuple with distinct elements in the relation $R_{P'}$ corresponding to motif $P'$, consisting of $\bar{r}'$: this comes from the identity homomorphism from $P'$ to $G_{P'}$. But since $P'$ is non-trivial, there cannot be a relation-preserving homomorphism $h$ from $P'$ to the graph formed just by fact $s(u, v)$, as this would entail that the relation preserving core of $P'$ is indeed $s(u, v)$, and hence $P'$ would be trivial, which we assumed otherwise. Then, there cannot be a tuple of different relations that includes $s$ in relation $R_{P'}$ in $\mathrm{LIFT}_{\mathcal{F}'}(G)$. This entails that the color $\mathsf{hcwl}_y(y)$ and $\mathsf{hcwl}_s(s)$ over $\mathrm{LIFT}_{\mathcal{F}'}(G)$ are already different after the first step. Hence, $\mathsf{col}_{\mathcal{F}'}(y(a, b)) \neq \mathsf{col}_{\mathcal{F}'}(s(c, d))$ as promised.

**Case 2**. Next assume that there is a motif $P'$ in $F'$ such that there are no core-onto homomorphisms from any motif $P$ in $\mathcal{F}$ to $P$ (but there may be regular node-relation homomorphisms).

Let then $H$ contain all different motifs (up to isomorphism) $h(P) = (h(\bar{r}), h(G_P))$[3] such that $P \in \mathcal{F}$ and $h$ is a homomorphism from $P$ to the relation preserving core of $P'$ By our assumptions, none of these homomorphisms is core-onto. Define two functions $f_1$ and $f_2$ that map each relation name from a motif in $H$ to a fresh relation, and that are the identity on nodes. Further, construct a graph $G_1$ as follows: for every motif $h(P)$ in $H$, replace every node in $f_1(h(P))$ by a fresh node, and add it to $G_1$. Construct graph $G_2$ in the same fashion except we use $f_2(h(P))$. Finally, extend $f_1$ to be the identity on all other relations in $(\bar{r}')$ (recall that $P' = (\bar{r}', G_{P'})$ not in any motif in $H$, so that $f_1(P')$ is a well-defined motif and its corresponding graph $G_{f_1(P')}$ is a well defined graph.

Our graph $G$ corresponds to the union of $G_1$, $G_2$ and $G_{f_1(P')}$. Clearly, $G_1$ and $G_2$ are isomorphic, let $g$ be one such isomorphism. By construction, we also have that $\mathrm{LIFT}_{\mathcal{F}}(G_1)$ is isomorphic to $\mathrm{LIFT}_{\mathcal{F}}(G_2)$ and $\mathrm{LIFT}_{\mathcal{F}'}(G_1)$ is isomorphic to $\mathrm{LIFT}_{\mathcal{F}'}(G_2)$.

For an arbitrary motif in $H$ and a fact $y(x, z)$ of this motif, consider facts $f_1(y)(a, b)$ and $f_2(y)(g(a), g(b))$ in $G_1$ and $G_2$, respectively, that result from processing triple $y(x, z)$ in this motif according to our construction (here all of $a, b, g(a), g(b)$ are fresh nodes). As in **Case 1**, we show that these triples are separated by $\mathsf{col}_{\mathcal{F}}'$ but not by $\mathsf{col}_{\mathcal{F}}$.

---

[3]Slightly abusing the notation, we use $h(G_P)$ to denote the motif in which every fact $b(a, c)$ is replaced by $h(b)((h(a), h(c))$.

First, note that $\text{LIFT}_{\mathcal{F}}(G) = \text{LIFT}_{\mathcal{F}}(G_1) \cup \text{LIFT}_{\mathcal{F}}(G_2)$, as (1) $G_1$ and $G_2$ are not connected, and (2) any homomorphism $h$ from a motif $P$ in $\mathcal{F}$ to $G_1 \cup G_{f_1(P')}$ has a corresponding homomorphism $h^*$ from $P$ to $G_1$ such that $h(P)$ and $h^*(P)$ are isomorphic, and therefore any tuple in the relation $R_P$ in $\text{LIFT}_{\mathcal{F}}(G_1 \cup G_{f_1(P')})$ is also in the relation $R_P$ of $\text{LIFT}_{\mathcal{F}}(G_1)$.

The fact that $\text{LIFT}_{\mathcal{F}}(G) = \text{LIFT}_{\mathcal{F}}(G_1) \cup \text{LIFT}_{\mathcal{F}}(G_2)$, that $\text{LIFT}_{\mathcal{F}}(G_1)$ and $\text{LIFT}_{\mathcal{F}}(G_2)$ are isomorphic and that they are not connected establishes that $\text{hcwl}_{f_1(Y)}(f_1(X)) = \text{hcwl}_{f_2(Y)}(f_2(X))$ for any relation name $X$ in $G$. Then, again since both connected components in $\text{LIFT}_{\mathcal{F}}(G)$ are isomorphic, and our game is invariant to isomorphisms (assuming the same coloring of relations), it must be the case that $\text{col}_{\mathcal{F}}(f_1(Y)(a,b)) = \text{col}_{\mathcal{F}}(f_2(Y)(g(a),g(b)))$.

Let us now look at $\text{LIFT}_{\mathcal{F}'}(G)$. Notice first that any tuple $t$ in $R_{P'}$ in $\text{LIFT}_{\mathcal{F}'}(G_2)$ has a corresponding tuple $f_2 \circ f_1^{-1}(t)$ in $\text{LIFT}_{\mathcal{F}'}(G_1)$. In other words, $f_2 \circ f_1^{-1}$ is a bijection from $\text{LIFT}_{\mathcal{F}'}(G_2)$ to $\text{LIFT}_{\mathcal{F}'}(G_1 \cup G_{f_1(P')})$ whose inverse is $f_1 \circ f_2^{-1}$. This is because these tuples arise from homomorphisms that start from $P'$ and end in some motif in $H$, and both $G_1$ and $G_2$ have the same copy of this motif.

To finish we claim that there one tuple $t$ in relation $R_{P'}$ in graph $\text{LIFT}_{\mathcal{F}'}(G_1 \cup G_{f_1(P')})$ such that $f_1 \circ f_2^{-1}(t)$ is not in relation $R_{P'}$ in graph $\text{LIFT}_{\mathcal{F}'}(G_2)$. This means that for any element $r$ in $t$ the cardinality of edges in $R_{P'}$ where $r$ participates is strictly higher than the cardinality of edges where $f_1 \circ f_2^{-1}(r)$ participate, which implies that $\text{hcwl}_r(s) \neq \text{hcwl}_{f_1 \circ f_2^{-1}(r)}(f_1 \circ f_2^{-1}(s))$ for any relation name $s$ that belongs to $t$ because the color in step (1) is already different. As in **Case 1** above, this can be used to show that $\text{col}_{\mathcal{F}'}(f_1(Y)(a,b)) \neq \text{col}_{F'}(f_2(Y)(g(a),g(b)))$.

The tuple $t$ corresponds to $f_1(\bar{r}')$. This tuple exists in relation $R_{P'}$ in $\text{LIFT}_{\mathcal{F}'}(G_1 \cup G_{f_1(P')})$ because $f_1$ is a homomorphism from $P'$ to $G_{f_1(P')}$ We show that $\text{LIFT}_{\mathcal{F}'}(G)$ does not contain any tuple consisting of all different nodes from $f_2(\bar{r}')$ in relation $R_{P'}$, which suffices for our claim. Note that this implies that all relation variables from $P'$ are in the preimage of $f_2$ (i.e., are part of a motif in $H$), otherwise the claim is obvious as the size of $\bar{r}'$ is bigger than the range of $f_2$.

Suppose for the sake of contradiction that such a tuple $f_2(\bar{r}')$ is in $R_{P'}$. By construction, any tuple from elements in $f_2(\bar{r}')$ must come from $\text{LIFT}_{\mathcal{F}'}(G_2)$. Hence, tuple $f_2(\bar{r}')$ is added to $\text{LIFT}_{\mathcal{F}'}(G_2)$ due to a homomorphism $\hat{h}$ from $P'$ to a motif $h(P)$ in $H$ (for a motif $P$ in $\mathcal{F}$ and a homomorphism $h$ from $P$ to the core of $P'$), and since the tuple contains all elements in $f_2(\bar{r}')$, this homomorphism $\hat{h}$ is relation-preserving. By composition of homomorphisms this also entails a relation-preserving homomorphism from the (relation-preserving) core of $P'$ to $h(P)$.

Further, since $h$ is a homomorphism from $P$ to the relation-preserving core of $P'$, the identity is a homomorphism from $h(P)$ to said core of $P'$, and since $h(P)$ contains all relations in $f_2(\bar{r}')$, this homomorphism is also relation-preserving. This is a contradiction because we have found that $h(P)$ is the relation preserving core of $P'$ and therefore $h$ is a core-onto homomorphism from $P$ to $P'$. $\qquad\square$

### D.4. Proof of Theorem 6.5

ULTRA has the same expressive power as $\text{MOTIF}(\mathcal{F}_2^{\text{path}})$.

*Proof.* By Proposition 6.2, we show that $\text{MOTIF}(\{h2t, h2h, t2t\})$ and $\text{MOTIF}(\{t2h, h2h, t2t\})$ has the same expressive power as $\text{MOTIF}(\{h2t, t2h, h2h, t2t\})$. We also note that $\{t2h, h2h, t2t\}$ and $\{h2t, h2h, t2t\}$ are the same as $\mathcal{F}_2^{\text{path}}$. Thus, it is enough to show that ULTRA, in its general form, has the same separating power than $\text{MOTIF}(\mathcal{F})$, where $\mathcal{F} = \{h2t, t2h, h2h, t2t\}$.

Recall that the separating power of $\text{MOTIF}(\mathcal{F})$ can be characterized by the coloring scheme $\text{col}_{\mathcal{F}}$ defined in Section C. Following the same strategy as in Propositions C.1 and C.2, it is straightforward to show that the separating power of ULTRA matches the following test.

Let $G = (V, E, R)$ be a KG. The first-stage colors $\text{rcol}_{\text{ultra}}^{(t)}$ are defined via the following rules:

$$\text{rcol}_{\text{ultra}}^{(0)}(q, r) = \mathbb{1}_{r=q} * 1$$
$$\text{rcol}_{\text{ultra}}^{(t+1)}(q, r) = \text{HASH}\left(\text{rcol}_{\text{ultra}}^{(t)}(q, r), \{\!\{ \left(\text{rcol}_{\text{ultra}}^{(t)}(q, r'), r_{\text{LIFT}}\right) \mid r' \in \mathcal{N}_{r_{\text{LIFT}}}(r), r_{\text{LIFT}} \in R_{\text{LIFT}} \}\!\}\right).$$

$\mathbb{1}_{r=q}$ takes value 1 if $r = q$, and 0 otherwise. The function $\text{HASH}$ is a fixed injective function taking values in the natural numbers.

The second-stage colors $\mathsf{ecol}_{\mathsf{ultra},T}^{(\ell)}$, where $T \geq 0$ is a parameter, are defined via the following rules:

$$\mathsf{ecol}_{\mathsf{ultra},T}^{(0)}(q(u,v)) = \mathbb{1}_{v=u} * \mathsf{rcol}_{\mathsf{ultra}}^{(T)}(q,q)$$
$$\mathsf{ecol}_{\mathsf{ultra},T}^{(\ell+1)}(q(u,v)) = \mathrm{HASH}\Big(\mathsf{ecol}_{\mathsf{ultra},T}^{(\ell)}(q(u,v)), \{\!\{\big(\mathsf{ecol}_{\mathsf{ultra},T}^{(\ell)}(q(u,w)), \mathsf{rcol}_{\mathsf{ultra}}^{(T)}(q,r)\big) \mid w \in \mathcal{N}_r(v), r \in R\}\!\}\Big).$$

We aim to show that for every links $q(u,v), q'(u',v')$ and $T, L \geq 0$, we have

$$\mathsf{ecol}_{\mathsf{ultra},T}^{(L)}(q(u,v)) = \mathsf{ecol}_{\mathsf{ultra},T}^{(L)}(q'(u',v')) \iff \mathsf{col}_{\mathcal{F},T}^{(L)}(q(u,v)) = \mathsf{col}_{\mathcal{F},T}^{(L)}(q'(u',v')).$$

Since the definition of the second-stage colorings $\mathsf{ecol}_{\mathsf{ultra},T}$ and $\mathsf{col}_{\mathcal{F},T}$ are identical, it suffices to show that for every $q, r, q', r' \in R$, and every $0 \leq t \leq T$, it holds that

$$\mathsf{rcol}_{\mathsf{ultra}}^{(t)}(q,r) = \mathsf{rcol}_{\mathsf{ultra}}^{(t)}(q',r') \iff \mathsf{hcwl}_{\mathcal{F}}^{(t)}(q,r) = \mathsf{hcwl}_{\mathcal{F}}^{(t)}(q',r').$$

This propositions follows from the following observations. The update rule of $\mathsf{rcol}_{\mathsf{ultra}}^{(t)}$ coincides with the one of $\mathsf{rawl}_2^{(t)}$ (defined in Section A) applied to $\mathrm{LIFT}_{\mathcal{F}}(G)$. On the other hand, the update rule of $\mathsf{hcwl}_{\mathcal{F}}^{(t)}$ coincides with the one of $\mathsf{hcwl}_2^{(t)}$ (defined in Section A) applied to $\mathrm{LIFT}_{\mathcal{F}}(G)$. It follows from Theorem H.4 in Huang et al. (2024) that $\mathsf{rawl}_2^{+(t)}$ is equivalent to $\mathsf{hcwl}_2^{(t)}$. Here $\mathsf{rawl}_2^{+(t)}$ corresponds to the test $\mathsf{rawl}_2^{(t)}$ applied to the augmented graph $G^+$, which extends each fact $r(u,v)$ in $G$ with its inverse fact $r^-(v,u)$ (see definition in Section A). The key observation is that over $\mathrm{LIFT}_{\mathcal{F}}(G)$, the test $\mathsf{rawl}_2^{(t)}$ and $\mathsf{rawl}_2^{+(t)}$ are also equivalent. This is because of the structure of $\mathcal{F} = \{h2t, t2h, h2h, t2t\}$, as observed in Appendix F: The relation $h2h$ is symmetric, and hence $h2h^-(q,r) \in \mathrm{LIFT}_{\mathcal{F}}(G)^+ \iff h2h(q,r) \in \mathrm{LIFT}_{\mathcal{F}}(G)^+$. Similarly for $t2t$. On the other hand, $h2t(q,r) \in \mathrm{LIFT}_{\mathcal{F}}(G) \iff t2h(r,q) \in \mathrm{LIFT}_{\mathcal{F}}(G)$. Hence, $h2t^-(q,r) \in \mathrm{LIFT}_{\mathcal{F}}(G)^+ \iff t2h(q,r) \in \mathrm{LIFT}_{\mathcal{F}}(G)^+$. Similarly for $t2h$. Therefore, adding inverses over $\mathrm{LIFT}_{\mathcal{F}}(G)$ is superfluous, and $\mathsf{rawl}_2^{(t)}$ and $\mathsf{rawl}_2^{+(t)}$ are equivalent. We conclude that $\mathsf{rawl}_2^{(t)}$ and $\mathsf{hcwl}_2^{(t)}$ are equivalent, and then $\mathsf{rcol}_{\mathsf{ultra}}^{(t)}$ and $\mathsf{hcwl}_{\mathcal{F}}^{(t)}$ are equivalent, as required. $\square$

## E. MOTIF architecture

In this section, we will introduce the architecture of MOTIF used in the experimental sections. We adopt HCNets (Huang et al., 2024) as the relation encoder for the relational hypergraphs, and following ULTRA (Galkin et al., 2024), we adopt a slight modification of NBFNet (Zhu et al., 2021) as the entity encoder.

**Model architectures.** Specifically, for the relation encoder, we have that for $0 \leq t \leq T$,

$$\boldsymbol{h}_{r|q}^{(0)} = \mathbb{1}_{r=q} * \mathbf{1}^d,$$
$$\boldsymbol{h}_{r|q}^{(t+1)} = \sigma\Big(\boldsymbol{W}^{(t)}\Big[\boldsymbol{h}_{v|q}^{(t)} \Big\| \sum_{(e,i) \in E_{\mathrm{LIFT}}(r)} \boldsymbol{z}_{\rho(e)}^{(t)} \odot \big(\odot_{j \neq i} (\alpha^{(t)} \boldsymbol{h}_{e(j)|q}^{(t)} + (1-\alpha^{(t)})\boldsymbol{p}_j)\big)\Big] + \boldsymbol{b}^{(t)}\Big),$$

For entity encoder, we follow ULTRA (Galkin et al., 2024) and adopt the same variation of NBFNet (Zhu et al., 2021), which computes the pairwise representation as follows for $0 \leq \ell \leq L$:

$$\boldsymbol{h}_{v|u,q}^{(0)} = \mathbb{1}_{v=u} * \boldsymbol{h}_{q|q}^{(T)}$$
$$\boldsymbol{h}_{v|u,q}^{(\ell+1)} = \sigma\Big(\boldsymbol{W}^{(\ell)}\Big[\boldsymbol{h}_{v|u,q}^{(0)} \Big\| \sum_{r \in R} \sum_{w \in \mathcal{N}_r(v)} \boldsymbol{h}_{w|u,q}^{(\ell)} \odot \mathrm{MLP}^{(\ell)}(\boldsymbol{h}_{r|q}^{(T)})\Big] + \boldsymbol{b}^{(\ell)}\Big),$$

where $\sigma$ is ReLU activation, $\boldsymbol{W}$ and $\boldsymbol{b}$ are learnable weight matrix and bias term per layer, $\mathbf{1}^d$ is an all-one vector of $d$ dimension, and $\mathbb{1}_C$ is the indicator function that returns 1 if condition $C$ is true, and 0 otherwise. We denote $*$ as scalar multiplication, and $\odot$ as element-wise multiplication of vectors. In the relation encoder, we note $\alpha$ to be a learnable scalar and $\boldsymbol{p}_i$ to be a learnable positional encoding at $i$-th position. Finally. we decode the probability of a query $q(u,v)$ by passing through representation $\boldsymbol{h}_{v|u,q}^{(L)}$ through a Multi-Layer Perceptron (MLP).

# F. ULTRA

In this section, we follow Galkin et al. (2024) and define ULTRA, a KGFM that computes invariants on nodes and relations on knowledge graphs.

**Model architectures.** Given a knowledge graph $G = (V, E, R, c)$ and a query $q(u, ?)$, ULTRA first constructs a knowledge graph $\text{LIFT}_{\mathcal{F}}(G) = (V_{\text{LIFT}}, E_{\text{LIFT}}, R_{\text{LIFT}})$ with each node representing a relation $r$, and 4 *fundamental* relations in $R_{\text{LIFT}}$, defined as follow:

- tail-to-head (*t2h*) edges, $t2h(r_1, r_2) \in E_{\text{LIFT}} \iff \exists u, v, w \in V, r_1(u, v) \land r_2(v, w) \in E$

- head-to-head (*h2h*) edges, $h2h(r_1, r_2) \in E_{\text{LIFT}} \iff \exists u, v, w \in V, r_1(v, u) \land r_2(v, w) \in E$

- head-to-tail (*h2t*) edges, $h2t(r_1, r_2) \in E_{\text{LIFT}} \iff \exists u, v, w \in V, r_1(v, u) \land r_2(w, v) \in E$

- tail-to-tail (*t2t*) edges, $t2t(r_1, r_2) \in E_{\text{LIFT}} \iff \exists u, v, w \in V, r_1(u, v) \land r_2(w, v) \in E$

Then, ULTRA applies a relation encoder to generate representation $\boldsymbol{h}_{r|q}$ via a variant of NBFNet (Zhu et al., 2021) that initialize with all-one vector $\mathbf{1}^d$ on the queried relations $q \in V_{\text{LIFT}}$. For $0 \leq t \leq T$, the relation encoder iteratively computes $\boldsymbol{h}_{r|q}^{(t)}$ as follow:

$$\boldsymbol{h}_{r|q}^{(0)} = \mathbb{1}_{r=q} * \mathbf{1}^d$$
$$\boldsymbol{h}_{r|q}^{(t+1)} = \sigma\Big(\boldsymbol{W}^{(t)}\Big[\boldsymbol{h}_{r|q}^{(0)}\| \sum_{r_{\text{LIFT}} \in R_{\text{LIFT}}} \sum_{r' \in \mathcal{N}_{R_{\text{LIFT}}}(r)} \boldsymbol{h}_{r'|q}^{(t)} \odot \boldsymbol{z}_{r_{\text{LIFT}}}\Big] + \boldsymbol{b}^{(t)}\Big),$$

where $\boldsymbol{z}_{r_{\text{LIFT}}}$ are learnable vectors for each one of the fundamental relations, $\sigma$ is ReLU activations, and similar to previous notation, $\|$ stands for concatenations, and $\boldsymbol{W}^{(t)}$ and $\boldsymbol{z}^{(t)}$ are learnable weight matrices and bias vectors for each $0 \leq t \leq T$, correspondingly.

Finally, after the relational embedding $\boldsymbol{h}_{v|q}^{(T)}$ is obtained given the query $q(u, ?)$, ULTRA proceeds to computes the final pairwise embedding for $0 \leq \ell \leq L$:

$$\boldsymbol{h}_{v|u,q}^{(0)} = \mathbb{1}_{v=u} * \boldsymbol{h}_{q|q}^{(T)}$$
$$\boldsymbol{h}_{v|u,q}^{(\ell+1)} = \sigma\Big(\boldsymbol{W}^{(\ell)}\Big[\boldsymbol{h}_{v|u,q}^{(0)}\| \sum_{r \in R} \sum_{w \in \mathcal{N}_r(v)} \boldsymbol{h}_{w|u,q}^{(\ell)} \odot \text{MLP}^{(\ell)}(\boldsymbol{h}_{r|q}^{(T)})\Big] + \boldsymbol{b}^{(\ell)}\Big),$$

**Frameworks.** We follow the notation of MOTIF and write ULTRA in its most general form. Later, we will also use ULTRA to refer to this general form when the context is clear. For the relational encoder, we have the following form:

$$\boldsymbol{h}_{r|q}^{(0)} = \text{INIT}_1(q, r)$$
$$\boldsymbol{h}_{r|q}^{(t+1)} = \text{UP}_1(\boldsymbol{h}_{r|q}^{(t)}, \text{AGG}_1(\{\!\!\{\text{MSG}_{r_{\text{LIFT}}}(\boldsymbol{h}_{r'|q}^{(t)})\mid r' \in \mathcal{N}_{r_{\text{LIFT}}}(r), r_{\text{LIFT}} \in R_{\text{LIFT}}\}\!\!\}))),$$

$\text{INIT}_1$, $\text{UP}_1$, $\text{AGG}_1$, and $\text{MSG}_r$ are differentiable *initialization*, *update*, *aggregation*, and fundamental-relation-specific *message* functions, respectively. Similarly for the entity encoder, we have this format:

$$\boldsymbol{h}_{v|u,q}^{(0)} = \text{INIT}_2(u, v, q)$$
$$\boldsymbol{h}_{v|u,q}^{(\ell+1)} = \text{UP}_2(\boldsymbol{h}_{v|u,q}^{(\ell)}, \text{AGG}_2(\{\!\!\{\text{MSG}(\boldsymbol{h}_{w|u,q}^{(\ell)}, \boldsymbol{h}_{r|q}^{(T)})\mid w \in \mathcal{N}_r(v), r \in R\}\!\!\}))),$$

where $\text{INIT}_2$, $\text{UP}_2$, $\text{AGG}_2$, and $\text{MSG}$ are differentiable *initialization*, *update*, *aggregation* and *message* functions, respectively.

**Characteristic of fundamental relation.** We also identify a few characteristics of the fundamental relationship from ULTRA when constructing the relation graphs:

1. **Self-loop.** *h2h* and *t2t* always exist for all relation, i.e., $\forall r \in R, h2h(r, r) \land t2t(r, r)$. This will induce two self-loops for all relation.

2. **Symmetric.** Relation *h2h* and *t2t* are *symmetric*: If $h2h(r_1, r_2) \in E_{\text{LIFT}}$ then we must have $h2h(r_2, r_1) \in E_{\text{LIFT}}$, as they satisfy the same assumption for homomorphism. The same applied for *t2t*.

3. **Inverse.** Relation *h2t* and *t2h* are always *inverse relation* to each other, i.e., if $h2t(r_1, r_2) \in E_{\text{LIFT}}$ then we must have $t2h(r_2, r_1) \in E_{\text{LIFT}}$, as they satisfy the same assumption, and vice versa.

As a standard practice, current link prediction models augment the knowledge graph via inverse relations. However, adding inverse relations implies a lot of symmetrical edges in the constructed relation graphs. First, note that for every $r(u, v) \in E$, we have $r^{-1}(v, u) \in E^{-}$. Thus, by default, we have that $t2h(r, r^{-1})$, $h2t(r, r^{-1})$, $h2t(r^{-1}, r)$, $t2h(r^{-1}, r) \in E_{\text{LIFT}}$.

Then we consider what happened in the constructed relation graphs:

- **t2h relations.** Assume $t2h(r_1, r_2) \in E_{\text{LIFT}}$, i.e., $\exists u, v, w \in V, r_1(u, v), r_2(v, w) \in E$, then

$$t2t(r_1, r_2^{-1}), h2h(r_1^{-1}, r_2), h2t(r_1^{-1}, r_2^{-1}) \in E_{\text{LIFT}}$$

- **h2h relation.** Assume $h2h(r_1, r_2) \in E_{\text{LIFT}}$, i.e., $\exists u, v, w \in V, r_1(v, u), r_2(v, w) \in E$, then

$$h2t(r_1, r_2^{-1}), t2h(r_1^{-1}, r_2), t2t(r_1^{-1}, r_2^{-1}) \in E_{\text{LIFT}}$$

- **h2t relation.** Assume $h2t(r_1, r_2) \in E_{\text{LIFT}}$, i.e., $\exists u, v, w \in V, r_1(v, u), r_2(w, v) \in E$, then

$$h2h(r_1, r_2^{-1}), t2t(r_1^{-1}, r_2), t2h(r_1^{-1}, r_2^{-1}) \in E_{\text{LIFT}}$$

- **t2t relation.** Assume $t2t(r_1, r_2) \in E_{\text{LIFT}}$, i.e., $\exists u, v, w \in V, r_1(u, v), r_2(w, v) \in E$, then

$$t2h(r_1, r_2^{-1}), h2t(r_1^{-1}, r_2), h2h(r_1^{-1}, r_2^{-1}) \in E_{\text{LIFT}}$$

## G. Sparse matrix multiplication for relational hypergraph construction

To compute the adjacency matrix for the relational hypergraph, we hereby define the implementation details of lift operation LIFT. We follow a similar strategy from Galkin et al. (2024) and compute the four motifs of ULTRA, *h2t,t2h,t2t,h2h*, via sparse matrix multiplication. We then proceed to show higher-order motif computation presented in MOTIF instances in the experiments.

### G.1. 2-path motifs ($\mathcal{F}_2^{\textbf{path}}$)

Given a knowledge graph $G = (V, E, R, c)$ with $n := |V|$ nodes and $m := |R|$ relations, we represent the (multi-relational) adjacency matrix $\boldsymbol{A} \in \mathbb{R}^{n \times m \times n}$ in sparse format. We can then gather the adjacency matrix by scatter max over the first node dimension and the last node dimension, obtaining two (sparse) matrices $\boldsymbol{E}_h \in \mathbb{R}^{n \times m}$ and $\boldsymbol{E}_t \in \mathbb{R}^{m \times n}$, representing any incoming edges of any relation from any nodes and outgoing edges of any relation from any nodes, respectively. Note that since the adjacency matrix is binary, i.e., all of the coordinates are either 0 or 1, taking scatter max is equivalent to taking logical OR operation. Note that these binary interactions are actually all possible 2-path motifs.

For motif computing interactions between relations are then equivalent to one sparse matrix multiplication (spmm) operation between adjacencies:

$$\boldsymbol{A}_{h2h} = \text{spmm}\left(\boldsymbol{E}_h^T, \boldsymbol{E}_h\right) \in \mathbb{R}^{m \times m}$$
$$\boldsymbol{A}_{t2t} = \text{spmm}\left(\boldsymbol{E}_t^T, \boldsymbol{E}_t\right) \in \mathbb{R}^{m \times m}$$
$$\boldsymbol{A}_{h2t} = \text{spmm}\left(\boldsymbol{E}_h^T, \boldsymbol{E}_t\right) \in \mathbb{R}^{m \times m}$$
$$\boldsymbol{A}_{t2h} = \text{spmm}\left(\boldsymbol{E}_t^T, \boldsymbol{E}_h\right) \in \mathbb{R}^{m \times m}$$

Note that we only take the existence of any edges, thus, the respective edge index is extracted from all non-zero values. Finally, we concatenate all the considered relations types as the final adjacency matrix of $\text{LIFT}_{\mathcal{F}}(G)$ through LIFT operation.

Figure 10. The 3-path motifs used by MOTIF($\mathcal{F}_3^{\text{path}}$) are shown here.

## G.2. 3-path motifs ($\mathcal{F}_3^{\text{path}}$)

Note that it is also possible to compute higher-order $k$-path ($\mathcal{F}_k^{\text{path}}$) motifs via a similar strategy, i.e., via iteratively carrying out spmm over the adjacency matrix $\boldsymbol{A}^{k-2}$. As an example, we first define the motifs considered in the experiment of MOTIF with all the 3-path motifs, namely, *tfh,tft,hfh,hft*, shown in Figure 10.

- tail-forward-head (*tfh*) edges,
  $tfh(r_1, r_2, r_3) \in E_{\text{LIFT}} \iff \exists u, v, w, x \in V, r_1(u, v) \wedge r_2(v, w) \wedge r_3(w, x)$

- tail-forward-tail (*tft*) edges,
  $tft(r_1, r_2, r_3) \in E_{\text{LIFT}} \iff \exists u, v, w, x \in V, r_1(u, v) \wedge r_2(v, w) \wedge r_3(x, w)$

- head-forward-head (*hfh*) edges,
  $hfh(r_1, r_2, r_3) \in E_{\text{LIFT}} \iff \exists u, v, w, x \in V, r_1(v, u) \wedge r_2(v, w) \wedge r_3(x, w)$

- head-forward-tail (*hft*) edges,
  $hft(r_1, r_2, r_3) \in E_{\text{LIFT}} \iff \exists u, v, w, x \in V, r_1(v, u) \wedge r_2(v, w) \wedge r_3(w, x)$

To compute these efficiently via spmm, we follow a similar strategy of generating the multi-relational adjacency matrix $\boldsymbol{A} \in \mathbb{R}^{n \times m \times n}$ and $\boldsymbol{E}_h \in \mathbb{R}^{n \times m}$ and $\boldsymbol{E}_t \in \mathbb{R}^{m \times n}$. Then, with the same strategy, the computation of these interactions among three relations is equivalent to one sparse matrix of size $\mathbb{R}^{m \times m \times m}$:

$$\boldsymbol{A}_{hfh} = \text{spmm}\left(\boldsymbol{E}_h^T, \boldsymbol{A}, \boldsymbol{E}_h\right) \in \mathbb{R}^{m \times m \times m}$$
$$\boldsymbol{A}_{tft} = \text{spmm}\left(\boldsymbol{E}_t^T, \boldsymbol{A}, \boldsymbol{E}_t\right) \in \mathbb{R}^{m \times m \times m}$$
$$\boldsymbol{A}_{hft} = \text{spmm}\left(\boldsymbol{E}_h^T, \boldsymbol{A}, \boldsymbol{E}_t\right) \in \mathbb{R}^{m \times m \times m}$$
$$\boldsymbol{A}_{tfh} = \text{spmm}\left(\boldsymbol{E}_t^T, \boldsymbol{A}, \boldsymbol{E}_h\right) \in \mathbb{R}^{m \times m \times m}$$

The final adjacency matrix of $\text{LIFT}_{\mathcal{F}}(G)$ can be generated analogously by concatenating all the considered relations types. Note that it is also possible to construct even higher-order Path-based motifs or other types of motifs with spmm or alternative efficient computation methods. We leave this as future work.

## H. Scalability analysis

Scalability is generally a concern for KGFMs since they carry out inductive (on node and relation) link prediction between a given pair of nodes, which relies heavily on the structural properties of these nodes as well as the relations (due to the lack of node and relation features). While incorporating more expressive motifs enhances the model's predictive power, it also introduces additional computational costs in terms of time and space complexity. This trade-off between expressiveness and scalability is critical in practical applications and warrants careful consideration. Specifically, we aim to answer the question: **What is the trade-off between expressiveness and scalability for MOTIF? (Q4)**.

To explore this trade-off, we conducted a scalability analysis, as reported in Table 4. All experiments were performed on the FB15k-237 dataset using a batch size of 64 on a single NVIDIA H100 GPU. We report the number of parameters, the

*Table 4.* Scalability analysis of link prediction using motifs applying on FB15k-237: number of edges in constructed relational hypergraphs, number of parameters used, training time per batch, inference time per batch, and GPU memory taken on a single H100. (batch size = 64)

| Model | Motif ($\mathcal{F}$) | #Parameter | #Edges in $\text{LIFT}_{\mathcal{F}}(G)$ | Training Time(s) | Inference Time(s) | GPU Memory(GB) |
|---|---|---|---|---|---|---|
| ULTRA | | 168,705 | 108,240 | 1.19 | 0.066 | 12.87 |
| MOTIF | $\mathcal{F}_3^{\text{path}}$ | 181,185 | 4,288,092 | 3.66 | 2.613 | 17.85 |
| | $\mathcal{F}_2^{\text{path}}$ | 179,649 | 81,180 | 1.80 | 0.100 | 13.14 |
| | $\{h2t\}$ | 178,881 | 27,060 | 1.56 | 0.087 | 13.10 |
| | $\emptyset$ | 178,497 | 0 | 1.48 | 0.065 | 12.88 |

number of edges in the constructed relational hypergraphs, training time per batch, inference time per batch, and GPU memory usage.

### H.1. Impact of motifs on scalability

The number of parameters in the models remains relatively stable as richer motifs are introduced, as the additional learnable parameters grow linearly with $|R|$ (the number of relations) and $d$ (embedding dimension) due to the inclusion of additional *fundamental* relation embeddings. However, the difference between MOTIF and ULTRA is notable: MOTIF includes additional positional encodings in the HCNet to capture edge directionality, leading to slightly more parameters.

The key scalability challenge arises from the rapid increase in the number of edges in the constructed relational hypergraphs $\text{LIFT}_{\mathcal{F}}(G)$ as more complex motifs are introduced. For example: binary edges scale with $O(|V|^2|R|)$, while higher-order motifs such as $\mathcal{F}_3^{\text{path}}$ introduce edges that scale with $O(|V|^3|R|)$. This results in an exponential increase as we consider higher order motifs such as $k$-path or $k$-star in the number of induced hyper-edges, leading to significantly higher computational and memory requirements.

In terms of training and inference time, higher-order motifs (e.g., $\mathcal{F}_3^{\text{path}}$) dramatically increase both training and inference times. For instance, training time per batch for $\mathcal{F}_3^{\text{path}}$ is 3.66 seconds compared to 1.80 seconds for the $\mathcal{F}_2^{\text{path}}$ motif. Additionally, GPU memory consumption also increases with more complex motifs. While $\mathcal{F}_3^{\text{path}}$ consumes 17.85 GB, the simpler motifs such as $\mathcal{F}_2^{\text{path}}$ and $\{h2t\}$ use approximately 13 GB.

### H.2. Optimizing scalability with Triton kernels

To mitigate the scalability bottlenecks, we have included custom implementation via Triton kernel[4] to account for the message passing process on both knowledge graphs and relational hypergraphs, which on average halved the training times and dramatically reduced the space usage of the algorithm (5 times reduction on average). The idea is to not materialize all the messages explicitly as in PyTorch Geometric (Fey & Lenssen, 2019), but directly write the neighboring features into the corresponding memory addresses. Compared with materializing all hyperedge messages which takes $O(k|E|)$ where $k$ is the maximum arity, computing with Triton kernel only is $O(|V|)$ in memory. This will enable fast and scalable message passing on both knowledge graph and relational hypergraphs in MOTIF. As a result, we observe that all of the instance of MOTIF can be run on a 20GB GPU. We have shown our implementation in the provided codebase.

While adding higher-order motifs significantly enhances performance, it also introduces scalability challenges, particularly in terms of memory and computational requirements. Our Triton kernel implementation alleviates some of these issues, but further research is needed to explore more efficient ways to scale models for extremely large knowledge graphs and relational hypergraphs.

## I. Computational complexity

In this section, we present the theoretical computational complexity of the model variants considered in the experiments (**Q4**). Given a knowledge graph $G = (V, E, R, c)$, we have that $|V|, |E|, |R|$ represents the size of vertices, edges, and relation types, respectively. $d$ is the hidden dimension of the model, and $L$ is the number of layers in the model. $m$ denotes the maximum arity of the hyperedges, and in the experiment, it's either 2 or 3.

---

[4]https://github.com/triton-lang/triton

## I.1. Generating relational hypergraph

**2-path ($\mathcal{F}_2^{\textbf{path}}$).** The complexity of generating the relation graph for ULTRA is completely determined via initial scatter max over the node dimension, which has a time complexity of $\mathcal{O}(|V|^2|R|)$, followed by sparse-matrix multiplication over $\boldsymbol{E}_h^T$ or $\boldsymbol{E}_t^T$ and $\boldsymbol{E}_h$ or $\boldsymbol{E}_t$. The time complexity of this operation is $\mathcal{O}(\mathsf{nnz}(\boldsymbol{X}) \times \mathsf{nnz}(\boldsymbol{Y}))$ where $\mathsf{nnz}$ is a function finding the number of non-zero element in sparse matrix $\boldsymbol{X} \in \{\boldsymbol{E}_h^T, \boldsymbol{E}_t^T\}$, $\boldsymbol{Y} \in \{\boldsymbol{E}_h, \boldsymbol{E}_t\}$. Note that if we do not use spmm, this time complexity will blow up to $\mathcal{O}(|V||R|^2)$. Thus, the total complexity for constructing the relation graph in ULTRA will become $\mathcal{O}(|V||R|(|V| + |R|))$.

**3-path ($\mathcal{F}_3^{\textbf{path}}$).** Similarly, for the construction of the relational hypergraph in MOTIF, i.e., all the 3-path and the 2-path, we need another spmm to additionally account for the forward adjacency matrix $\boldsymbol{A}$. The time complexity of this operation under sparse matrix multiplication is then $\mathcal{O}(\mathsf{nnz}(\boldsymbol{X}) \times \mathsf{nnz}(\boldsymbol{A}) \times \mathsf{nnz}(\boldsymbol{Y}))$, $\boldsymbol{X} \in \{\boldsymbol{E}_h^T, \boldsymbol{E}_t^T\}$, $\boldsymbol{Y} \in \{\boldsymbol{E}_h, \boldsymbol{E}_t\}$. Without spmm, this time complexity will become $\mathcal{O}(|V|^2|R|^2 + |V||R|^3)$, which dominates the construction of $\boldsymbol{E}_h, \boldsymbol{E}_t$ together with all the 2-Path motifs and thus also serve as the total complexity.

Additionally, it's important to note that the relational hypergraph can be generated as a preprocessing step during inference. During training, we typically exclude the positive triplets currently being used for training from the training knowledge graph to prevent overfitting. This exclusion often leads to changes in the relational hypergraph structure, requiring us to rebuild the relational hypergraph for each batch. Therefore, the complexity of constructing these relational hypergraphs needs to be considered.

**$m$-Star ($\mathcal{F}_m^{\textbf{star}}$).** In the synthetic experiments, we consider $m$-Star motifs for variety of $m$, ranging from 2 to 8. It is computationally more expensive to generate $m$-Star since there does not exist an spmm computation for fast computation. To compute $m$-Star, we need to go through each node $v$ and check the combination $\binom{\deg(v)}{m}$ and check the existence individually to fill up the table. Thus, the complexity will be $\mathcal{O}(|V|\binom{|R|}{m})$, which is exponential to the number of relations $R$ and thus hard to implement in the relation-rich real-world datasets. As a result, we exclude the motifs of $m$-Star in real-world experiments.

## I.2. C-MPNNs and HC-MPNNs

We take the complexity analysis results from Huang et al. (2023) and Huang et al. (2024), shown in Table 5. In this table, we report both the complexity of a single forward pass as well as the amortized complexity of a query $q(u, v)$.

## I.3. ULTRA and MOTIF

The complexity of ULTRA is thus the combination of relation graph generation with 2-path motifs, applying C-MPNN on the constructed relation graph $\mathrm{LIFT}_{\mathcal{F}}(G)$, and then another C-MPNN on knowledge graphs. (Note that NBFNet is an instance of C-MPNN, shown in Huang et al. (2023)). Thus, given a ULTRA model with $T$ layers on relation encoders and $L$ layers on its entity encoder, the overall complexity of a forward pass is going to be

$$\mathcal{O}(|V||R|(|V| + |R|) + L(|R|^2d + |R|d^2) + L(|E|d + |V|d^2))$$

as the number of edges in the constructed relation graph is bounded by $O(|R|^2)$, and the number of nodes in the relation graph is $O(|R|)$. The complexity of MOTIF equipped with 3-path with $T$ layers on relation encoders and $L$ layers on its entity encoder can be derived in a similar manner:

$$\mathcal{O}(|V|^2|R|^2 + |V||R|^3 + L(|R|^3d + |R|d^2) + L(|E|d + |V|d^2)).$$

However, note that due to spmm, the construction of the relation (hyper)graph is very efficient in practical scenarios. Additionally, we note that the amortized complexity of a query of ULTRA is

$$\mathcal{O}\Big(|R|(|V| + |R|) + \frac{T(|R|^2d + |R|d^2)}{|V|} + L(\frac{|E|d}{|V|} + d^2)\Big)$$

and that for MOTIF equipped with 3-path is

$$\mathcal{O}\Big(|V||R|^2 + |R|^3 + \frac{T(|R|^3d + |R|d^2)}{|V|} + L(\frac{|E|d}{|V|} + d^2)\Big).$$

*Table 5.* Asymptotic runtime complexities for relation and entity encoder.

| Model | Complexity of a forward pass | Amortized complexity of a query |
|---|---|---|
| C-MPNNs | $\mathcal{O}(L(|E|d + |V|d^2))$ | $\mathcal{O}(L(\frac{|E|d}{|V|} + d^2))$ |
| HC-MPNNs | $\mathcal{O}(L(m|E|d + |V|d^2))$ | $\mathcal{O}(L(\frac{m|E|d}{|V|} + d^2))$ |

## J. On adding node and relation features

In general, the task of link prediction on knowledge graphs does not consider any node features. Thus, on the surface, it seems that MOTIF does not take the node features into account, and so do many models (Teru et al., 2020) that utilize the labeling tricks (Zhang et al., 2021) and conditional message passing (Zhu et al., 2021; 2023; Galkin et al., 2024; Huang et al., 2023; 2024).

**Adding node features.** Incorporating node features, such as text embeddings, is relatively simple and can be done by simply concatenating the node feature vector, $\boldsymbol{x}_v$, with the current representation, $\boldsymbol{h}_v$, to produce $\boldsymbol{h}_v^* = [\boldsymbol{h}_v || \boldsymbol{x}_v]$. Since the initialization of MOTIFs only requires satisfying target node distinguishability, concatenating the node features will preserve this property. As a result, all theoretical results remain applicable to MOTIF when node features are included. It's also important to note that this concatenation method has already been successfully employed in knowledge graphs with node features, as demonstrated in Zhang et al. (2021); Galkin et al. (2024).

**Adding relation features.** The same principal can apply since MOTIF explicitly computes the (conditioned) relation representations. Specifically, given any feature vector of relations $\boldsymbol{x}_r$ such as text embeddings, we can consider the augmented relation embedding $\boldsymbol{h}_r^* = [\boldsymbol{h}_r || \boldsymbol{x}_r]$ in place of the original computed relation embeddings $\boldsymbol{h}_r$. We leave how adding relation features could potentially boost the zero-shot and fine-tuned performance of KGFMs as an important future work.

## K. Proof of the counter-example in Section 6.2

We present a counter-example of link prediction in Figure 7, showing that ULTRA cannot distinguish between links $r_3(u, v_1)$ and $r_3(u, v_2)$ whereas MOTIF($\mathcal{F}_3^{\text{path}}$) can correctly distinguish. We first define the knowledge graph of interest $G = (V, E, R, c)$ where

$$V = \{u, x, y, v_1, v_2\} \tag{1}$$
$$R = \{r_1, r_2, r_3\} \tag{2}$$
$$E = \{r_1(y, v_1), r_1(v_1, y), r_2(u, x), r_2(y, v_2), r_2(v_2, y), r_3(x, y), r_3(y, x)\} \tag{3}$$

We first show that ULTRA cannot distinguish between $r_3(u, v_1)$ and $r_3(u, v_2)$. First we note that the LIFT operation defines on motifs of ULTRA, i.e., $\{h2t, t2h, h2h, t2t\}$, will generates an complete relation graph $\text{LIFT}_{\mathcal{F}}(G)$ for $G$. Formally speaking, it holds that $\forall P \in \mathcal{F}, \forall r, r' \in R, P(r, r') \in E_{\text{LIFT}}$.

Now, it is enough to show that ULTRA will assign the same coloring to $(u, v_1)$ and $(u, v_2)$. We first claim that $\boldsymbol{h}_{r_1|r_3}^{(T)} = \boldsymbol{h}_{r_2|r_3}^{(T)}$ for all $T \geq 0$. The base case holds since $\boldsymbol{h}_{r_1|r_3}^{(0)} = \boldsymbol{h}_{r_2|r_3}^{(0)} = \mathbf{1}^d$. For inductive case, assume $\boldsymbol{h}_{r_1|r_3}^{(t)} = \boldsymbol{h}_{r_1|r_3}^{(t)}$, then it holds

that

$$h_{r_1|r_3}^{(t+1)} = \sigma\Big(W^{(t)}\Big[h_{r_1|r_3}^{(0)} \| \sum_{r_{\text{LIFT}} \in R_{\text{LIFT}}} \sum_{r' \in \mathcal{N}_{r_{\text{LIFT}}}(r_1)} h_{r'|r_3}^{(t)} \odot z_{r_{\text{LIFT}}}\Big] + b^{(t)}\Big),$$

$$= \sigma\Big(W^{(t)}\Big[h_{r_1|r_3}^{(0)} \| \sum_{r_{\text{LIFT}} \in R_{\text{LIFT}}} \sum_{r' \in R} h_{r'|r_3}^{(t)} \odot z_{r_{\text{LIFT}}}\Big] + b^{(t)}\Big),$$

$$= \sigma\Big(W^{(t)}\Big[h_{r_2|r_3}^{(0)} \| \sum_{r_{\text{LIFT}} \in R_{\text{LIFT}}} \sum_{r' \in R} h_{r'|r_3}^{(t)} \odot z_{r_{\text{LIFT}}}\Big] + b^{(t)}\Big),$$

$$= \sigma\Big(W^{(t)}\Big[h_{r_2|r_3}^{(0)} \| \sum_{r_{\text{LIFT}} \in R_{\text{LIFT}}} \sum_{r' \in \mathcal{N}_{r_{\text{LIFT}}}(r_2)} h_{r'|r_3}^{(t)} \odot z_{r_{\text{LIFT}}}\Big] + b^{(t)}\Big),$$

$$= h_{r_2|r_3}^{(t+1)}$$

Now it is straightforward to see that the representation of $(u, v_1)$ and $(u, v_2)$ will be identical since the entity encoder of ULTRA cannot distinguish between relation $r_1$ and $r_2$, thus viewing node $v_1$ and $v_2$ isomorphic to each other when conditioned on $u$.

However, we notice that $\text{MOTIF}(\mathcal{F}_3^{\text{path}})$ can distinguish between $r_1$ and $r_2$ by computing $h_{r_1|r_3}^{(t)} \neq h_{r_2|r_3}^{(t)}$, since the generated relational hypergraph will contain an hyperedges $hft(r_2, r_3, r_1)$ by noting the homomorphism on path $u \rightarrow r_2 \rightarrow x \rightarrow r_3 \rightarrow y \rightarrow r_1 \rightarrow v_1$ but it does not contain $hft(r_1, r_3, r_2)$.

## L. Details in synthetic experiments

**Dataset construction.** Given $2 \leq k \leq 7$, and for each knowledge graph $G_i = (V_i, E_i, R_i)$ in $\mathsf{ConnectHub}(k)$, we generate and partition the relations into two classes and a query relation $R_i = P_i \cup N_i \cup \{q\}$ where $P_i = \{p_1, \cdots, p_l\}$, and $N_i = \{n_1, \cdots, n_l\}$ for some $l > k$. Then, we construct each component as follows:

- A *hub* $H_i$ where we have the center node $u$, and for all relation $p \in P_i$, there exists distinct $v \in V_i$ such that the edge $p(v, u)$ exists. Note that the center node $u$ will have a degree larger than $k$ by construction.

- Multiple *positive communities* with relations $P_i$ where for each subset of relations $\tilde{P}_i \subseteq P_i$ satisfying $|\tilde{P}_i| \leq k$, we have a center node $x$ and there will exists distinct nodes $y_1, \cdots, y_{|\tilde{P}_i|}$ such that $p_j(y_j, x)$ exists for all $1 \leq j \leq |\tilde{P}_i|$.

- Multiple *negative communities* with relations $N_i$ where for each subset of relations $\tilde{N}_i \subseteq N_i$ satisfying $|\tilde{N}_i| \leq k$, we have a center node $x$ and there will exists distinct nodes $y_1, \cdots, y_{|\tilde{N}_i|}$ such that $n_j(y_j, x)$ exists for all $1 \leq j \leq |\tilde{N}_i|$.

Finally, the graph $G_i$ is a disjoint union of all these components. The detailed dataset statistics are reported in $Table\ 6$.

**Experiment details.** We use ULTRA with sum aggregation on both relational and entity levels, and similarly in all variants of $\text{MOTIF}(\mathcal{F}_m^{\text{star}})$. The message function on the entity level is chosen to be elemental-wise summation to avoid loss of information from relation types during the first message passing step. We use 2 layers for both ULTRA and all variants of $\text{MOTIF}(\mathcal{F}_m^{\text{star}})$, each with 32 dimension on both relation and entity model, and the Adam optimizer is used with a learning rate of 0.001, trained for 500 epochs in all experiments.

## M. Further experimental details

**Datasets.** In this section, we report the details of all experiments reported in the body of this paper. For pertaining experiments, we train the MOTIF instance on three transductive knowledge graph completion datasets, following Galkin et al. (2024): FB15k237 (Toutanova & Chen, 2015), WN18RR (Dettmers et al., 2018), and CoDEx Medium (Safavi & Koutra, 2020). We then carry out zero-shot and fine-tuning experiments on the following datasets, categorized into three groups:

- **Inductive** $e, r$. Inductive link prediction on both new nodes and new relations. Including 13 datasets in INGRAM (Lee et al., 2023): FB-25, FB-50, FB-75, FB-100, WK-25, WK-50, WK-75, WK-100, NL-0, NL-25, NL-50, NL-75, NL-100;

*Table 6.* Dataset construction parameters for ConnectHub.

| ConnectHub($k$) | # Training/Testing graphs | # Training/Testing triplets | # Relations |
|---|---|---|---|
| $k = 2$ | 7/3 | 360/210 | [3,13] |
| $k = 3$ | 7/3 | 1642/1078 | [4,14] |
| $k = 4$ | 4/2 | 824/1058 | [5,11] |
| $k = 5$ | 1/1 | 112/224 | [6,8] |
| $k = 6$ | 1/1 | 238/476 | [7,9] |

and 10 datasets in MTDEA (Zhou et al., 2023a): MT1 tax, MT1 health, MT2 org, MT2 sci, MT3 art, MT3 infra, MT4 sci, MT4 health, Metafram, FBNELL.

- **Inductive $e$.** Inductive link prediction on nodes only. Including 12 datasets from GraIL (Teru et al., 2020): WN-v1, WN-v2, WN-v3, WN-v4, FB-v1, FB-v2, FB-v3, FB-v4, NL-v1, NL-v2, NL-v3, NL-v4; 4 datasets from INDIGO (Liu et al., 2021): HM 1k, HM 3k, HM 5k, HM Indigo; and 2 datasets from Nodepiece (Galkin et al., 2022): ILPC Small, ILPC Large.

- **Transductive.** Transductive link prediction on seen nodes and seen relations. Including CoDEx Small, CoDEx Large (Safavi & Koutra, 2020), NELL-995 (Xiong et al., 2017), YAGO 310 (Mahdisoltani et al., 2015), WDsinger, NELL23k, FB15k237(10), FB15k237(20), FB15k237(50) (Lv et al., 2020), and Hetionet (Himmelstein et al., 2017).

Full tables of zero-shot performance of MOTIF($\mathcal{F}_3^{\text{path}}$) are presented in Table 7 and Table 8, and full tables of fine-tune performance of MOTIF($\mathcal{F}_3^{\text{path}}$) averaged 3 runs are presented in Table 9, Table 10. We also carry out end-to-end experiments on all considered datasets, with full tables averaging 3 runs presented in Table 11 and Table 12. We report the statistics of these datasets in Table 13, Table 14, and Table 15, respectively. We finally report the detail hyperparameters used in Table 16, and the epoch used for fine-tuning and end-to-end training in Table 17. Note that we do not include all 13 datasets shown in Galkin et al. (2024) due to the size of constructed relational hypergraphs when equipped with 3-path motifs.

**Training.** Following convention, on each knowledge graph and for each triplet $r(u, v)$, we augment the corresponding inverse triplet $r^{-1}(v, u)$ where $r^{-1}$ is a fresh relation symbol. All trained MOTIF and its variants aim to minimize the negative log-likelihood of both positive and negative facts. In line with the *partial completeness assumption* (Galárraga et al., 2013), we generate negative samples by randomly corrupting either the head or the tail entity. The conditional probability of a fact $q(u, v)$ is parameterized as $\mathbb{P}(v \mid u, q) = \sigma(f(\boldsymbol{h}_{v|u,q}^{(L)}))$, where $\sigma$ denotes the sigmoid function and $f$ is a two-layer MLP. We adopted layer-normalization (Ba et al., 2016) and short-cut connection after each aggregation and before applying ReLU on both encoders and a dropout rate of 0.2 on relation encoders only. We discard the edges that directly connect query node pairs to prevent overfitting. The best checkpoint for each model instance is selected based on its performance on the validation set.

Inspired by Sun et al. (2019), we employ *self-adversarial negative sampling*, drawing negative triples from the following distribution, with $\alpha$ serving as the *adversarial temperature*:

$$\mathcal{L}(v \mid u, q) = -\log p(v \mid u, q) - \sum_{i=1}^{k} w_{i,\alpha} \log(1 - p(v_i' \mid u_i', q))$$

Here, $k$ represents the number of negative samples for each positive sample, and $(u_i', q, v_i')$ is the $i$-th negative sample. Finally, $w_i$ is the weight for the $i$-th negative sample, defined as

$$w_{i,\alpha} := \text{Softmax}\left(\frac{\log(1 - p(v_i' \mid u_i', q))}{\alpha}\right).$$

Table 7. Detailed zero-shot inductive link prediction MRR and hits@10 for ULTRA and MOTIF($\mathcal{F}_3^{\text{path}}$).

| | Dataset | ULTRA | | MOTIF | |
| --- | --- | --- | --- | --- | --- |
| | | MRR | H@10 | MRR | H@10 |
| Inductive $e, r$ | FB-100 | **0.449** | **0.642** | 0.428 | 0.628 |
| | FB-75 | **0.403** | 0.604 | 0.399 | **0.614** |
| | FB-50 | **0.338** | 0.543 | **0.338** | **0.546** |
| | FB-25 | **0.388** | **0.640** | 0.384 | **0.640** |
| | WK-100 | **0.164** | **0.286** | **0.164** | 0.282 |
| | WK-75 | 0.365 | 0.537 | **0.366** | **0.540** |
| | WK-50 | **0.166** | **0.324** | 0.163 | 0.314 |
| | WK-25 | **0.316** | **0.532** | 0.311 | 0.493 |
| | NL-100 | **0.471** | **0.651** | 0.438 | 0.647 |
| | NL-75 | **0.368** | **0.547** | 0.314 | 0.512 |
| | NL-50 | **0.407** | **0.570** | 0.373 | 0.532 |
| | NL-25 | **0.395** | **0.569** | 0.348 | 0.498 |
| | NL-0 | **0.342** | **0.523** | 0.324 | 0.497 |
| | MT1 tax | 0.224 | 0.305 | **0.325** | **0.448** |
| | MT1 health | 0.298 | 0.374 | **0.326** | **0.398** |
| | MT2 org | **0.095** | **0.159** | 0.092 | 0.154 |
| | MT2 sci | 0.258 | 0.354 | **0.286** | **0.435** |
| | MT3 art | 0.259 | 0.402 | **0.269** | **0.414** |
| | MT3 infra | 0.619 | 0.755 | **0.658** | **0.786** |
| | MT4 sci | 0.274 | 0.449 | **0.283** | **0.451** |
| | MT4 health | 0.624 | 0.737 | **0.627** | **0.762** |
| | Metafam | 0.238 | 0.644 | **0.344** | **0.829** |
| | FBNELL | **0.485** | 0.652 | 0.468 | **0.664** |
| Inductive $e$ | WN-v1 | 0.648 | 0.768 | **0.682** | **0.778** |
| | WN-v2 | **0.663** | 0.765 | **0.663** | **0.771** |
| | WN-v3 | 0.376 | 0.476 | **0.420** | **0.538** |
| | WN-v4 | 0.611 | 0.705 | **0.640** | **0.718** |
| | FB-v1 | 0.498 | 0.656 | **0.503** | **0.692** |
| | FB-v2 | **0.512** | 0.700 | 0.511 | **0.716** |
| | FB-v3 | 0.491 | 0.654 | **0.500** | **0.692** |
| | FB-v4 | 0.486 | **0.677** | **0.487** | **0.677** |
| | NL-v1 | **0.785** | **0.913** | 0.674 | 0.871 |
| | NL-v2 | 0.526 | 0.707 | **0.564** | **0.769** |
| | NL-v3 | 0.515 | 0.702 | **0.533** | **0.724** |
| | NL-v4 | 0.479 | **0.712** | **0.503** | 0.711 |
| | ILPC Small | **0.302** | 0.443 | 0.295 | **0.444** |
| | ILPC Large | **0.290** | **0.424** | 0.285 | 0.415 |
| | HM 1k | 0.059 | 0.092 | **0.063** | **0.097** |
| | HM 3k | 0.037 | 0.077 | **0.055** | **0.084** |
| | HM 5k | 0.034 | 0.071 | **0.050** | **0.073** |
| | HM Indigo | **0.440** | **0.648** | 0.426 | 0.637 |

*Table 8.* Detailed zero-shot transductive link prediction MRR and hits@10 for ULTRA and MOTIF($\mathcal{F}_3^{\text{path}}$).

| | Dataset | ULTRA | | MOTIF | |
|---|---|---|---|---|---|
| | | MRR | H@10 | MRR | H@10 |
| **Pretrained** | WN18RR | 0.480 | **0.614** | **0.507** | 0.609 |
| | FB15k237 | **0.368** | **0.564** | 0.336 | 0.532 |
| | CoDEx Medium | **0.372** | **0.525** | 0.357 | 0.509 |
| **Transductive** | CoDEx Small | 0.472 | 0.667 | **0.474** | **0.670** |
| | CoDEx Large | 0.338 | **0.469** | **0.339** | **0.469** |
| | NELL-995 | 0.406 | 0.543 | **0.491** | **0.623** |
| | YAGO 310 | **0.451** | **0.615** | 0.441 | 0.607 |
| | WDsinger | 0.382 | 0.498 | **0.397** | **0.514** |
| | NELL23k | **0.239** | **0.408** | 0.220 | 0.384 |
| | FB15k237(10) | **0.248** | **0.398** | 0.236 | 0.384 |
| | FB15k237(20) | **0.272** | **0.436** | 0.259 | 0.422 |
| | FB15k237(50) | **0.324** | **0.526** | 0.312 | 0.508 |
| | Hetionet | **0.257** | 0.379 | 0.256 | **0.383** |

*Table 9.* Detailed finetuned inductive link prediction MRR and hits@10 for ULTRA and MOTIF($\mathcal{F}_3^{\text{path}}$).

| Dataset | ULTRA | | MOTIF | |
|---|---|---|---|---|
| | MRR | H@10 | MRR | H@10 |
| **Inductive $e, r$** FB-100 | $0.444_{\pm.003}$ | $\mathbf{0.643}_{\pm.004}$ | $0.439_{\pm.001}$ | $0.642_{\pm.002}$ |
| FB-75 | $\mathbf{0.400}_{\pm.003}$ | $0.598_{\pm.004}$ | $0.399_{\pm.002}$ | $\mathbf{0.607}_{\pm.002}$ |
| FB-50 | $0.334_{\pm.002}$ | $0.538_{\pm.004}$ | $\mathbf{0.340}_{\pm.002}$ | $\mathbf{0.544}_{\pm.002}$ |
| FB-25 | $0.383_{\pm.001}$ | $\mathbf{0.635}_{\pm.002}$ | $\mathbf{0.388}_{\pm.000}$ | $\mathbf{0.635}_{\pm.002}$ |
| WK-100 | $0.168_{\pm.005}$ | $0.286_{\pm.003}$ | $\mathbf{0.173}_{\pm.003}$ | $\mathbf{0.284}_{\pm.009}$ |
| WK-75 | $\mathbf{0.380}_{\pm.001}$ | $0.530_{\pm.001}$ | $0.371_{\pm.008}$ | $\mathbf{0.535}_{\pm.012}$ |
| WK-50 | $0.140_{\pm.010}$ | $0.280_{\pm.012}$ | $\mathbf{0.160}_{\pm.006}$ | $\mathbf{0.304}_{\pm.002}$ |
| WK-25 | $\mathbf{0.321}_{\pm.003}$ | $\mathbf{0.535}_{\pm.007}$ | $0.317_{\pm.005}$ | $0.505_{\pm.004}$ |
| NL-100 | $0.458_{\pm.012}$ | $\mathbf{0.684}_{\pm.011}$ | $\mathbf{0.464}_{\pm.001}$ | $0.682_{\pm.001}$ |
| NL-75 | $\mathbf{0.374}_{\pm.007}$ | $\mathbf{0.570}_{\pm.005}$ | $0.360_{\pm.001}$ | $0.548_{\pm.004}$ |
| NL-50 | $\mathbf{0.418}_{\pm.005}$ | $\mathbf{0.595}_{\pm.005}$ | $0.414_{\pm.004}$ | $0.573_{\pm.005}$ |
| NL-25 | $\mathbf{0.407}_{\pm.009}$ | $\mathbf{0.596}_{\pm.012}$ | $0.390_{\pm.008}$ | $0.580_{\pm.027}$ |
| NL-0 | $\mathbf{0.329}_{\pm.010}$ | $0.551_{\pm.012}$ | $0.328_{\pm.003}$ | $\mathbf{0.556}_{\pm.007}$ |
| MT1 tax | $0.330_{\pm.046}$ | $0.459_{\pm.056}$ | $\mathbf{0.416}_{\pm.067}$ | $\mathbf{0.522}_{\pm.005}$ |
| MT1 health | $0.380_{\pm.0002}$ | $0.467_{\pm.006}$ | $\mathbf{0.385}_{\pm.002}$ | $\mathbf{0.473}_{\pm.003}$ |
| MT2 org | $0.104_{\pm.007}$ | $\mathbf{0.170}_{\pm.001}$ | $\mathbf{0.106}_{\pm.001}$ | $\mathbf{0.170}_{\pm.003}$ |
| MT2 sci | $0.311_{\pm.010}$ | $0.451_{\pm.042}$ | $\mathbf{0.326}_{\pm.002}$ | $\mathbf{0.520}_{\pm.023}$ |
| MT3 art | $0.306_{\pm.003}$ | $\mathbf{0.473}_{\pm.003}$ | $\mathbf{0.315}_{\pm.013}$ | $0.469_{\pm.004}$ |
| MT3 infra | $0.657_{\pm.008}$ | $0.807_{\pm.007}$ | $\mathbf{0.683}_{\pm.007}$ | $\mathbf{0.827}_{\pm.001}$ |
| MT4 sci | $0.303_{\pm.007}$ | $0.478_{\pm.003}$ | $\mathbf{0.309}_{\pm.001}$ | $\mathbf{0.483}_{\pm.005}$ |
| MT4 health | $\mathbf{0.704}_{\pm.002}$ | $0.785_{\pm.002}$ | $0.703_{\pm.001}$ | $\mathbf{0.787}_{\pm.002}$ |
| Metafam | $0.997_{\pm.003}$ | $\mathbf{1.000}_{\pm.000}$ | $\mathbf{1.000}_{\pm.000}$ | $\mathbf{1.000}_{\pm.000}$ |
| FBNELL | $\mathbf{0.481}_{\pm.004}$ | $0.661_{\pm.011}$ | $\mathbf{0.481}_{\pm.004}$ | $\mathbf{0.664}_{\pm.008}$ |
| **Inductive $e$** WN-v1 | $0.685_{\pm.003}$ | $0.793_{\pm.003}$ | $\mathbf{0.703}_{\pm.005}$ | $\mathbf{0.806}_{\pm.005}$ |
| WN-v2 | $0.679_{\pm.002}$ | $0.779_{\pm.003}$ | $\mathbf{0.680}_{\pm.002}$ | $\mathbf{0.781}_{\pm.009}$ |
| WN-v3 | $0.411_{\pm.008}$ | $0.546_{\pm.006}$ | $\mathbf{0.466}_{\pm.004}$ | $\mathbf{0.590}_{\pm.002}$ |
| WN-v4 | $0.614_{\pm.003}$ | $0.720_{\pm.001}$ | $\mathbf{0.659}_{\pm.003}$ | $\mathbf{0.733}_{\pm.004}$ |
| FB-v1 | $0.509_{\pm.002}$ | $0.670_{\pm.004}$ | $\mathbf{0.530}_{\pm.003}$ | $\mathbf{0.702}_{\pm.001}$ |
| FB-v2 | $0.524_{\pm.003}$ | $0.710_{\pm.004}$ | $\mathbf{0.557}_{\pm.004}$ | $\mathbf{0.744}_{\pm.003}$ |
| FB-v3 | $0.504_{\pm.001}$ | $0.663_{\pm.003}$ | $\mathbf{0.519}_{\pm.001}$ | $\mathbf{0.684}_{\pm.002}$ |
| FB-v4 | $0.496_{\pm.001}$ | $0.684_{\pm.001}$ | $\mathbf{0.508}_{\pm.002}$ | $\mathbf{0.695}_{\pm.002}$ |
| NL-v1 | $\mathbf{0.757}_{\pm.021}$ | $\mathbf{0.878}_{\pm.035}$ | $0.712_{\pm.067}$ | $0.873_{\pm.012}$ |
| NL-v2 | $\mathbf{0.575}_{\pm.004}$ | $0.761_{\pm.007}$ | $0.566_{\pm.003}$ | $\mathbf{0.765}_{\pm.003}$ |
| NL-v3 | $0.563_{\pm.004}$ | $0.755_{\pm.006}$ | $\mathbf{0.580}_{\pm.001}$ | $\mathbf{0.764}_{\pm.001}$ |
| NL-v4 | $0.469_{\pm.020}$ | $0.733_{\pm.011}$ | $\mathbf{0.507}_{\pm.062}$ | $\mathbf{0.740}_{\pm.007}$ |
| ILPC Small | $\mathbf{0.303}_{\pm.001}$ | $\mathbf{0.453}_{\pm.002}$ | $0.302_{\pm.001}$ | $0.449_{\pm.001}$ |
| ILPC Large | $\mathbf{0.308}_{\pm.002}$ | $0.431_{\pm.001}$ | $0.307_{\pm.001}$ | $\mathbf{0.432}_{\pm.001}$ |
| HM 1k | $0.042_{\pm.002}$ | $0.100_{\pm.007}$ | $\mathbf{0.067}_{\pm.009}$ | $\mathbf{0.107}_{\pm.010}$ |
| HM 3k | $0.030_{\pm.002}$ | $0.090_{\pm.003}$ | $\mathbf{0.054}_{\pm.006}$ | $\mathbf{0.103}_{\pm.002}$ |
| HM 5k | $0.025_{\pm.001}$ | $0.068_{\pm.003}$ | $\mathbf{0.049}_{\pm.001}$ | $\mathbf{0.091}_{\pm.001}$ |
| HM Indigo | $\mathbf{0.432}_{\pm.001}$ | $\mathbf{0.639}_{\pm.002}$ | $0.426_{\pm.001}$ | $0.635_{\pm.001}$ |

*Table 10.* Detailed finetuned transductive link prediction MRR and hits@10 for ULTRA and MOTIF($\mathcal{F}_3^{\text{path}}$).

| Dataset | | ULTRA | | MOTIF | |
|---|---|---|---|---|---|
| | | MRR | H@10 | MRR | H@10 |
| **Pretrained** | WN18RR | $0.480_{\pm.000}$ | $0.614_{\pm.000}$ | $\textbf{0.529}_{\pm.002}$ | $\textbf{0.628}_{\pm.001}$ |
| | FB15k237 | $\textbf{0.368}_{\pm.000}$ | $\textbf{0.564}_{\pm.000}$ | $0.357_{\pm.004}$ | $0.550_{\pm.005}$ |
| | CoDEx Medium | $\textbf{0.372}_{\pm.000}$ | $\textbf{0.525}_{\pm.000}$ | $0.361_{\pm.001}$ | $0.517_{\pm.001}$ |
| **Transductive** | CoDEx Small | $\textbf{0.490}_{\pm.003}$ | $\textbf{0.686}_{\pm.003}$ | $0.490_{\pm.001}$ | $0.680_{\pm.003}$ |
| | CoDEx Large | $0.343_{\pm.002}$ | $0.478_{\pm.002}$ | $\textbf{0.355}_{\pm.002}$ | $\textbf{0.489}_{\pm.003}$ |
| | NELL-995 | $0.509_{\pm.013}$ | $\textbf{0.660}_{\pm.006}$ | $\textbf{0.514}_{\pm.006}$ | $0.655_{\pm.002}$ |
| | YAGO 310 | $0.557_{\pm.009}$ | $0.710_{\pm.003}$ | $\textbf{0.603}_{\pm.010}$ | $\textbf{0.735}_{\pm.003}$ |
| | WDsinger | $0.417_{\pm.002}$ | $0.526_{\pm.002}$ | $\textbf{0.423}_{\pm.001}$ | $\textbf{0.532}_{\pm.001}$ |
| | NELL23k | $\textbf{0.268}_{\pm.001}$ | $\textbf{0.450}_{\pm.001}$ | $0.256_{\pm.001}$ | $0.441_{\pm.001}$ |
| | FB15k237(10) | $\textbf{0.254}_{\pm.001}$ | $\textbf{0.411}_{\pm.001}$ | $\textbf{0.254}_{\pm.001}$ | $\textbf{0.411}_{\pm.001}$ |
| | FB15k237(20) | $\textbf{0.274}_{\pm.001}$ | $\textbf{0.445}_{\pm.002}$ | $0.273_{\pm.001}$ | $0.444_{\pm.001}$ |
| | FB15k237(50) | $\textbf{0.325}_{\pm.002}$ | $\textbf{0.528}_{\pm.002}$ | $0.323_{\pm.002}$ | $0.523_{\pm.005}$ |
| | Hetionet | $0.399_{\pm.005}$ | $0.538_{\pm.004}$ | $\textbf{0.446}_{\pm.002}$ | $\textbf{0.575}_{\pm.004}$ |

Table 11. Detailed end-to-end inductive link prediction MRR and hits@10 for ULTRA and MOTIF($\mathcal{F}_3^{\text{path}}$).

| Dataset | | ULTRA | | MOTIF | |
|---|---|---|---|---|---|
| | | MRR | H@10 | MRR | H@10 |
| **Inductive** $e, r$ | FB-100 | $\mathbf{0.425}_{\pm.004}$ | $\mathbf{0.628}_{\pm.003}$ | $0.418_{\pm.002}$ | $0.614_{\pm.001}$ |
| | FB-75 | $0.385_{\pm.001}$ | $\mathbf{0.592}_{\pm.003}$ | $\mathbf{0.389}_{\pm.004}$ | $\mathbf{0.592}_{\pm.007}$ |
| | FB-50 | $0.323_{\pm.005}$ | $0.523_{\pm.006}$ | $\mathbf{0.334}_{\pm.001}$ | $\mathbf{0.532}_{\pm.001}$ |
| | FB-25 | $0.365_{\pm.004}$ | $0.612_{\pm.005}$ | $\mathbf{0.376}_{\pm.001}$ | $\mathbf{0.621}_{\pm.001}$ |
| | WK-100 | $0.159_{\pm.002}$ | $0.269_{\pm.007}$ | $\mathbf{0.163}_{\pm.016}$ | $\mathbf{0.273}_{\pm.013}$ |
| | WK-75 | $\mathbf{0.365}_{\pm.006}$ | $0.501_{\pm.020}$ | $0.354_{\pm.006}$ | $\mathbf{0.504}_{\pm.012}$ |
| | WK-50 | $0.149_{\pm.007}$ | $0.290_{\pm.006}$ | $\mathbf{0.150}_{\pm.008}$ | $\mathbf{0.297}_{\pm.022}$ |
| | WK-25 | $\mathbf{0.305}_{\pm.008}$ | $0.486_{\pm.006}$ | $0.299_{\pm.004}$ | $\mathbf{0.489}_{\pm.011}$ |
| | NL-100 | $0.410_{\pm.010}$ | $0.648_{\pm.008}$ | $\mathbf{0.458}_{\pm.021}$ | $\mathbf{0.663}_{\pm.004}$ |
| | NL-75 | $\mathbf{0.364}_{\pm.006}$ | $0.562_{\pm.004}$ | $0.363_{\pm.001}$ | $\mathbf{0.568}_{\pm.010}$ |
| | NL-50 | $\mathbf{0.416}_{\pm.001}$ | $\mathbf{0.597}_{\pm.003}$ | $0.408_{\pm.002}$ | $0.591_{\pm.008}$ |
| | NL-25 | $\mathbf{0.405}_{\pm.006}$ | $\mathbf{0.612}_{\pm.018}$ | $0.387_{\pm.007}$ | $0.601_{\pm.023}$ |
| | NL-0 | $0.323_{\pm.020}$ | $0.520_{\pm.028}$ | $\mathbf{0.329}_{\pm.001}$ | $\mathbf{0.555}_{\pm.004}$ |
| | MT1 tax | $0.392_{\pm.040}$ | $0.517_{\pm.005}$ | $\mathbf{0.431}_{\pm.045}$ | $\mathbf{0.521}_{\pm.005}$ |
| | MT1 health | $\mathbf{0.378}_{\pm.004}$ | $0.466_{\pm.006}$ | $\mathbf{0.378}_{\pm.001}$ | $\mathbf{0.473}_{\pm.004}$ |
| | MT2 org | $\mathbf{0.100}_{\pm.001}$ | $0.166_{\pm.001}$ | $\mathbf{0.100}_{\pm.001}$ | $\mathbf{0.168}_{\pm.001}$ |
| | MT2 sci | $0.323_{\pm.002}$ | $0.524_{\pm.006}$ | $\mathbf{0.358}_{\pm.013}$ | $\mathbf{0.535}_{\pm.006}$ |
| | MT3 art | $0.302_{\pm.001}$ | $0.459_{\pm.001}$ | $\mathbf{0.313}_{\pm.003}$ | $\mathbf{0.470}_{\pm.006}$ |
| | MT3 infra | $0.671_{\pm.005}$ | $0.815_{\pm.006}$ | $\mathbf{0.687}_{\pm.003}$ | $\mathbf{0.823}_{\pm.002}$ |
| | MT4 sci | $\mathbf{0.290}_{\pm.003}$ | $\mathbf{0.457}_{\pm.005}$ | $\mathbf{0.290}_{\pm.001}$ | $\mathbf{0.457}_{\pm.001}$ |
| | MT4 health | $0.684_{\pm.007}$ | $0.771_{\pm.002}$ | $\mathbf{0.701}_{\pm.001}$ | $\mathbf{0.781}_{\pm.001}$ |
| | Metafam | $\mathbf{0.989}_{\pm.010}$ | $\mathbf{1.000}_{\pm.000}$ | $0.885_{\pm.003}$ | $\mathbf{1.000}_{\pm.000}$ |
| | FBNELL | $\mathbf{0.489}_{\pm.012}$ | $\mathbf{0.678}_{\pm.006}$ | $0.469_{\pm.028}$ | $0.634_{\pm.013}$ |
| **Inductive** $e$ | WN-v1 | $0.668_{\pm.007}$ | $0.788_{\pm.005}$ | $\mathbf{0.679}_{\pm.002}$ | $\mathbf{0.792}_{\pm.006}$ |
| | WN-v2 | $0.296_{\pm.074}$ | $0.597_{\pm.040}$ | $\mathbf{0.684}_{\pm.013}$ | $\mathbf{0.795}_{\pm.006}$ |
| | WN-v3 | $0.332_{\pm.010}$ | $0.530_{\pm.012}$ | $\mathbf{0.428}_{\pm.017}$ | $\mathbf{0.562}_{\pm.013}$ |
| | WN-v4 | $0.622_{\pm.018}$ | $0.709_{\pm.003}$ | $\mathbf{0.635}_{\pm.031}$ | $\mathbf{0.720}_{\pm.016}$ |
| | FB-v1 | $0.506_{\pm.004}$ | $0.672_{\pm.004}$ | $\mathbf{0.524}_{\pm.014}$ | $\mathbf{0.689}_{\pm.013}$ |
| | FB-v2 | $0.524_{\pm.003}$ | $0.723_{\pm.004}$ | $\mathbf{0.537}_{\pm.002}$ | $\mathbf{0.737}_{\pm.002}$ |
| | FB-v3 | $0.502_{\pm.002}$ | $0.670_{\pm.002}$ | $\mathbf{0.509}_{\pm.003}$ | $\mathbf{0.679}_{\pm.001}$ |
| | FB-v4 | $0.487_{\pm.003}$ | $0.678_{\pm.003}$ | $\mathbf{0.494}_{\pm.002}$ | $\mathbf{0.685}_{\pm.005}$ |
| | NL-v1 | $0.671_{\pm.015}$ | $0.799_{\pm.036}$ | $\mathbf{0.777}_{\pm.007}$ | $\mathbf{0.866}_{\pm.013}$ |
| | NL-v2 | $0.526_{\pm.017}$ | $0.745_{\pm.007}$ | $\mathbf{0.561}_{\pm.008}$ | $\mathbf{0.775}_{\pm.010}$ |
| | NL-v3 | $0.563_{\pm.011}$ | $\mathbf{0.758}_{\pm.005}$ | $\mathbf{0.565}_{\pm.009}$ | $0.752_{\pm.009}$ |
| | NL-v4 | $0.463_{\pm.038}$ | $0.703_{\pm.030}$ | $\mathbf{0.512}_{\pm.008}$ | $\mathbf{0.737}_{\pm.008}$ |
| | ILPC Small | $\mathbf{0.296}_{\pm.002}$ | $\mathbf{0.445}_{\pm.004}$ | $0.291_{\pm.001}$ | $0.438_{\pm.002}$ |
| | ILPC Large | $\mathbf{0.290}_{\pm.006}$ | $\mathbf{0.417}_{\pm.013}$ | $0.284_{\pm.001}$ | $\mathbf{0.417}_{\pm.002}$ |
| | HM 1k | $\mathbf{0.029}_{\pm.004}$ | $\mathbf{0.072}_{\pm.006}$ | $0.017_{\pm.001}$ | $0.040_{\pm.002}$ |
| | HM 3k | $0.027_{\pm.001}$ | $0.075_{\pm.003}$ | $\mathbf{0.030}_{\pm.001}$ | $\mathbf{0.089}_{\pm.001}$ |
| | HM 5k | $0.027_{\pm.003}$ | $0.065_{\pm.001}$ | $\mathbf{0.028}_{\pm.001}$ | $\mathbf{0.079}_{\pm.001}$ |
| | HM Indigo | $\mathbf{0.399}_{\pm.001}$ | $\mathbf{0.614}_{\pm.001}$ | $0.369_{\pm.006}$ | $0.587_{\pm.006}$ |

Table 12. Detailed end-to-end transductive link prediction MRR and hits@10 for ULTRA and MOTIF($\mathcal{F}_3^{\text{path}}$).

| Dataset | | ULTRA | | MOTIF | |
|---|---|---|---|---|---|
| | | MRR | H@10 | MRR | H@10 |
| | WN18RR | $0.489_{\pm.005}$ | $0.620_{\pm.001}$ | $\mathbf{0.531}_{\pm.004}$ | $\mathbf{0.636}_{\pm.005}$ |
| | FB15k237 | $\mathbf{0.352}_{\pm.001}$ | $\mathbf{0.546}_{\pm.002}$ | $0.327_{\pm.001}$ | $0.520_{\pm.001}$ |
| | CoDEx Medium | $\mathbf{0.367}_{\pm.003}$ | $\mathbf{0.521}_{\pm.001}$ | $0.364_{\pm.068}$ | $0.518_{\pm.087}$ |
| | CoDEx Small | $0.488_{\pm.001}$ | $\mathbf{0.679}_{\pm.001}$ | $\mathbf{0.490}_{\pm.003}$ | $0.678_{\pm.001}$ |
| | CoDEx Large | $\mathbf{0.332}_{\pm.002}$ | $\mathbf{0.473}_{\pm.001}$ | $0.332_{\pm.001}$ | $0.467_{\pm.001}$ |
| | NELL-995 | $0.509_{\pm.009}$ | $0.646_{\pm.016}$ | $\mathbf{0.510}_{\pm.004}$ | $\mathbf{0.655}_{\pm.009}$ |
| **Transductive** | YAGO 310 | $0.567_{\pm.005}$ | $0.710_{\pm.001}$ | $\mathbf{0.583}_{\pm.001}$ | $\mathbf{0.724}_{\pm.001}$ |
| | WDsinger | $0.413_{\pm.001}$ | $0.525_{\pm.001}$ | $\mathbf{0.421}_{\pm.053}$ | $\mathbf{0.530}_{\pm.051}$ |
| | NELL23k | $0.261_{\pm.001}$ | $0.445_{\pm.002}$ | $\mathbf{0.263}_{\pm.003}$ | $\mathbf{0.447}_{\pm.001}$ |
| | FB15k237(10) | $0.248_{\pm.001}$ | $0.398_{\pm.001}$ | $\mathbf{0.255}_{\pm.001}$ | $\mathbf{0.411}_{\pm.001}$ |
| | FB15k237(20) | $\mathbf{0.272}_{\pm.001}$ | $0.436_{\pm.002}$ | $0.271_{\pm.001}$ | $\mathbf{0.442}_{\pm.001}$ |
| | FB15k237(50) | $\mathbf{0.320}_{\pm.002}$ | $\mathbf{0.523}_{\pm.003}$ | $0.316_{\pm.002}$ | $0.511_{\pm.001}$ |
| | Hetionet | $\mathbf{0.424}_{\pm.003}$ | $\mathbf{0.553}_{\pm.002}$ | $0.419_{\pm.003}$ | $0.548_{\pm.001}$ |

Table 13. Dataset statistics for **inductive**-$e, r$ link prediction datasets. Triples are the number of edges given at training, validation, or test graphs, respectively, whereas Valid and Test denote triples to be predicted in the validation and test graphs.

| Dataset | Training Graph | | | Validation Graph | | | | Test Graph | | | |
|---|---|---|---|---|---|---|---|---|---|---|---|
| | Entities | Rels | Triples | Entities | Rels | Triples | Valid | Entities | Rels | Triples | Test |
| FB-25 | 5190 | 163 | 91571 | 4097 | 216 | 17147 | 5716 | 4097 | 216 | 17147 | 5716 |
| FB-50 | 5190 | 153 | 85375 | 4445 | 205 | 11636 | 3879 | 4445 | 205 | 11636 | 3879 |
| FB-75 | 4659 | 134 | 62809 | 2792 | 186 | 9316 | 3106 | 2792 | 186 | 9316 | 3106 |
| FB-100 | 4659 | 134 | 62809 | 2624 | 77 | 6987 | 2329 | 2624 | 77 | 6987 | 2329 |
| WK-25 | 12659 | 47 | 41873 | 3228 | 74 | 3391 | 1130 | 3228 | 74 | 3391 | 1131 |
| WK-50 | 12022 | 72 | 82481 | 9328 | 93 | 9672 | 3224 | 9328 | 93 | 9672 | 3225 |
| WK-75 | 6853 | 52 | 28741 | 2722 | 65 | 3430 | 1143 | 2722 | 65 | 3430 | 1144 |
| WK-100 | 9784 | 67 | 49875 | 12136 | 37 | 13487 | 4496 | 12136 | 37 | 13487 | 4496 |
| NL-0 | 1814 | 134 | 7796 | 2026 | 112 | 2287 | 763 | 2026 | 112 | 2287 | 763 |
| NL-25 | 4396 | 106 | 17578 | 2146 | 120 | 2230 | 743 | 2146 | 120 | 2230 | 744 |
| NL-50 | 4396 | 106 | 17578 | 2335 | 119 | 2576 | 859 | 2335 | 119 | 2576 | 859 |
| NL-75 | 2607 | 96 | 11058 | 1578 | 116 | 1818 | 606 | 1578 | 116 | 1818 | 607 |
| NL-100 | 1258 | 55 | 7832 | 1709 | 53 | 2378 | 793 | 1709 | 53 | 2378 | 793 |
| Metafam | 1316 | 28 | 13821 | 1316 | 28 | 13821 | 590 | 656 | 28 | 7257 | 184 |
| FBNELL | 4636 | 100 | 10275 | 4636 | 100 | 10275 | 1055 | 4752 | 183 | 10685 | 597 |
| Wiki MT1 tax | 10000 | 10 | 17178 | 10000 | 10 | 17178 | 1908 | 10000 | 9 | 16526 | 1834 |
| Wiki MT1 health | 10000 | 7 | 14371 | 10000 | 7 | 14371 | 1596 | 10000 | 7 | 14110 | 1566 |
| Wiki MT2 org | 10000 | 10 | 23233 | 10000 | 10 | 23233 | 2581 | 10000 | 11 | 21976 | 2441 |
| Wiki MT2 sci | 10000 | 16 | 16471 | 10000 | 16 | 16471 | 1830 | 10000 | 16 | 14852 | 1650 |
| Wiki MT3 art | 10000 | 45 | 27262 | 10000 | 45 | 27262 | 3026 | 10000 | 45 | 28023 | 3113 |
| Wiki MT3 infra | 10000 | 24 | 21990 | 10000 | 24 | 21990 | 2443 | 10000 | 27 | 21646 | 2405 |
| Wiki MT4 sci | 10000 | 42 | 12576 | 10000 | 42 | 12576 | 1397 | 10000 | 42 | 12516 | 1388 |
| Wiki MT4 health | 10000 | 21 | 15539 | 10000 | 21 | 15539 | 1725 | 10000 | 20 | 15337 | 1703 |

*Table 14.* Dataset statistics for inductive-$e$ link prediction datasets. Triples are the number of edges given at training, validation, or test graphs, respectively, whereas Valid and Test denote triples to be predicted in the validation and test graphs.

| Dataset | Rels | Training Graph | | Validation Graph | | | Test Graph | | |
|---|---|---|---|---|---|---|---|---|---|
| | | Entities | Triples | Entities | Triples | Valid | Entities | Triples | Test |
| FB-v1 | 180 | 1594 | 4245 | 1594 | 4245 | 489 | 1093 | 1993 | 411 |
| FB-v2 | 200 | 2608 | 9739 | 2608 | 9739 | 1166 | 1660 | 4145 | 947 |
| FB-v3 | 215 | 3668 | 17986 | 3668 | 17986 | 2194 | 2501 | 7406 | 1731 |
| FB-v4 | 219 | 4707 | 27203 | 4707 | 27203 | 3352 | 3051 | 11714 | 2840 |
| WN-v1 | 9 | 2746 | 5410 | 2746 | 5410 | 630 | 922 | 1618 | 373 |
| WN-v2 | 10 | 6954 | 15262 | 6954 | 15262 | 1838 | 2757 | 4011 | 852 |
| WN-v3 | 11 | 12078 | 25901 | 12078 | 25901 | 3097 | 5084 | 6327 | 1143 |
| WN-v4 | 9 | 3861 | 7940 | 3861 | 7940 | 934 | 7084 | 12334 | 2823 |
| NL-v1 | 14 | 3103 | 4687 | 3103 | 4687 | 414 | 225 | 833 | 201 |
| NL-v2 | 88 | 2564 | 8219 | 2564 | 8219 | 922 | 2086 | 4586 | 935 |
| NL-v3 | 142 | 4647 | 16393 | 4647 | 16393 | 1851 | 3566 | 8048 | 1620 |
| NL-v4 | 76 | 2092 | 7546 | 2092 | 7546 | 876 | 2795 | 7073 | 1447 |
| ILPC Small | 48 | 10230 | 78616 | 6653 | 20960 | 2908 | 6653 | 20960 | 2902 |
| ILPC Large | 65 | 46626 | 202446 | 29246 | 77044 | 10179 | 29246 | 77044 | 10184 |
| HM 1k | 11 | 36237 | 93364 | 36311 | 93364 | 1771 | 9899 | 18638 | 476 |
| HM 3k | 11 | 32118 | 71097 | 32250 | 71097 | 1201 | 19218 | 38285 | 1349 |
| HM 5k | 11 | 28601 | 57601 | 28744 | 57601 | 900 | 23792 | 48425 | 2124 |
| HM Indigo | 229 | 12721 | 121601 | 12797 | 121601 | 14121 | 14775 | 250195 | 14904 |

*Table 15.* Dataset statistics for transductive link prediction datasets. Task denotes the prediction task: $h/t$ is predicting both heads and tails, and $t$ is predicting only tails.

| Dataset | Entities | Rels | Train | Valid | Test | Task |
|---|---|---|---|---|---|---|
| FB15k237 | 14541 | 237 | 272115 | 17535 | 20466 | $h/t$ |
| WN18RR | 40943 | 11 | 86835 | 3034 | 3134 | $h/t$ |
| CoDEx Small | 2034 | 42 | 32888 | 1827 | 1828 | $h/t$ |
| CoDEx Medium | 17050 | 51 | 185584 | 10310 | 10311 | $h/t$ |
| CoDEx Large | 77951 | 69 | 551193 | 30622 | 30622 | $h/t$ |
| NELL995 | 74536 | 200 | 149678 | 543 | 2818 | $h/t$ |
| YAGO310 | 123182 | 37 | 1079040 | 5000 | 5000 | $h/t$ |
| WDsinger | 10282 | 135 | 16142 | 2163 | 2203 | $h/t$ |
| NELL23k | 22925 | 200 | 25445 | 4961 | 4952 | $h/t$ |
| FB15k237(10) | 11512 | 237 | 27211 | 15624 | 18150 | $t$ |
| FB15k237(20) | 13166 | 237 | 54423 | 16963 | 19776 | $t$ |
| FB15k237(50) | 14149 | 237 | 136057 | 17449 | 20324 | $t$ |
| Hetionet | 45158 | 24 | 2025177 | 112510 | 112510 | $h/t$ |

*Table 16.* MOTIF hyper-parameters for pretraining, fine-tuning, and training from end to end.

| | Hyperparameter | MOTIF |
|---|---|---|
| Enc$_1$ | # Layers $T$ | 6 |
| | Hidden dimension | 64 |
| | Dropout | 0.2 |
| Enc$_2$ | # Layers $L$ | 6 |
| | Hidden dimension | 64 |
| | Dec | 2-layer MLP |
| | Dropout | 0 |
| Pre-training | Optimizer | AdamW |
| | Learning rate | 0.0005 |
| | Training steps | 40,000 |
| | Adversarial temperature | 1 |
| | # Negatives | 512 |
| | Batch size | 32 |
| | Training graph mixture | FB15k237, WN18RR, CoDEx Medium |
| Fine-tuning | Optimizer | AdamW |
| | Learning rate | 0.0005 |
| | Adversarial temperature | 1 |
| | # Negatives | 256 |
| | Batch size | 16 |
| End-to-End | Optimizer | AdamW |
| | Learning rate | 0.0005 |
| | Adversarial temperature | 1 |
| | # Negatives | 256 |
| | Batch size | 16 |

*Table 17.* Hyperparameters for fine-tuning MOTIF and from end to end. Full represents a whole epoch with all batches being used.

| Datasets | Finetune | | End-to-End | |
|---|---|---|---|---|
| | Epoch | Batch per Epoch | Epoch | Batch per Epoch |
| FB 25-100 | 3 | full | 10 | full |
| WK 25-100 | 3 | full | 10 | full |
| NL 0-100 | 3 | full | 10 | full |
| MT1-MT4 | 3 | full | 10 | full |
| Metafam, FBNELL | 3 | full | 10 | full |
| FB v1-v4 | 1 | full | 10 | full |
| WN v1-v4 | 1 | full | 10 | full |
| NL v1-v4 | 3 | full | 10 | full |
| ILPC Small | 3 | full | 10 | full |
| ILPC Large | 1 | 1000 | 10 | 1000 |
| HM 1k-5k, Indigo | 1 | 100 | 10 | 1000 |
| FB15k237 | 1 | full | 10 | 2000 |
| WN18RR | 1 | full | 10 | 2000 |
| CoDEx Small | 1 | 4000 | 10 | 4000 |
| CoDEx Medium | 1 | 4000 | 10 | 8000 |
| CoDEx Large | 1 | 2000 | 10 | 4000 |
| NELL-995 | 1 | full | 10 | 2000 |
| YAGO310 | 1 | 2000 | 10 | 4000 |
| WDsinger | 3 | full | 10 | 2000 |
| NELL23k | 3 | full | 10 | 4000 |
| FB15k237(10) | 1 | full | 10 | 2000 |
| FB15k237(20) | 1 | full | 10 | 2000 |
| FB15k237(50) | 1 | 1000 | 10 | 2000 |
| Hetionet | 1 | 4000 | 10 | 2000 |

