# OpenReview forum: "How Expressive are Knowledge Graph Foundation Models?"
_ICML.cc/2025/Conference — ICML 2025 poster_

### Official Review · Reviewer_UN87 · 2025-03-11

**Overall Recommendation:** 1

**Summary:**

The manuscript introduces a Knowledge Graph Foundation Model (KGFM) termed MOTIF, which extends ULTRA’s relation graph into a relational hypergraph using manually defined motifs to incorporate additional information for computing a relation’s conditional representation. The authors theoretically establish a condition under which a motif enhances the model’s expressivity and demonstrate that both larger path motifs and broader star motifs improve the expressiveness of the encodings, though this finding is somewhat trivial. Experimental results indicate that the proposed MOTIF outperforms ULTRA in terms of both expressivity and overall performance on real-world datasets. However, the manuscript does not provide a direct comparison with TRIX, an existing KGFM that also addresses expressive power, raising questions about the significance of its contributions.

## Update after Rebuttal
This reviewer appreciates the authors' detailed responses. However, some concerns remain regarding the proposed method's heuristic nature, practicality and scalability.

**Claims And Evidence:**

Some claims in the manuscript are unclear and lack sufficient supporting evidence:
1. The authors reference [1], yet they assert that their work provides the first rigorous analysis of the expressive power of Knowledge Graph Foundation Models (KGFMs). However, [1] already investigates the expressive power of KGFMs and demonstrates that the proposed model, TRIX, is more expressive than ULTRA in distinguishing triplets. While the authors acknowledge [1], they overlook the contribution of [1] on the expressive power of KGFMs.
2. The authors claim that “relational hypergraphs are a generalization of KGs used to represent higher-arity relational data”. However, existing literature predominantly employs n-ary relational graphs or hyper-relational knowledge graphs for this purpose.
3. Theoretical results presented in the manuscript do not hold for some instances of MOTIF. The authors claim that MOTIF can represent a variant of InGram when the weighted relation graph is replaced. However, InGram requires that INIT_1 and INIT_2 be set to random initialization for all entities and relations. Given that all links are already differentiated through distinct representations of entities and relations, incorporating a motif that cannot be covered via a core-onto homomorphism from any motif already in $\mathcal{F}$ cannot contribute additional expressiveness to the encodings. This indicates that the configuration of MOTIF used to produce theoretical results does not appropriately generalize to InGram.
4. Certain statements in the manuscript are unclear. For example, the authors claim that “while the set of links distinguished by *any instance* in MOTIF($\mathcal{F}$) is the same as those in MOTIF({h2t, h2h, t2t}) or MOTIF({t2h, h2h, t2t}), the ULTRA architecture is a strict subset of MOTIF($\mathcal{F}$), so we cannot directly transfer these results to ULTRA.” in lines 250-255. This implies that ULTRA is a strict subset of MOTIF($\mathcal{F}$) but not an instance of it, which is contradictory. How can model A be a subset of B while simultaneously not being an instance of B?

[1] TRIX: A More Expressive Model for Zero-shot Domain Transfer in Knowledge Graphs, LoG 2024.

**Essential References Not Discussed:**

Although the authors have cited [1], a paper that discusses the expressive power of KGFMs, they do not provide a direct comparison with it. Given that [1] explicitly analyzes the expressive power of KGFMs and demonstrates its superiority over ULTRA in distinguishing triplets, the manuscript should clearly compare its approach with [1], particularly in terms of expressive power. Without this comparison, it is difficult to assess whether the proposed method provides a significant theoretical or practical advancement over existing work.

[1] TRIX: A More Expressive Model for Zero-shot Domain Transfer in Knowledge Graphs, LoG 2024.

**Experimental Designs Or Analyses:**

Yes, I have examined the soundness and validity of the experimental designs and have the following concerns:
1. There is a potential issue related to test leakage. The authors pretrain MOTIF on FB15K-237, WN18RR, and CoDEX Medium. However, it is unclear whether there is any test leakage in the inductive datasets derived from these pretrained datasets, particularly those based on FB15K-237 (e.g., FB-v1\~v4, FB-25\~100) and WN18RR(e.g., WN-v1\~v4). The authors should clarify whether any overlap exists between the training and test sets to ensure the validity of the evaluation. Additionally, the proposed method should be re-evaluated in a setting that explicitly prevents test leakage.
2. A direct comparison with TRIX is necessary, as TRIX outperforms MOTIF in zero-shot setting and achieves comparable performance when fine-tuned. For example, on the inductive $e,r$ setting with 23 graphs, TRIX achieves an MRR of 0.368 in the zero-shot scenario, surpassing MOTIF’s MRR of 0.349. When fine-tuned, both models achieve an MRR of 0.401. These results suggest that TRIX is at least as competitive as MOTIF, raising questions about the significance of the proposed model’s contributions from a practical perspective.

**Methods And Evaluation Criteria:**

The proposed method has the following drawbacks:
1. The proposed method is heuristic. Although the authors establish a condition for motifs that enhances expressive power, verifying whether this condition holds for a given motif requires manually comparing existing motifs with candidate motifs. This limits the practicality and scalability of the approach, particularly when applying it to improve the expressive power.
2. The list of possible motifs is limited. The authors define and utilize only two types of motifs: path motifs and star motifs. Furthermore, only path motifs are employed in experiments on real-world datasets. A more comprehensive analysis should include ablation studies on the selection of motifs and provide additional examples of possible motif types to better assess their impact.

**Other Comments Or Suggestions:**

Typos
1. Section 4, lines 151-152: "We present MOTIF, as a general framework for KGFMs" -> "We present MOTIF, a general framework for KGFMs".
2. Section 4.1., lines 127-128: "and G=(V,E,R) a KG" -> "and G=(V,E,R) be a KG".

**Other Strengths And Weaknesses:**

Strength

The proposed relational hypergraph is a reasonable extension of relation graphs, allowing for the capture of more complex relationships between relations.

Weaknesses
1. Please refer to the concerns and recommendations discussed in ‘Claims and Evidence’, ‘Methods and Evaluation Criteria’, and ‘Essential References Not Discussed’ for further elaboration on weaknesses regarding the manuscript's clarity, theoretical claims, and comparative analysis.
2. Different concepts are notated in the same way. For instance, the authors use a character followed by parentheses (e.g., a()) to represent functions, (hyper)edges, and indicators, which could cause confusion for readers.

**Questions For Authors:**

1. Please provide a direct comparison with TRIX in terms of expressive power and performance.
2. The current description of the theoretical claims is confusing, as it suggests that the theoretical results apply to all instances of MOTIF, while the theoretical findings presented in Section 6.1 do not hold for some instances of MOTIF. Please clarify it.
3. Can the procedure for adding motifs be automated? If so, how might this be achieved?

**Relation To Broader Scientific Literature:**

The paper builds upon prior work on Knowledge Graph Foundation Models (KGFMs), particularly ULTRA, by proposing an extension that aims to enhance expressiveness. ULTRA itself was introduced as a relation graph-based approach for predicting missing links in arbitrary knowledge graphs.

The paper aligns with prior work, such as TRIX, which has already explored the expressive power of KGFMs, demonstrating that models like ULTRA have inherent limitations in distinguishing triplets. The authors claim to extend ULTRA’s expressiveness by leveraging relational hypergraphs, but given prior work [1], the novelty of this contribution requires careful consideration.
Additionally, its reliance on motifs raises questions about scalability and practicality concerns that need to be further investigated.

[1] TRIX: A More Expressive Model for Zero-shot Domain Transfer in Knowledge Graphs, LoG 2024.

**Theoretical Claims:**

I have checked the logical flow of the proof.

---

> ### Author Rebuttal · Authors · 2025-03-31
>
> We note that  TRIX is a **contemporaneous work** first published on 16 Nov 2024, i.e. **within 4 months** of ICML submission. We nevertheless provide answers to each of the raised concerns.
>
> ---
> **Claims And Evidence**
> 1. TRIX [1] is the first to provide an expressivity analysis and we will acknowledge this. However, none of our results can be derived from the TRIX paper. While [1] analyzes the expressivity of TRIX in relation to existing models, it does not provide a general tool for studying expressive power and therefore offers limited insight for evaluating future KGFMs. In contrast, the motivation of our work is to provide a more general framework (MOTIF), which captures existing KGFMs (e.g. ULTRA) and their extensions with higher-order motifs. Our theoretical results advance the understanding of KGFMs beyond known architectures.
>
> 2. These graph models serve distinct purposes and generalize KGs to higher-arity data. While MOTIF uses relational hypergraphs, designing KGFM architectures using n-ary or hyper-relational KG is an interesting avenue for the future.
>
> 3. (Q2) To clarify, the variant of InGram we discussed assumes that both initializations are node invariants, as stated in **Remark 1** and **Remark 2**. Thus, the theoretical results we present only apply under these conditions and do not apply to InGram using random initialization because this initialization breaks node invariance.  We will emphasize this to avoid confusion.
>
> 4. The confusion stems from the ambiguous use of "ULTRA", which refers both to a framework and a specific architecture. To clarify: ULTRA as a **framework** is a subset of MOTIF—every instance it defines is also a MOTIF instance. ULTRA as an **architecture** (evaluated empirically) is one such instance. We’ll revise the manuscript to make this distinction explicit.
>
> ---
> **Methods**
> 1. (Q3) Current KGFMs typically use small motifs (e.g., short paths or stars), making manual verification straightforward.  The motif-selection process via core-onto homomorphisms can also be automated as shown in modern database engines: EmptyHeaded [Aberger et al., TODS 2017] used generation of small motifs followed by automated homomorphism checks.
>
> 2. We performed an ablation study (Table 2) showing that richer motifs -- from 3-paths down to none -- consistently worsen expressivity and performance. As noted in **App. F**, we only focus on path motifs for real-world datasets due to their efficient construction via sparse matrix multiplication. While our framework can generalize to arbitrary motifs using precomputation of SQL queries and database engines, we leave empirical exploration to future work due to practical constraints.
>
> ---
> **Experimental Designs**
> 1. We follow prior work (ULTRA) by pretraining on the same three datasets and setup to ensure fair comparison.
>
> - There is no test leakage: all relation graphs in the train set and zero-shot inference set are different, as well as the original graph structures. While some datasets are derived from others, this only affects models with entity/relation-specific embeddings, which ULTRA and MOTIF do not use. Moreover, extracting subsets alters neighborhood structures enough to become significantly different from their larger transductive counterparts.
>
>  - Nevertheless, we did pretraining solely on YAGO310 and evaluated averaged zero-shot performance over all inductive datasets for ULTRA and MOTIF(3-path), to re-confirm the consistent performance improvement:
>
>   | | MRR|H@10|
> |-|-|-|
> |ULTRA|0.350|0.522|
> |MOTIF|**0.358**|**0.530**|
>
> 2. (Q1) Our main experimental goal is to validate that richer motifs enhance expressivity within the MOTIF framework (**Q1, Sec. 6**), rather than chasing SOTA results. We use ULTRA for comparison since it is a MOTIF instance with binary motifs, making it ideal for testing our theoretical claims.
> MOTIF and TRIX differ fundamentally in expressiveness: each can distinguish relation pairs the other cannot, rendering them incomparable:
> - TRIX can implicitly count homomorphism matches, whereas MOTIF only checks for existence. E.g., in a KG with E = {r₁(u₁,v₁), r₂(v₁,w₁), r₃(u₂,v₂), r₄(v₂,w₂), r₃(u₃,v₃), r₄(v₃,w₃)}, TRIX distinguishes (r₁, r₂) from (r₃, r₄) based on their frequency, while MOTIF cannot.
> - MOTIF supports arbitrary motifs, such as the PARA with edges {α(x,y), β(x,y)} from RMPI paper, enabling finer structural distinctions. In a KG with E = {r₁(x₁,x₂), r₂(x₁,x₂), r₁(x₃,x₄), r₂(x₃,x₄), r₃(y₁,y₂), r₄(y₁,y₄), r₃(y₃,y₂), r₄(y₃,y₄)}, TRIX cannot distinguish (r₁, r₂) from (r₃, r₄) due to isomorphic relation graphs under h2h/t2t, while MOTIF can distinguish them via PARA, which maps only to (r₁, r₂).
>
> Incorporating higher-order motifs into TRIX could similarly boost its expressivity, highlighting our framework as a **complementary** way for improving KGFMs.
>
> ---
> **Weaknesses**
>
> We will refine our notation to ensure clarity and distinctness among these different concepts in the revised paper.

---

### Official Review · Reviewer_C2GP · 2025-03-12

**Overall Recommendation:** 2

**Summary:**

This paper presents a modified design for existing KGFMs, incorporating arbitrary motifs rather than being limited to binary motifs, as in previous approaches. This enhancement increases expressive power. Synthetic and real-world experiments validate the proposed improvements.

**Claims And Evidence:**

Yes. However, some unclear definitions may impact the comprehension of these claims to some extent, as outlined in the 'Other Weaknesses' section later.

**Essential References Not Discussed:**

N/A

**Experimental Designs Or Analyses:**

Yes, especially since I find the synthetic experiments interesting, as they effectively demonstrate the modification's inferior performance. It would be beneficial for the authors to provide similar examples using real-world data to better justify the necessity of this modification.

**Methods And Evaluation Criteria:**

Somehow, yes. The problem formulation is overly general, and the motivation behind the proposed modification does not sufficiently justify its necessity. Specifically, more explanations or concrete examples are needed to illustrate the limitations of binary motifs, particularly in the context of downstream tasks.

**Other Comments Or Suggestions:**

1. In line 56, it would be helpful to provide a clearer explanation of the term "binary motifs," as it appears to be a key concept.
2. More generally, how are different entities and relations matched across various KGs? Additionally, it would strengthen the paper if the authors could provide support or evidence for the effectiveness of "structural similarities" (line 47) to better ground KGFM in the introduction.
3. A minor suggestion: In lines 111–113, it may be clearer to use distinct notations for factual links and potential links.

**Other Strengths And Weaknesses:**

## Weaknesses:
1. The definition of relational hypergraphs is problematic. For instance, R and V are ambiguously defined. It would be helpful to explicitly specify the domain and range of all functions involved.
2. The sentence between lines 139 and 142 is unclear. Additionally, more intuitive explanations of link/relation invariants are needed.
3. More generally, the writing could be improved for better clarity, particularly in explaining the method (e.g., through additional examples). Furthermore, the paper should provide deeper insights into the intuition behind the modification and the necessity of implementing it.
4. The experimental results on real-world data are not very convincing. It would be beneficial to better demonstrate the necessity of the proposed approach.

**Questions For Authors:**

N/A

**Relation To Broader Scientific Literature:**

KGFMs are inherently closely connected to scientific literature. The proposed modification can be viewed as an enhancement that improves analogical reasoning across different scenarios by incorporating more complex structural patterns in KGs.

**Theoretical Claims:**

I did not thoroughly verify the proofs, as the theoretical claims are not critical to this paper. My primary concerns lie in the motivations."

---

> ### Author Rebuttal · Authors · 2025-03-31
>
> > *“**Methods**: ...The problem formulation is overly general, ... ”*
>
> The reason our problem formulation is deliberately general is that our primary motivation is to rigorously analyze and broadly improve the expressive power of existing KGFMs, including widely used models like ULTRA. Our theoretical framework (MOTIF) thus intentionally generalizes beyond specific architectures.
>
> > *“**Experimental Designs**: …It would be beneficial for the authors to provide similar examples using real-world data ….”*
> > *“W4. The experimental results on real-world data are not very convincing....”*
>
> To provide more concrete insight into why higher-order motifs are necessary, we specifically analyzed scenarios where higher-order motifs excel in downstream tasks. Empirically, we have conducted Synthetic Experiments in **Table 1** to pinpoint the exact place where binary motifs are not enough, proven by the failure of ULTRA on all constructed datasets. For real-world experiments, we found that MOTIF, equipped with higher-order motifs, significantly outperforms binary motifs on datasets containing relatively few relations, such as WordNet-based datasets, which contain only around 10 relations. This can be reflected in the average performance from WN v1-WN v4 here from zero-shot and end-to-end results:
>
> |Setting|Model|MRR|H@10|
> |-|-|-|-|
> |Zero-shot|ULTRA|0.575|0.679|
> |Zero-shot|MOTIF|**0.601**|**0.701**|
> |End-to-End|ULTRA|0.480|0.656|
> |End-to-End|MOTIF|**0.607**|**0.717**|
>
> In **Sec. 7.3**, we conduct a detailed investigation showing that these cases revealed that binary motifs (as used by ULTRA) often construct relation graphs with structurally very similar nodes, thereby failing to distinguish different relations adequately, as shown in **Fig. 8**. In contrast, higher-order motifs frequently break such node invariances, leading to more discriminative and informative relation representations. We will discuss these and demonstrate the necessity of higher-order motifs using real-world examples.
>
> >*“**Theoretical Claims**: I did not thoroughly verify the proofs, as the theoretical claims are not critical to this paper..."*
>
> We would like to respectfully emphasize that one of the primary motivations and key contributions of this work is precisely to **establish a rigorous and systematic theoretical framework to analyze and understand the expressive power of existing KGFMs**, such as ULTRA. We believe these theoretical results significantly deepen our fundamental understanding of why certain KGFM variants outperform others in practice, and thus we respectively suggest that they should not be overlooked.
>
> ---
> **Strengths and Weaknesses:**
>
>
> >*“W1. The definition of relational hypergraphs is problematic… W2. … more intuitive explanations of link/relation invariants are needed. W3. More generally, the writing could be improved for better clarity,...”*
>
> We thank the reviewer for carefully pointing out these areas for improvement. We agree that the clarity and precision of our definitions, explanations, and examples can be improved for better presentation. We will explicitly clarify the definitions of relational hypergraphs (W1), provide clearer and more intuitive explanations for link/relation invariants (W2), and enhance overall readability with additional illustrative examples (W3).
>
> >“W4. “The experimental results on real-world data are not very convincing....”
>
>  Please see  our detailed response on experimental design and analysis.
>
> ---
> **Other comments:**
>
> >*“S1. In line 56, it would be helpful to provide a clearer explanation of the term "binary motifs," as it appears to be a key concept.”*
>
> We will clarify this in the revised manuscript by explicitly defining a binary motif as a motif $P = (G_M, \bar{r})$ with a motif graph $G_M = (V_M, E_M, R_M)$ containing exactly two relation types ($|R_M|=2$).
>
> >*“S2. More generally, how are different entities and relations matched across various KGs? ...”*
>
> Our framework inherently matches similar relations across different KGs by constructing similar relational hypergraphs based on their structural roles. Specifically, relations with structurally similar contexts in different KGs will yield similar embeddings thanks to conditional MPNN since they can effectively capture the induced similar hypergraph neighborhoods without manual matching of entities and relations IDs. The same principle applies to entities: entities embedded in structurally similar local neighborhoods involving similar relations will naturally obtain similar embeddings across KGs.
>
> **Fig. 1** in our manuscript already illustrates how structurally similar relations across different KGs (e.g., provide ↔ supply, research ↔ produce) receive similar embeddings due to similar relational contexts. We will further expand this figure with additional explanatory details in the revised manuscript to ground this key concept.
>
> >*“S3. A minor suggestion: In lines 111–113...”*
>
> We will modify these in the updated manuscripts.

---

> > ### Comment · Reviewer_C2GP · 2025-04-03
> >
> > Dear Authors,
> >
> > Thank you for your responses. I appreciate the theoretical contributions of your submission. However, I find that the clarity of the paper still requires significant effort to follow.
> >
> > Despite these considerations, I have decided to maintain my original score due to the following concerns:
> >
> >
> > 1. Alignment between theory and practical knowledge graphs: While the main text presents only the average performance, a closer look at the appendix (Tables 7–10) reveals that different datasets exhibit distinct properties—some aligning with MOTIF, while others do not. This suggests that improved expressiveness does not necessarily lead to better performance on downstream tasks. Unlike synthetic experiments, where graph structures are transparent, real-world knowledge graphs vary significantly, requiring a more comprehensive discussion of these differences.
> > 2. Transferability of structural insights across domains: Closely related to the previous point, it is crucial to clarify what kind of information can be reliably transferred between knowledge graphs from different domains. Figure 1 does not sufficiently establish the validity of structural similarity, as the dashed lines may not hold in other contexts, even if the other connections are preserved.
> >
> >
> > Additionally, as noted by Reviewer UN87, proper acknowledgment of prior work is essential. Addressing this issue thoroughly will likely require more time to ensure due credit is given.
> >
> >
> > Best regards,

---

> > > ### Author Response · Authors · 2025-04-07
> > >
> > > >*1. Alignment between theory and practical knowledge graphs...”*
> > >
> > > We understand the reviewer’s concern and generally agree with this point. In fact, this is exactly why we experimented additionally with synthetic datasets to precisely validate the theoretical expressiveness. There are many confounding factors when experimenting on real-world data, which makes it harder to validate the theory, while still possible. In our case, we find that the expressiveness gains of MOTIF **DO** translate into real-world improvements, particularly on datasets with richer structural patterns that binary motifs cannot capture.
> > >
> > > Specifically, **Metafam** is a dataset deliberately constructed to capture **conflicting and compositional relational patterns**, commonly presented in real-world multi-relational settings (e.g., social or genealogical graphs). Consistent with our theoretical insights, MOTIF achieves significantly better zero-shot performance on these datasets—most notably, a **45% improvement on Metafam** over baselines.
> > >
> > > Similarly, in **WIKITOPICS-MT**, where different knowledge subgraphs are aggregated to form multi-task scenarios, MOTIF outperforms existing models by effectively leveraging structural cues across topics, as shown below:
> > >
> > > |Model|Avg. MRR|Avg. Hits@10|
> > > |-|-|-|
> > > |ULTRA|0.331| 0.442|
> > > |MOTIF|**0.358**| **0.481**|
> > >
> > > Please note that this result is the average across 8 datasets, and in many cases gains are much more prominent, eg, on MT1 tax MOTIF shows 50% relative improvement.
> > > This suggests that enhanced expressiveness is directly beneficial when the dataset structure aligns with MOTIF’s capabilities.
> > >
> > > We agree that not all datasets exhibit such structure, which is why we include a broad set of benchmarks with 54 datasets to illustrate this variability (whereas other papers often provide comparison only on the datasets they show gains). Nonetheless, the strong performance gains on structurally expressive datasets provide concrete evidence that our theoretical contributions can meaningfully impact real-world tasks. We will elaborate further on these dataset-specific differences and clarify the practical implications of model expressiveness in the revised manuscript.
> > >
> > >
> > >
> > > > *“2. Transferability of structural insights across domains...”*
> > >
> > >
> > > We would like to emphasize that **Figure 1 is not intended to claim that the dashed link (e.g., produce(Intel, SemiConductors)) universally holds across all knowledge graphs sharing similar structure**. Rather, it illustrates that **structural motifs provide the model with richer contextual evidence**, *enabling it to learn whether such a link is likely to hold based on observed patterns during training*.
> > >
> > > We fully agree that structural similarity does not guarantee link existence. Our point is that, with expressive motifs, the model can now better recognize when a candidate link appears in a similar structural context to those seen in training—even across disjoint relational vocabularies—and learn to distinguish between cases where the link should or should not be predicted.
> > >
> > > In this sense, we **do not assume the transferability of individual links** but rather propose that **relation invariants derived from structural motifs offer a more nuanced basis for generalization across domains**. We will revise the manuscript to clarify this and prevent any misinterpretation.
> > >
> > >
> > > > *“Additionally, as noted by Reviewer UN87, proper acknowledgment of prior work is essential. Addressing this issue thoroughly will likely require more time to ensure due credit is given.”*
> > >
> > > As discussed in our detailed response to Reviewer UN87, **TRIX [1] is a contemporary work that was first publicly released on November 16, 2024**, less than four months before our own submission. With that being said, we **acknowledge TRIX as the first to provide an analysis of expressive power** within its specific modeling setup. We have also provided the **fundamental expressive differences** between TRIX and MOTIF, rendering them incomparable.
> > >
> > > However, our work offers a principled framework (MOTIF) for studying and designing expressive KGFMs that strictly **extend** the capabilities of existing models such as ULTRA and others. Moreover, we emphasize the versatility of our approach of using arbitrary motifs: our framework can be easily adapted to extend the capabilities of other models that are not directly comparable to ULTRA, such as TRIX. We believe this is an interesting avenue for future research. We also clarify that **none of our theoretical results can be derived from the TRIX paper**, as TRIX lacks the general tools we develop for analyzing and improving expressiveness through arbitrary motifs.
> > >
> > > We will revise the manuscript to acknowledge TRIX explicitly and clarify distinctions where necessary.

---

### Official Review · Reviewer_EEM2 · 2025-03-13

**Overall Recommendation:** 4

**Summary:**

The authors in this paper introduce MOTIF, a framework for enhancing expressiveness of KGFMs. The authors have conducted a rigorous, theoretical study on the expressive power of KGFMs that have been designed to generalize to unseen KGs with different relational vocabs, making them highly useful for inductive link prediction. They argue that previous methods rely mostly on binary motifs, limiting their generalization capability and in order to address this, they propose MOTIF which allows for higher-order relational motifs that enable richer relation-interactions.Yes, the authors rigorously support their claim that the choice of motifs determines the expressive power of KGFMs.

**Claims And Evidence:**

Yes, the authors rigorously support their claim that the choice of motifs determines the expressive power of KGFMs.

**Essential References Not Discussed:**

None.

**Experimental Designs Or Analyses:**

Yes.
1.	Which motif type contributes the most?
2.	Was there an ablation study conducted?
3.	Can smaller graph modifications affect the performance?
4.	How should a reader comprehend the upper bound of the motif complexity? At what upper bound does the performance start to degrade?

**Methods And Evaluation Criteria:**

Yes, but how much overhead does MOTIF introduce? Are these human-readable? Do MOTIFs scale to large-scale KGs without affecting the performance?

**Other Comments Or Suggestions:**

There are some minor typos that would need to be revised to improve readability, such as, “because this tuples”, “Fist we note”.

**Other Strengths And Weaknesses:**

The authors in this paper present a strong argument based on theoretical and empirical insights. This can be a strong contribution to the KGFM domain. I would like to understand the flexibility of incorporating the framework into existing KGs.

**Questions For Authors:**

Q1: How much overhead does MOTIF introduce?

Q2: What is the computational cost of using denser, richer motifs?

Q3: Are motifs interpretable by humans?

Q4: Is there an upper bound to adding motifs that start to deteriorate the performance?

**Relation To Broader Scientific Literature:**

The authors extend prior work KGFMs by introducing MOTIF, a framework that generalizes these models using higher-order relational motifs to improve inductive link prediction and relation generalization.

**Theoretical Claims:**

Yes, however, what is the lower bound on the computational costs? Also, the denser the KGs are, the more chances are that there would be noise. So how does the framework adapt to this? The argument assumes that more expressive models lead to better generalization.

---

> ### Author Rebuttal · Authors · 2025-03-31
>
> We thank the Reviewer for considering ours a "strong contribution".
>
> >*“**Methods.** Yes, but how much overhead does MOTIF introduce? Are these human-readable? Do MOTIFs scale to large-scale KGs without affecting the performance?”*
>
> >*“Q1: How much overhead does MOTIF introduce?”*
>
> This question is **Q4** in our experimental section. A detailed comparison with theoretical computational complexity and practical scalability is shown in **App. H** and **App. G**.
>
> While introducing higher-order motifs will rapidly increase the number of constructed hyperedges in relational hypergraphs, the MOTIF instances considered in experiments remain feasible to scale effectively to large-scale KGs with improved performance, thanks to our custom implementation of message-passing with Triton kernels that largely reduce the computation overheads.
>
> > *“Q3: Are motifs interpretable by humans?”*
>
> If this refers to interpretability, motifs correspond directly to small and well-defined subgraph patterns, such as paths or stars, which describe specific patterns appearing among relations to help capture the semantic meaning of relations. E.g., a "h2h" motif aims to identify whether the heads of two relations semantically share the same node type.
>
> > “**Theoretical Claims.** Yes, however, what is the lower bound on the computational costs? Also, the denser the KGs are, the more chances are that there would be noise. So how does the framework adapt to this?...”
>
> We have provided a detailed analysis of the computational complexity in **App. H**. To alleviate noisy KGs, the framework can be easily adapted by adding regularization known for GNNs, such as edge dropout. In fact, we have conducted preliminary experiments using edge dropout, and MOTIF maintains stable performance, though the gains from these robustness techniques were marginal.
>
>
> >*“**Experimental Designs.** Yes. 1. Which motif type contributes the most? 2. Was there an ablation study conducted? 3. Can smaller graph modifications affect the performance? 4. How should a reader comprehend the upper bound of the motif complexity? At what upper bound does the performance start to degrade?”*
>
> >*“Q2: What is the computational cost of using denser, richer motifs?”*
>
> >*“Q4: Is there an upper bound to adding motifs that start to deteriorate the performance?”*
>
> Thank you for these questions. We address each below:
>
> - *“1. Which motif type contributes the most? 2. Was there an ablation study conducted?”*
>
> We have performed an ablation study that systematically removed motifs, transitioning from the full set of 3-path motifs to simpler sets (2-path motifs, single motif "h2t", and no motifs), as shown in **Table 2** of the main paper. We observe that even with a single motif ("h2t"), the model retains some predictive ability, as essential structural information necessary for relation representation is still captured. However, richer motifs consistently provide the strongest performance improvements due to enhanced expressivity. We are happy to include additional experiments over other binary motifs “h2h” and “t2t” in the updated manuscripts.
>
> - *“3. Can smaller graph modifications affect the performance?”*
>
> We are not entirely sure what the reviewer refers to as our method does not modify the graph structure. Nevertheless, to validate  whether perturbation of the KG will affect the performance, we have experimented by randomly dropping half of the edges each time during training. We observed that this did not lead to noticeable performance degradation.
>
> - *“4. How should a reader comprehend the upper bound of the motif complexity? Q2: What is the computational cost of using denser, richer motifs?”*
>
> Regarding motif complexity, computing and storing higher-order motifs incur additional complexity, which scales with respect to the number of relations. For instance, to compute k-path motifs, we would need a computation complexity of $O(|V||R|^{k-1}(|V|+|R|))$ without using sparse matrix multiplications. We include our detailed explanation again in **App. H**.
>
> - *“Q4: Is there an upper bound to adding motifs that start to deteriorate the performance?”*
>
> Theoretically, adding motifs that can not be mapped via a core-onto homomorphism to an existing one will provably be more expressive. Empirically, whether such additions lead to performance improvements or deterioration depends significantly on the dataset and specific task characteristics.  Intuitively, introducing an excessive number of motifs could reduce the signal-to-noise ratio in the constructed relation hypergraphs, potentially deteriorating the model's performance.
>
> >*“**Other Strengths And Weaknesses.** … I would like to understand the flexibility of incorporating the framework into existing KGs.”*
>
> Our framework is flexible and has been tested on 54 KGs from various domains and task settings, as shown in the experimental section.
>
> >*“**Other Comments.** ...minor typos ...”*
>
> We will modify this in the updated manuscripts.

---

> > ### Comment · Reviewer_EEM2 · 2025-04-02
> >
> > Thank you for addressing my questions and concerns. I will maintain my recommendation.

---

### Decision · Program_Chairs · 2025-05-01

**Decision:**

Accept (poster)

**Comment:**

The paper delves into the expressiveness of Knowledge Graph Foundation Models (KGFMs) and proposes a novel framework called MOTIF to enhance their expressive power by utilizing richer motifs beyond binary interactions.
The authors rigorously establish the theoretical underpinnings of their proposed framework and empirically validate their claims through extensive experimentation, demonstrating the framework's effectiveness in capturing complex relational structures and its superiority over existing models like ULTRA.

The idea to introduce higher order motifs to improve the expressiveness of KGFMs makes sense to me.  I have similar concerns about the additional computing cost as Reviewer EEM2, which has been addressed by the authors in the response.

The authors addresses potential concerns in the rebuttals regarding the comparison with concurrent work, TRIX, clarifying that while TRIX is a relevant and contemporaneous work, it is not mandatory to directly compare with it given the ICML policy on concurrent work.

The authors also addressed the reviewers' concerns by providing detailed responses and revisions to the manuscript, addressing issues such as the framework's scalability, the practicality of the proposed method, and the necessity for more intuitive explanations and examples.